# Aerosol pH and its influencing factors in Beijing

**Jing Ding[2, 1], Pusheng Zhao[1*], Jie Su[1], Qun Dong[1], Xiang Du[2, 1], and Yufen Zhang[2]**

[1] Institute of Urban Meteorology, China Meteorological Administration, Beijing 100089, China

[2] State Environmental Protection Key Laboratory of Urban Ambient Air Particulate Matter Pollution

Prevention and Control, College of Environmental Science and Engineering, Nankai University,

Tianjin 300071, China

**\* Correspondence:** P. S. Zhao (pszhao@ium.cn)

**Abstract**

9        The acidity or pH is an important feature of ambient aerosol. At present, the aerosol pH in the

North China Plain, either seasonal variation or size-resolved characteristics, need to be further
studied. In addition, it is also worthy of discussion about what factors have a greater impact on pH
and how these factors affect pH. In view of these, the hourly water-soluble ions ($SO_4^{2-}$, $NO_3^-$, $Cl^-$,
$NH_4^+$, $Na^+$, $K^+$, $Mg^{2+}$, and $Ca^{2+}$) of $PM_{2.5}$ and trace gases (HCl, $HNO_3$, $HNO_2$, $SO_2$, and $NH_3$) were
online measured by a MARGA system in four seasons during 2016 and 2017 in Beijing.
Furthermore, the size-resolved aerosol was also sampled by a MOUDI sampler and analyzed for the
chemical compositions of different sizes. On the basis of these data, the particle hydronium ion
concentration per volume air ($H_{air}^+$), aerosol liquid water content (ALWC), and $PM_{2.5}$ pH were
calculated by using ISORROPIA-II. Moreover, the sensitivities of $H_{air}^+$, ALWC, aerosol pH to all
the main influencing factors were discussed. In Beijing, the $PM_{2.5}$ pH over four seasons showed
moderately acid. The $PM_{2.5}$ acidity in NCP was both driven by aerosol composition and particle
water. The sensitivity analysis revealed that $SO_4^{2-}$, T, $NH_4^T$, and RH (only in summer) are crucial
factors affecting the $PM_{2.5}$ pH. The $SO_4^{2-}$ had a key role for aerosol acidity, especially in winter and
spring. The impact of $NO_3^-$ on $PM_{2.5}$ pH was different in four seasons. Although $NH_3$ in the NCP
was abundant, the $PM_{2.5}$ pH was far from neutral, which mainly attributed to the limited ALWC.
Elevated $Ca^{2+}$ concentration could increase the aerosol pH because of the buffering capacity of $Ca^{2+}$
to the acid species and the weak water solubility of $CaSO_4$. The sensitivity analysis also implied
that decreasing $NO_3^T$ could reduce the $\varepsilon(NH_4^+)$ effectively. In contrast, the nitrate response to $NH_4^T$
control was highly nonlinear. According to the size-resolved results, the pH for coarse mode, which
was near or even higher than 7, was much higher than that for fine mode. It must be noted that the
aerosol pH in coarse mode showed a marked decrease when under heavily polluted condition.
***Key words:*** Aerosol pH, ISORROPIA-II, Influencing factors, Beijing

## 1. Introduction

Acidity or pH, which drives many processes related to particle composition, gas-aerosol partitioning and aerosol secondary formation, is an important aerosol property (Jang et al., 2002; Eddingsaas et al., 2010; Surratt et al., 2010). The aerosol acidity has a significant effect on the aerosol secondary formation through the gas-aerosol partitioning of semi-volatile and volatile species (Pathak et al., 2011a; Guo et al., 2016). Recent studies have shown that aerosol acidity could promote the generation of secondary organic aerosol by affecting the aerosol acid-catalyzed reactions (Rengarajan et al., 2011). Moreover, metals can become soluble by acid dissociation under lower aerosol pH (Shi et al., 2011; Meskhidze et al., 2003) or by forming a ligand with organic species, such as oxalate at higher pH (Schwertmann et al., 1991). In addition, higher aerosol acidity can lower the acidification buffer capacity and affects the formation of acid rain. The investigation of aerosol acidity is conducive to better understand the important role of aerosols in acid deposition and atmospheric chemical reactions.

The hygroscopic components in the aerosols include water-soluble inorganic ions and part of organic acid (Peng, 2001; Wang et al., 2017). The deliquescence relative humidity (DRH) for the mixed-salt is lower than that of any single component (Seinfeld and Pandis, 2016), hence the ambient aerosols are generally droplets containing liquid water. The aerosol pH actually is the pH of the aerosol liquid water. The aerosol acidity is frequently estimated by the charge balance of measurable cations and anions. A net negative balance correlated with an acidic aerosol and vice versa (Zhang et al., 2007; Pathak et al., 2011b; Zhao et al., 2017). Generally, a larger value of the ion balance implies a stronger acidity or stronger alkaline. Nevertheless, an ion balance or other similar proxies fail to represent the true aerosol pH because they cannot predict $H^+$ concentration in the liquid phase accurately (Guo et al., 2015; Hennigan et al., 2015), which could be calculated by hydrogen ion concentration per volume air ($H_{air}^+$) and the aerosol liquid water content (ALWC).

It is critical to obtain the ALWC in calculating aerosol acidity. One way to calculate the ALWC is based upon the assumption that the volume of ALWC is equal to subtracting the volume of dry aerosol particles from that of wet particles (Guo et al., 2015; Bian et al. 2014; Engelhart et al. 2011). Under this assumption, ALWC could be calculated by the size-resolved hygroscopic growth factors ($g(D, RH)$) combining particle size distribution (PNSDs) or by the hygroscopic growth factor of

aerosol scattering coefficient ($f$(RH)) (Bian et al. 2014; Guo et al., 2015; Kuang et al., 2017a). The $g$(D, RH), defined as the ratio of the diameter of the wet particle at a certain relative humidity to the corresponding diameter at dry conditions, can be measured by a H-TDMA (Hygroscopic Tandem Differential Mobility Analyzer) (Liu et al., 1978; Swietlicki et al., 2008; Liu et al., 2011). The $f$(RH) can be observed by the wet & dry nephelometer system (Covert et al., 1972; Rood et al. 1985; Yan et al., 2009; Kuang et al., 2016, 2017b).

Another way to calculate the ALWC is based on the aerosol chemical components with thermodynamic models, such as ISORROPIA-II, AIM, ADDEM etc. (Nenes et al., 1998; Fountoukis and Nenes, 2007, Clegg et al., 1998, Topping et al., 2005a, b). Based on the aerosol chemical components as well as temperature and relative humidity, the aerosol thermodynamic models can output both ALWC and $H_{air}^+$, which offers a more precise approach to acquire aerosol pH (Pye et al., 2013). Among these thermodynamic models, ISORROPIA and ISORROPIA-II are widely used owing to its rigorous calculation and performance on computational speed. ISORROPIA simulates the gas-particle partitioning in the $H_2SO_4$, $NH_3$, $HNO_3$, $HCl$, $Na^+$, $H_2O$ system, while its second version, ISORROPIA-II, adds $Ca^{2+}$, $K^+$, $Mg^{2+}$ and the corresponding salts to the simulated particle components in thermodynamic equilibrium with water vapor and gas-phase precursors.

Comparisons were made in some studies to investigate the consistency of calculated ALWC derived from the above methods. In the North China Plain (NCP), Bian et al. (2014) found that the ALWC calculated using size-resolved hygroscopic growth factors and the PNSD agreed well with that calculated using ISORROPIA II at higher relative humidity (>60%). Relatively good consistency was also found in the study of Engelhart et al. (2011) in the USA based on the similar method. Guo et al. (2015) compared the ALWC calculated by $f$ (RH) with the total predicted water by organics and inorganics. The total predicted water was highly correlated and on average within 10 % of the $f$ (RH) measured water. Though good consistencies in ALWC were found among these methods, the $H_{air}^+$ could only be obtained by the thermodynamic models, which had been applied to predict aerosol acidity in many studies (Nowak et al., 2006; Fountoukis et al., 2009; Weber et al., 2016; Fang et al., 2017).

The characteristics of aerosol chemical components are different among multiple size ranges.

Among inorganic ions, $SO_4^{2-}$, $NO_3^-$, $Cl^-$, $K^+$, $NH_4^+$ mainly concentrate in fine mode except for the
dust days (Meier et al., 2009; Pan et al., 2009; Tian et al., 2014), whereas $Mg^{2+}$, $Ca^{2+}$ are abundant
in coarse mode (Zhao et al., 2017). The aerosol acidity is affected by coupling among many variables.
Therefore, it could be expected that the aerosol pH is also diverse under different particle size. The
gas precursor ($NH_3$, $HNO_3$, and $HCl$) of main water-soluble ions, as well as ambient temperature
and relative humidity, are also important factors affecting the aerosol acidity. In some countries
where particle matter concentration is very low, the pH diurnal variation was mainly driven by
meteorological conditions (Guo et al., 2015, 2016; Bougiatioti et al., 2016). In China, however, the
annual average $PM_{2.5}$ concentration in some megacities was ~2 times higher than the national
standard value (35 μg m$^{-3}$) and the inorganic ions accounted for 40%~50% to $PM_{2.5}$, especially in
the North China Plain (Zou et al., 2018; Huang et al., 2017; Gao et al., 2018). Hence it can be
expected that the aerosol composition is also a crucial factor on pH, which cannot be ignored.
The North China Plain is the region with the most severe aerosol pollution in China. Nevertheless,
only a few studies have focused on aerosol pH in this region. Some studies conducted in NCP
showed that the aerosol acidity was close to neutral, while in some other studies the fine particles
showed moderately acidic (Cheng et al., 2016; Wang et al., 2016; Liu et al., 2017; Shi et al., 2017).
These results were all significantly higher than that in the United States or Europe, where aerosols
were often highly acidic with a pH lower than 3.0 (Guo et al., 2015, 2016; Bougiatioti et al., 2016;
Weber et al., 2016; Young et al., 2013). The differences in aerosol pH in NCP mainly resulted from
the different methods (ion balance & thermodynamic equilibrium models) or different data sets.
Moreover, the variation of $PM_{2.5}$ chemical composition in NCP in recent years also contributed to
the differences in aerosol pH. The observations in previous studies exploring aerosol acidity in NCP
were almost conducted before 2015. In the recent three years, the chemical composition of $PM_{2.5}$ in
Beijing has undergone tremendous changes. Nitrate has replaced sulfate and is dominant in
inorganic ions in most cases (Zhao et al., 2017; Huang et al., 2017; Ma et al., 2017). Moreover,
studies about seasonal variation of aerosol pH and size-resolved aerosol pH are rare in NCP, and the
key factors affecting aerosol acidity are still not well understood.
In this work, thermodynamic model ISORROPIA-II with the forward mode was utilized to
predict ALWC and aerosol pH in Beijing. The hourly measured $PM_{2.5}$ inorganic ions and precursor
gases in four seasons during 2016 to 2017 were used to analyze the seasonal and diurnal variation
of aerosol acidity, and the sensitivity analysis was conducted to identify the key factors that affecting
the aerosol pH. In our previous studies, the multi-stage cascade impactors (MOUDI-122) were used
for size-resolved aerosol sampling from 2013 to 2015. The actual relative humidity inside the
impactors was calculated, and the size distributions of water-soluble ions, organic carbon, and
elemental carbon in three seasons were discussed (Zhao et al., 2017; Su et al., 2018). Based on these
size-resolved results, the pH for aerosol in different size ranges could also be predicted.
**2. Data Collection and Methods**
**2.1 Site**
The measurements were performed at the Institute of Urban Meteorology in Haidian district of
Beijing (39°56'N, 116°17'E). The sampling site was located next to a high-density residential area,
without significant air pollution emissions around the site. Therefore, the observation data could
represent the air quality levels of the urban area of Beijing.
**2.2 Online data collection**
Water-soluble ions ($SO_4^{2-}$, $NO_3^-$, $Cl^-$, $NH_4^+$, $Na^+$, $K^+$, $Mg^{2+}$, and $Ca^{2+}$) of $PM_{2.5}$ and trace gases
(HCl, $HNO_3$, $HNO_2$, $SO_2$, and $NH_3$) in the ambient air were measured by an online analyzer
(MARGA) at hourly temporal resolution during the spring (April and May in 2016), winter
(February in 2017), summer (July and August in 2017) and autumn (September and October in
2017). The more details about MARGA can be found at ten Brink et al. (2007). The $PM_{2.5}$ and $PM_{10}$
mass concentrations (TEOM 1405DF), the hourly ambient temperature and relative humidity were
also synchronously attained.
Hourly concentrations of $PM_{2.5}$, $PM_{10}$, and water-soluble ions in $PM_{2.5}$, as well as meteorological
parameters during the observation, are shown in Figure 1. In the spring, two dust events occurred
(21-22, April and 5-6, May). During the first dust events, the wind came predominantly from the
north with mean wind speed 3.5 m s$^{-1}$. The $PM_{10}$ concentration reached 425 μg m$^{-3}$ while the $PM_{2.5}$
concentration was only 46 μg m$^{-3}$ on the peak hour. Similarly, the second dust event resulted from
the strong wind coming from the northwest direction. In the following pH analysis based on
MARGA data, it was assumed that the particles were internally mixed, and the chemical
compositions were the same for particles of different sizes in $PM_{2.5}$. Hence, these two dust events
were excluded from this analysis.
**Figure 1**
**2.3 size-resolved chemical compositions**
A Micro-Orifice Uniform Deposit Impactor (MOUDI-120) was used to collect size-resolved
aerosol samples with the calibrated 50% cut sizes of 0.056, 0.10, 0.18, 0.32, 0.56, 1.0, 1.8, 3.1, 6.2,
9.9 and 18 μm. Size-resolved sampling was conducted during July 12-18, 2013; January 13-19,
2014; July 3-5, 2014; October 9-20, 2014; and January 26-28, 2015. Fifteen, fourteen, and eighteen
sets of samples were obtained for the summer, autumn, and winter, respectively. Except for two sets
of samples, all the samples were collected in daytime (from 08:00 to 19:00) and nighttime (from
20:00 to 7:00 the next day), respectively. One hour of preparation time was set for filter changing
and nozzle plate washing with ethanol. The water-soluble ions were analyzed from the samples by
using an ion chromatography (DIONEX ICS-1000). The detailed information about the features of
MOUDI-120, and the procedures of sampling, pre-treatment, and laboratory chemical analysis
(including the quality assurance & quality control) were described in our previous papers (Zhao et
al., 2017; Su et al., 2018). It should be noted that there was no observation of gas precursors during
the periods of MOUDI sampling.
**2.4 Aerosol pH prediction**
As mentioned in the Introduction, pH of ambient aerosols can be predicted by the thermodynamic
model such as AIM and ISORROPIA. AIM is considered as an accurate benchmark model while
ISORROPIA has been optimized for use in chemical transport models. Currently, ISORROPIA-II,
adding $K^+$, $Mg^{2+}$, and $Ca^{2+}$ (Fountoukis and Nenes, 2007), can calculate the equilibrium $H_{air}^+$
(particle hydronium ion concentration per volume air) and ALWC with reasonable accuracy by
taking water-soluble ions mass concentration, temperature, and relative humidity as input. The $H_{air}^+$
and ALWC were then used to predict aerosol pH by the Eq. (1).
$$pH = -\log_{10} H_{aq}^+ \cong -\log_{10} \frac{1000 H_{air}^+}{ALWC_i} \tag{1}$$
Where $H_{aq}^+$ (mole $L^{-1}$) is the hydronium ion concentration in the ambient particle liquid water. $H_{aq}^+$
can also be deemed to be the $H_{air}^+$ (μg $m^{-3}$) divided by the concentration of ALWC associated with
inorganic species, $ALWC_i$ (μg $m^{-3}$). Both inorganic and part of organic species in particles are
hygroscopic. However, the pH prediction is not highly sensitive to the water uptake by organic
species ($ALWC_o$) (Guo et al., 2015, 2016). The similar result was also found in Beijing in Liu et al.
(2017). Hence the aerosol pH could be fairly predicted by ISORROPIA-II with just measurements
of inorganic species in most cases. However, it should be noted that the potential error could be
incurred by ignoring $ALWC_o$ in regions where hygroscopic organic species has a relatively high
contribution to fine particles.
In ISORROPIA-II, forward and reverse mode are provided to predict ALWC and $H_{air}^+$. In forward
mode, T, RH, and the total (i.e. gas+aerosol) concentrations of $NH_3$, $H_2SO_4$, HCl, and $HNO_3$ need
to be input. Reverse mode calculates the equilibrium partitioning given the concentrations of only
aerosol compositions together with RH and T as input. In this work, the online ion chromatography
MARGA was used to measure both inorganic ions of $PM_{2.5}$ and precursor gases. Moreover, several
studies had shown that the ion balance and reverse-mode calculations of thermodynamic
equilibrium models were not applicable to interpret the aerosol acidity (Hennigan et al., 2015; Liu
et al. 2017; Song et al., 2018). The forward mode was also reported less sensitive to measurement
error than the reverse mode (Hennigan et al., 2015; Song et al., 2018). Hence, ISORROPIA-II was
run in the "forward mode" for aerosols in the metastable condition in this study.
When using ISORROPIA-II to calculate the $PM_{2.5}$ acidity, all particles were assumed internally
mixed and the bulk properties were used, without considering the variability of chemical
compositions with particle size. In the ambient atmosphere, the aerosol chemical composition is
complicated, hence the deliquescent relative humidity of aerosol is generally low (Seinfeld and
Pandis, 2016) and the particles usually exist in the form of droplets, which makes the assumption
that the particles are in a liquid state (metastable condition) reasonable. However, when the particles
are exposed to a quite low RH, the state of particles may change. Figure 2 and Figure S1-S4 exhibit
the comparisons between predicted and measured $NH_3$, $HNO_3$, HCl, $NH_4^+$, $NO_3^-$, $Cl^-$, $\varepsilon(NH_4^+)$
($NH_4^+/(NH_3+NH_4^+)$, mol/mol), $\varepsilon(NO_3^-)$ ($NO_3^-/(HNO_3+NO_3^-)$, mol/mol)), and $\varepsilon(Cl^-)$ ($Cl^-/(HCl+Cl^-)$,
mol/mol) based on real-time ion chromatography data, which are all colored by the corresponding
RH. It can be seen that agreements between predicted and measured $NH_3$, $NH_4^+$, $NO_3^-$, and $Cl^-$ are
pretty well, the $R^2$ of linear regressions are all higher than 0.94, and the slopes are around 1.
Moreover, the agreement between predicted and measured $\varepsilon(NH_4^+)$ is better when compared with

206    $\varepsilon(NO_3^-)$ and $\varepsilon(Cl^-)$. The slope of linear regression between predicted and measured $\varepsilon(NH_4^+)$ was

207    0.93, 0.91, 0.95, and 0.96 and the $R^2$ is 0.87, 0.93, 0.89, and 0.97 in spring, winter, summer, and

208    autumn, respectively. However, measured and predicted partitioning of $HNO_3$ and HCl show

209    significant discrepancies ($R^2$ of 0.28 and 0.18), which may attribute to the much lower gas

210    concentrations compared with the particle concentrations, as well as the gas denuder measurement

211    uncertainties from particle collection artifacts (Guo et al., 2018). Obviously, more scatter points

212    deviate from the 1:1 line when ISORROPIA-II runs at RH≤30%, which is much evident in winter

213    and spring. For data with RH ≤ 30%, the predictions are significantly improved when assuming

214    aerosol in stable mode (solid + liquid) (Figure S5-S6). However, the aerosol liquid water was almost

215    zero and cannot be used to predict aerosol pH. It reveals that it is not reasonable to predict the

216    aerosol pH using the thermodynamic model when the RH is relatively low. Consequently, we only

217    discussed the $PM_{2.5}$ pH for data with RH higher than 30% in this work.

218               **Figure 2**

219     Running ISORROPIA-II in the forward mode with only aerosol concentrations as input may

220    result in a bias in predicted pH due to repartitioning of ammonia in the model, leading to a lower

221    predicted pH when gas-phase data are not available (Hennigan et al., 2015). In this work, since no

222    gas phase was available for the size-resolved pH prediction. We determined aerosol pH through an

223    iteration procedure that used the measured particulate species and ISORROPIA-II to predict gas

224    species, the detailed information could be found in Fang et al. (2017) and Guo et al. (2016). As a

225    brief summary, the predicted $NH_3$, $HNO_3$, and HCl concentrations from the $i$-1 run were applied to

226    the $i$th iteration, until the gas concentrations converged. Based on these iterative gas phase

227    concentrations, the ion concentrations from samples collected by the MOUDI as well as the

228    averaged RH and T during each sampling period were used to determine aerosol pH for different

229    size ranges. Just like calculating the pH of $PM_{2.5}$, it was also assumed that all the particles at each

230    size bin were internally mixed and had the same pH.

231     The comparisons of iterative and predicted $NH_3$, $HNO_3$, and HCl as well as measured and predicted

232    $NO_3^-$, $NH_4^+$, $Cl^-$, $\varepsilon(NH_4^+)$, $\varepsilon(NO_{3-})$, and $\varepsilon(Cl^-)$ for data from MOUDI samples are showed in

233    Figure 3. The previous study showed that coarse mode particles were very difficult to reach

234    equilibrium with the gaseous precursors due to kinetic limitations (Dassios et al., 1999; Cruz et al.,

2000). Assuming coarse mode particles in equilibrium with the gas phase could result in a large bias
between measured and predicted $NO_3^-$ and $NH_4^+$ in coarse mode particles (Fang et al, 2017). We
also find that in this work, it can be clearly seen that assuming coarse mode particles in equilibrium
with the gas phase could overpredict $NO_3^-$ and $Cl^-$ and underestimate $NH_4^+$ in the coarse mode (the
blue scatters), which could subsequently underestimate the coarse mode aerosol pH. Compared with
the coarse mode particles, the measured and predicted $NO_3^-$, $NH_4^+$, and $Cl^-$ agreed very well in fine
mode particles. Considering the kinetic limitations and nonideal gas-particle partitioning in coarse
mode particles, the aerosol pH in coarse mode was determined by ignoring the gas phase.
**Figure 3**
**2.5 Sensitivities of aerosol pH to $SO_4^{2-}$, $NO_3^T$, $NH_4^T$, $Cl^T$, RH, and T**
In the real ambient air, the thermodynamic process of the aerosol is complicated, it is not easy to
tell the effect of one factor on the aerosol pH. The ALWC, $H_{air}^+$, aerosol pH, $\varepsilon(NH_4^+)$, $\varepsilon(NO_3^-)$, and
$\varepsilon(Cl^-)$ are all the output of ISORROPIA-II. Together, they reflect an objective state of particles.
Considering the relative independence between input parameters, it is reasonable to discuss the
influence of input variables on output parameters with the results of ISORROPIA-II. Thus, in this
paper, we focus on the sensitivity analysis of single-factor variation, which can reflect the variation
tendency of aerosol pH caused by the change of each variable.
In the ISORROPIA-II, the input parameters include $SO_4^T$ (total sulfate (gas+aerosol) expressed
as equivalent $H_2SO_4$), $NO_3^T$ (total nitrate (gas+aerosol) expressed as equivalent $HNO_3$), $NH_4^T$ (total
ammonium (gas+aerosol) expressed as equivalent $NH_3$), $Cl^T$ (total chloride (gas+aerosol) expressed
as equivalent HCl), $Na^+$, $Ca^{2+}$, $K^+$, $Mg^{2+}$, RH, and T. After running, the gas and aerosol phase of
$NO_3^T$, $NH_4^T$, and $Cl^T$ would be reapportioned and output. In view of this, it is more reasonable to
analyze the impact of $NO_3^T$, $NH_4^T$, and $Cl^T$ on aerosol pH, rather than the impact of a single gas or
aerosol phase of $NO_3^T$, $NH_4^T$, and $Cl^T$ on aerosol pH. In addition, the mass concentration of $K^+$ and
$Mg^{2+}$ was low, so the variables in the sensitivity analysis were determined as $SO_4^{2-}$, $NO_3^T$, $NH_4^T$,
$Cl^T$, $Ca^{2+}$, RH, and T. When assessing how a variable affects ALWC, $H_{air}^+$, and aerosol pH, the real-
time measured values of this variable and the averaged values of other variables in each season were
input ISORROPIA-II. The magnitude of the relative standard deviation (RSD) of calculated aerosol
pH can reflect the impact of one variable on the aerosol acidity. The higher the RSD, the greater the
impact, vice versa. The average value and variation range for each variable in all four seasons are
listed in Table S1 and Figure S7.
The sensitivity analysis in this work aimed at the $PM_{2.5}$ (*ie* fine particles) because the $PM_{2.5}$
components in four seasons were available and had a high temporal resolution (1h). In addition, the
data set had a wide range, covering different levels of haze events. Noted that the sensitivity analysis
in this work only reflected the characteristics during the observation periods, further work is needed
to determine whether the sensitivity analysis is valid in other environments.
**3. Results and Discussion**
**3.1 Overall summary of $PM_{2.5}$ pH over four seasons**
The averaged $PM_{2.5}$ concentrations were 62±36, 60±69, 39±24, and 59±48 μg m$^{-3}$ for observation
periods of spring, winter, summer, and autumn, respectively (Table 1). Among all ions measured,
$NO_3^-$, $SO_4^{2-}$, and $NH_4^+$ were three dominant species, accounting for 83% ~ 87% of total ions.
Compared with other seasons, the averaged concentration of primary inorganic ions ($Cl^-$, $Na^+$, $K^+$,
$Mg^{2+}$, $Ca^{2+}$) was higher in spring. The aerosol in Beijing showed the moderate acidity with $PM_{2.5}$
pH of 4.0±1.0, 4.5±0.7, 3.8±1.2, and 4.3±0.8 for spring, winter, summer, and autumn observation,
respectively (data at RH ≤30% were excluded). The overall winter $PM_{2.5}$ pH was comparable to the
result found in Beijing, 4.2 from Liu et al. (2017) and 4.5 from Guo et al. (2017), but lower than
that (4.9, winter and spring) in Tianjin (Shi et al., 2017), another mega city about 120 km away from
Beijing. The summer $PM_{2.5}$ pH was lowest among all four seasons. The seasonal variation of $PM_{2.5}$
pH in this work was similar to the result from Tan et al. (2018) except for spring, which was winter
(4.11 ± 1.37) > autumn (3.13 ± 1.20) > spring (2.12 ± 0.72) > summer (1.82 ± 0.53). Noted that the
observation in Tan et al. (2018) was conducted in Beijing in 2014, the distinction in the aerosol
compositions was probably responsible for the lower $PM_{2.5}$ pH in their work.
**Table 1**
To further investigate the $PM_{2.5}$ pH performance under different pollution levels over four seasons,
the $PM_{2.5}$ concentrations were classified into three groups with 0~75 μg m$^{-3}$, 75~150 μg m$^{-3}$,
and >150 μg m$^{-3}$, representing the clean, polluted, and heavily polluted conditions, respectively. The
relationship between $PM_{2.5}$ and its pH is shown in Figure S8. The $PM_{2.5}$ pH under clean condition
spanned 2~7 while the $PM_{2.5}$ pH under polluted and heavily polluted conditions mostly concentrated

in 3~5. Table 1 shows that as the air quality deteriorated, aerosol components, as well as ALWC and $H_{air}^+$, all increased for each season, but the differences in $PM_{2.5}$ pH for three pollution levels were not statistically significant. In terms of the averaged values, the $PM_{2.5}$ pH under the clean condition was the highest (Table 1), then followed by polluted and heavily polluted conditions in spring, summer, and autumn. In winter, however, the averaged pH under polluted condition (4.8±1.0) was the highest, then followed by clean (4.5±0.6) and heavily polluted conditions (4.4±0.7).

Time series of mass fraction of $NO_3^-$, $SO_4^{2-}$, $NH_4^+$, $Cl^-$, and crustal ions ($Mg^{2+}$ and $Ca^{2+}$) in total ions, as well as pH in all four seasons, are showed in Figure 4. It can be seen that on clean days, high $PM_{2.5}$ pH (>6) was generally companied by high mass fraction of crustal ions, while the relatively low $PM_{2.5}$ pH (<3) was companied by high mass fraction of $SO_4^{2-}$ and low mass fraction of crustal ion, which was most obvious in summer (large part of $PM_{2.5}$ pH with RH≤30% were excluded in spring and winter). On polluted and heavily polluted days, the aerosol chemical composition was similar, mainly dominated by $NO_3^-$, hence the differences of $PM_{2.5}$ pH on polluted and heavily polluted days were small. Compared with the mass concentration of $PM_{2.5}$, the different aerosol chemical compositions might be the essence that drove aerosol acidity. The impact of aerosol compositions on $PM_{2.5}$ pH is discussed in Section 3.4.

**Figure 4**

Beijing is surrounded by mountains on three sides. Haze episodes usually occur with southwest and southeast winds as well as calm winds in Beijing. The industry is mainly concentrated in the south of Beijing, leading to the higher $PM_{2.5}$ concentration in Beijing by the regional transport and accumulation. Wind dependence of $PM_{2.5}$, $NO_3^-$, $SO_4^{2-}$, $NH_4^+$ and the averaged $PM_{2.5}$ pH are shown in Figure 5 and Figure S9. In spring, summer, and autumn, the $PM_{2.5}$ pH in northern direction were generally higher than that in the southwest direction, but the high pH in summer also occurred with southwest strong winds (wind speed >3 m s$^{-1}$). Generally, the northerly winds usually occur with cold front systems, which could sweep away air pollutants but raised dust in which the crustal ion species ($Ca^{2+}$, $Mg^{2+}$) were higher. In winter, the $PM_{2.5}$ pH distributed relatively evenly in each wind direction, but we surprisingly found that the pH in northerly winds is as low as 3~4, which was

consistent with the high mass fraction of $SO_4^{2-}$ on the clean days caused by the northerly winds.
**Figure 5**
**3.2 Diurnal variation of ALWC, $H_{air}^+$, and $PM_{2.5}$ pH**
The diurnal variations of $NO_3^-$, $SO_4^{2-}$, ALWC, $H_{air}^+$ and $PM_{2.5}$ pH are exhibited in Figure 6. The
diurnal variations for ALWC, $H_{air}^+$, and pH was similar over four seasons. Generally, nighttime
mean ALWC was higher than daytime and reached a peak at near 04:00 ~ 06:00 (local time). After
sunrise, the increasing temperatures resulted in a rapid drop in RH, leading to the obvious loss of
particle water, ALWC reached the lowest level in the afternoon. $H_{air}^+$ was highest in the afternoon
and then followed by nighttime, and $H_{air}^+$ was relatively low in the forenoon. The low ALWC and
high $H_{air}^+$ resulted in the minimum pH in the afternoon. The averaged nighttime pH is 0.3~0.4 unit
higher than that on daytime. Noted that the diurnal variations of $PM_{2.5}$ pH here were for the cases
with RH higher than 30%. If the data at RH$\leq$30% were included, the diurnal variations of $H_{air}^+$, pH,
and $SO_4^{2-}$ in winter were changed (Figure S10). $H_{air}^+$ and $SO_4^{2-}$ were both higher at nighttime since
the nocturnal boundary layer height was generally low in winter and easily resulted in the
accumulation of $SO_4^{2-}$, hence leading to a lower pH at the night.
The diurnal variation of $NO_3^-$ in winter and spring agreed well with the aerosol acidity.
Nevertheless, in summer and autumn, the agreement was not well. Figure S11 shows the relationship
between mass concentrations of $SO_4^{2-}$ and $NO_3^-$ and $PM_{2.5}$ pH at different ALWC levels for all four
seasons. At the relatively low ALWC, the increasing $SO_4^{2-}$ could decrease the pH obviously; at the
relatively high ALWC, the negative correlation still existed between $SO_4^{2-}$ mass concentration and
$PM_{2.5}$ pH. On the contrary, a weak positive correlation was found between $NO_3^-$ and pH at the
relatively low ALWC and the $PM_{2.5}$ pH was almost invariable with the $NO_3^-$ mass concentration at
the relatively high ALWC. Compared with the $NO_3^-$, the $SO_4^{2-}$ had a greater effect on $PM_{2.5}$ pH.
When the ALWC was high enough (for example, higher than 100 μg m$^{-3}$), the impact of dilution of
ALWC to the $H_{air}^+$ was more significant.
**Figure 6**
Guo et al. (2015) found that the ALWC diurnal variation was significant, and the diurnal pattern
in pH was mainly driven by particle water dilution. However, in this work, both $H_{air}^+$ and ALWC
had significant diurnal variations, and the aerosol acidity variation agreed well with sulfate,
indicating the aerosol acidity in NCP was both driven by aerosol composition and particle water.
For example, in the winter of NCP, the $PM_{2.5}$ mass concentration in Beijing was several to dozens
times higher than that in the US, which means there are more seeds in the limited particle water, and
the RH was generally low, hence the dilution of aerosol liquid water to $H_{air}^+$ doesn't work at all, the
diurnal variation of aerosol components was more important.

**3.3 Gas-particle separation**
Table 2 exhibits the measured $\varepsilon(NH_4^+)$, $\varepsilon(NO_3^-)$, and $\varepsilon(Cl^-)$ at different RH levels. The measured
$\varepsilon(NH_4^+)$, $\varepsilon(NO_3^-)$, and $\varepsilon(Cl^-)$ increased with the elevated RH in all four seasons, indicating more
$NH_4^T$, $NO_3^T$, and $Cl^T$ were partitioned into particle phase at higher RH. In winter and spring, $NO_3^T$
and $Cl^T$ were dominated by particle phases, $\varepsilon(NO_3^-)$ and $\varepsilon(Cl^-)$ was higher than 65%. Whereas in
summer and autumn, the lower RH generally companied by higher ambient temperature, more than
half of the $NO_3^T$ and $Cl^T$ were partitioned into the gaseous phase. When the RH reached above 60%,
more than 90% of $NO_3^T$ and 70% of $Cl^T$ were in the particle phase for all four seasons. Compared
with $\varepsilon(NO_3^-)$ and $\varepsilon(Cl^-)$, the $\varepsilon(NH_4^+)$ was pretty lower. In spring, summer, and autumn, the average
$\varepsilon(NH_4^+)$ was still lower than 0.3 even when the RH >60%, which might attribute to the higher $NH_3$
mass concentration in the atmosphere. The averaged $NH_3$ was 21.5±8.7 μg m⁻³, 19.6±6.4 μg m⁻³,
and 16.8±8.0 μg m⁻³ in spring, summer, and autumn, respectively. In winter, the average $\varepsilon(NH_4^+)$
were much higher than that in other seasons with the relatively lower $NH_3$ mass concentration
(4.9±2.8 μg m⁻³).

**Table 2.**

**3.4 Factors affecting ALWC, $H_{air}^+$, $PM_{2.5}$ pH, and gas-particle partitioning**
As mentioned above, the aerosol chemical composition has a non-negligible effect on $PM_{2.5}$ pH.
In this work, the effects of $SO_4^{2-}$, $NO_3^T$, $NH_4^T$, $Cl^T$, $Ca^{2+}$, RH, and T on $PM_{2.5}$ pH were performed
through a sensitivity analysis over four seasons.
As shown in Table 3, for ALWC, the largest relative standard deviation (RSD) was observed
when RH was taken as the evaluated factor, then followed by $SO_4^{2-}$ or $NO_3^-$, which means the RH
had the greatest influence on ALWC, and $SO_4^{2-}$ and $NO_3^-$ were major hygroscopic components in
the aerosol. The $SO_4^{2-}$, RH, $NO_3^T$, and $NH_4^T$ were all important influential factors for $H_{air}^+$,

especially $SO_4^{2-}$. The $SO_4^{2-}$ and T were two crucial factors affecting the $PM_{2.5}$ pH variation. The $PM_{2.5}$ pH was also sensitive to $NH_4^T$ when it was in a lower range and sensitive to RH only in summer. The relationship between pH and $NH_4^T$ was nonlinear, the impact of $NH_4^T$ on pH weakened as $NH_4^T$ increased. In spring, the crucial factor for the $PM_{2.5}$ pH variation was $SO_4^{2-}$ while it was $SO_4^{2-}$ and $NH_4^T$ in winter. In summer, the most important factor affecting $PM_{2.5}$ pH was RH, then followed by $NH_4^T$ and $SO_4^{2-}$. In autumn, the effect of $NH_4^T$ on $PM_{2.5}$ pH was considerable, $SO_4^{2-}$ and T were also important. Figure 7-9 and S12-S17 show how these factors affecting the ALWC, $H_{air}^+$, and aerosol acidity over four seasons. The sensitivity analysis for ALWC and $H_{air}^+$ were similar over four seasons, while the sensitivity of $PM_{2.5}$ pH to RH and $NO_3^T$ in four seasons were different from each other. In this study, winter and summer were chosen for a detailed discussion of sensitivity analysis because more heavy pollution episodes happened in winter while the photochemical reaction was relatively strong in summer.

**Table 3**

**Figure 7**

**Figure 8**

**Figure 9**

**RH:** RH had a different impact on $PM_{2.5}$ pH in different seasons. In winter, the $PM_{2.5}$ pH decreased with the increasing RH, whereas the $PM_{2.5}$ pH increased with the increasing RH in summer. In spring and autumn, the RH between 30~83% had little impact on $PM_{2.5}$ pH. The explanation for this is that the increased RH actually diluted the solution and promoted ionization, releasing $H_{air}^+$ and increasing ALWC as well, but the gradient was different. In winter, variation in $H_{air}^+$ caused by RH changes was much larger than variation in ALWC, whereas it showed an opposite tendency in summer. In autumn and spring, variation in $H_{air}^+$ caused by RH changes was slightly higher than the variation in ALWC. The different impact of RH on $PM_{2.5}$ pH indicated that the dilution effect of ALWC on $H_{air}^+$ was obvious only in summer, the high RH during the severe haze in winter could increase the aerosol acidity.

**T:** At high ambient temperature, $\varepsilon(NH_4^+)$, $\varepsilon(NO_3^-)$, and $\varepsilon(Cl^-)$ all showed a decreased tendency (Figure 10 and S19). The procedure of $NH_4^+ \rightarrow NH_3$ releases one $H^+$ to particle phase, whereas the

procedure of $NO_3^- \rightarrow HNO_3$ or $Cl^- \rightarrow HCl$ both need one $H^+$ from the particle phase. Compared with
the loss of $NO_3^-$ from $NH_4NO_3$ as well as $Cl^-$ from $NH_4Cl$, greater loss of $NH_4^+$ from $NH_4NO_3$,
$NH_4Cl$, and $(NH_4)_2SO_4$ resulted in a net increase in particle $H^+$ and lower pH. In addition, the
molality-based equilibrium constant ($H^*$) of $NH_3$-$NH_4^+$ partitioning decreased faster with
increasing temperature when compared with that of $HNO_3$-$NO_3^-$ partitioning, resulting in a net
increase in particle $H^+$ (Guo et al., 2018). Moreover, higher ambient temperature tends to lower
ALWC, which further decreases the $PM_{2.5}$ pH. The wide range of ambient temperature in autumn
made a significant impact on $PM_{2.5}$ pH in the sensitivity analysis.

**Figure 10**

**$SO_4^{2-}$:** $SO_4^{2-}$ had a key role in aerosol acidity, especially in winter and spring (Figure 9, S14, S17).
In the sensitivity test, the $PM_{2.5}$ pH decreased by about 1.6 (4.1 to 2.5), 4.9 (5.1 to 0.2), 1.0 (3.6 to
2.6), and 0.9 (4.0 to 3.1) unit with $SO_4^{2-}$ concentration went up from 0 to 40 $\mu g\ m^{-3}$ in spring, winter,
summer, and autumn, respectively. In spring and winter, the ALWC was low, the variation of $SO_4^{2-}$
mass concentration could generate dramatic changes in $H_{air}^+$. In section 3.1, the $PM_{2.5}$ pH was lowest
in summer whereas highest in winter, which was consistent with the $SO_4^{2-}$ mass faction in total ions.
The $SO_4^{2-}$ mass faction in total ions in summer was highest among four seasons with 32.4%±11.1%,
whereas it was lowest in winter with 20.9%±4.4%.
**$NO_3^T$:** The impact of $NO_3^-$ on $PM_{2.5}$ pH was also different, which was related to the averages of
input $NH_4^T$ in different seasons. In winter, the $PM_{2.5}$ pH decreased with increasing $NO_3^T$
concentration, whereas little impact was found in summer (Figure 9). In spring and autumn, the
$PM_{2.5}$ pH increases first and then dropped with the increasing $NO_3^T$ concentration (Figure S14, S17).
In winter, the $NH_4^T$ mass concentration was relatively low. As $NO_3^T$ increases, all $NH_3$ could be
converted into $NH_4^+$ ($\varepsilon(NH_4^+) \approx 1$). However, if $HNO_3$ continued to dissolve and released $H_{air}^+$, it
would result in the decrease of $PM_{2.5}$ pH. In summer, the averages of $NO_3^T$ and $Cl^T$ was relatively
low but the $NH_4^T$ was excessive, the highest $\varepsilon(NH_4^+)$ was only 0.6 with the corresponding highest
$NO_3^T$. The excessive $NH_3$ could provide continuous buffering to the increasing $NO_3^T$, together with
a significant dilution of ALWC on $H_{air}^+$, leading to the little changes in $PM_{2.5}$ pH. In spring and
autumn, the increasing pH with elevated $NO_3^T$ in lower range attributed to the dilution of ALWC to
$H_{air}^+$. $H_{air}^+$ concentration increased exponentially with elevated $NO_3^T$ concentration, especially at
higher $NO_3^T$ concentrations, whereas the ALWC increased linearly with elevated $NO_3^T$
concentration (Figure S12-S17), hence ALWC played a dominant role when the $NO_3^T$ concentration
was low. With the further increase of $NO_3^T$, the variation in $H_{air}^+$ caused by $NO_3^T$ addition was larger
than the variation in ALWC, leading to the decrease of $PM_{2.5}$ pH. Besides, the relationship between
$NO_3^T$ and $\varepsilon(NH_4^+)$ in the sensitivity analysis showed that decreasing $NO_3^T$ could lower the $\varepsilon(NH_4^+)$
effectively (Figure 11 and S20), which helped $NH_3$ maintain in the gas phase.

**Figure 11**

**$NH_4^T$:** The relationship between $PM_{2.5}$ pH and $NH_4^T$ was nonlinear. $NH_4^T$ in lower range had a
significant impact on the $PM_{2.5}$ pH (Table S2), and higher $NH_4^T$ generated limited pH change
(Figure 9, S14, S17). Elevated $NH_4^T$ could reduce $H_{air}^+$ exponentially and slightly increase ALWC
when the other input parameters were held constant. As the $NH_4^T$ increased, $H_{air}^+$ was consumed
swiftly during the dissolution of $NH_3$ and the further reaction with $SO_4^{2-}$, $NO_3^-$, and $Cl^-$. The elevated
$NH_4^T$ increased the $\varepsilon(NO_3^-)$ and $\varepsilon(Cl^-)$ when $NO_3^T$ and $Cl^T$ were fixed (Figure 11 and S20), which
means the elevated $NH_4^T$ altered the gas-particle partition and shifted more $NO_3^T$ and $Cl^T$ into
particle phase, leading to the deliquescence of additional nitrate and chloride and an increase of
ALWC. It seems that $NH_3$ emission control is a good way to reduce $NO_3^-$. However, the relationship
between $NH_4^T$ and $\varepsilon(NO_3^-)$ in the sensitivity analysis (Figure 11 and S20) showed that the $\varepsilon(NO_3^-)$
response to $NH_4^T$ control was highly nonlinear, which means the decrease of nitrate would happen
only when the $NH_4^T$ was greatly reduced. The same result was also obtained from a study of Guo et
al (2018).
The ratio of [TA]/2[TS] provides a qualitative description for the ammonia abundance, where
[TA] and [TS] are the total (gas + aqueous + solid) molar concentrations of ammonia and sulfate.
The rich-ammonia is defined as [TA] > 2[TS], while if the [TA] ≤ 2[TS], then it is defined as poor-
ammonia (Seinfeld and Pandis, 2016). In this work, the ratio of [TA]/2[TS] was much higher than
1 and belonged to rich-ammonia (Figure. S21). Although $NH_3$ in the NCP was abundant, the $PM_{2.5}$
pH was far from neutral, which might attribute to the limited ALWC. Compared to the liquid water
content in clouds and precipitation, ALWC was much lower, hence the dilution of aerosol liquid
water to $H_{air}^+$ was weak.
**$Cl^T$:** $Cl^T$ had a relatively larger impact on the $PM_{2.5}$ pH in winter and spring compared to summer
and autumn. Except for winter, the $Cl^T$ mass concentration was generally lower than 10 μg m$^{-3}$,
which accounted for the little impact on $PM_{2.5}$ pH. On account of the low level of $Cl^T$, the dilution
of ALWC on $H_{air}^+$ played a dominant role, generating the $PM_{2.5}$ pH increase with elevated $Cl^T$.
However, similar to $NO_3^T$, higher $Cl^T$ could decrease the $PM_{2.5}$ pH.
**$Ca^{2+}$:** In fine particles, $Ca^{2+}$ mass concentration was generally low. In the output of ISORROPIA-
II, Ca existed as $CaSO_4$ (slightly soluble). Elevated $Ca^{2+}$ concentration could increase the $PM_{2.5}$ pH
by decreasing $H_{air}^+$ and ALWC (Figure S18), the decreased $H_{air}^+$ resulted from the buffering capacity
of $Ca^{2+}$ to the acid species, while the decreased ALWC resulted from the weak water solubility of
$CaSO_4$. As discussed in Section 3.1, on clean conditions, the $PM_{2.5}$ pH could reach 6~7 when the
mass fraction of $Ca^{2+}$ was high, hence the role of mineral ions on $PM_{2.5}$ pH could not be ignored in
seasons (such as spring) or regions where mineral dust was an important source of fine particles.
Due to the strict control measures for road dust, construction sites, and other bare ground, the
nonvolatile cations in $PM_{2.5}$ decreased significantly in NCP.

**3.5 Size distribution of aerosol components and pH**

According to the average $PM_{2.5}$ concentration during every sampling periods, all the samples
were also classified into three groups (clean, polluted, heavily polluted) with the same rule described
in Section 3.1. A severe haze episode occurred during the autumn sampling, hence there were more
heavily polluted samples for autumn than that in other seasons. Figure 12 shows the averaged size
distributions of PM components and pH on clean, polluted, and heavily polluted conditions in
summer, autumn, and winter, respectively. The $NO_3^-$, $SO_4^{2-}$, $NH_4^+$, $Cl^-$, $K^+$, OC, and EC mainly
concentrated in the size range with aerodynamic diameters between 0.32~3.1μm, while $Mg^{2+}$ and
$Ca^{2+}$ predominantly distributed in the coarse mode. As shown in Figure 12, the concentration levels
for all chemical components increased with the increasing pollution. During the haze episodes, the
sulfate and nitrate in the accumulated mode increased significantly. However, the increase of $Mg^{2+}$
and $Ca^{2+}$ in the coarse mode were not as obvious as secondary ions, mainly due to the low wind
speed and calm atmosphere which made it more difficult to raise dust during the heavy pollution.
More detailed information about size distributions of mass concentration for all analyzed species
during three seasons is shown in Zhao et al. (2017) and Su et al. (2018). As mentioned in section
2.4, assuming coarse mode particles in equilibrium with the gas phase could overpredict $NO_3^-$ and
$Cl^-$ and underestimate $NH_4^+$ in the coarse mode (Figure 3), which subsequently underestimated the
coarse mode aerosol pH. Thus, the gas phase was ignored for pH calculation of the coarse particles
(>3.1μm).
**Figure 12**
The aerosol pH for both fine mode and coarse mode in summer was lowest among three seasons,
then followed by autumn and winter. The seasonal variation of aerosol pH derived from MOUDI
data was consistent with that derived from real-time $PM_{2.5}$ chemical components measurement. In
summer, the predominance of sulfate in the fine mode and high ambient temperature resulted in a
low pH, ranging between 1.8 and 3.9. Aerosol pH for fine particles in autumn and winter was in the
range of 2.4 ~ 6.3 and 3.5 ~ 6.5, respectively. The difference of aerosol pH between size bins in fine
mode was not significant, probably owing to the excessive $NH_3$ (Guo et al., 2017).
As for coarse particles, the predicted pH was approximately near or even higher than 7 for all of
the three seasons in this work, which mainly attributed to the buffering capacity of the coarse mode
mineral dust. Simulations with extreme cases that $Ca^{2+}$ and $Mg^{2+}$ were removed from the input files
were conducted. The results showed that the presence of $Ca^{2+}$ and $Mg^{2+}$ had a crucial effect on
coarse mode aerosol pH (Figure S22), the difference of aerosol pH (with and without $Ca^{2+}$ and $Mg^{2+}$)
for particles larger than 1 μm increased with the increasing particle size. For particles smaller than
1 μm, the removal of $Ca^{2+}$ and $Mg^{2+}$ had little effect on aerosol pH.
The aerosol pH in coarse mode decreased significantly when under the heavily polluted condition,
especially in autumn and winter. For example, the pH in stage 3 (3.1-6.2 μm) declined from 7.8
under the clean condition to 4.5 under the heavily polluted condition in winter, implying that the
aerosols in coarse mode during severe hazy days would become weak acid from neutral. The
obvious increase of nitrate in coarse mode might responsible for this. Moreover, the significant
decrease of mass ratios of $Ca^{2+}$ and $Mg^{2+}$ resulted in the loss of coarse mode buffering capacity.
The size distributions of aerosol pH and all analyzed chemical components in the daytime and
nighttime are illustrated in Figure S23. For summer and autumn, the pH in the nighttime was higher
than that in the daytime. The diurnal variation for aerosol pH based on MOUDI data was consistent
with the online data. Whereas in winter, the pH was higher in the daytime. In winter, the averaged
RH during the sampling period was relatively low, leading to a low ALWC, but the $SO_4^{2-}$ and $NO_3^-$
in the nighttime were obviously higher due to the lower boundary layer height. Therefore, $H_{air}^+$ was
more abundant in nighttime while the low ALWC had little effect on pH.
**5. Summary and conclusions**
On the basis of online measurements, the measured and predicted $NH_3$, $NH_4^+$, $NO_3^-$, $Cl^-$, and
$\varepsilon(NH_4^+)$ by using ISORROPIA-II agreed pretty well when RH was higher than 30%. It is not
reasonable to assume aerosol in a liquid state (metastable) and the aerosol pH could not be accurately
predicted by a thermodynamic model where the RH is relatively low. Thus, we only discussed the
$PM_{2.5}$ pH for data with RH higher than 30% in this work.
In Beijing, the mean $PM_{2.5}$ pH over four seasons (RH≥30%) was 4.0±1.0 (spring),
4.5±0.7(winter), 3.8±1.2(summer), 4.3±0.8 (autumn), respectively, showing the moderate acidity.
In this work, both $H_{air}^+$ and ALWC had significant diurnal variation, and the $PM_{2.5}$ acidity variation
agreed well with sulfate, indicating the aerosol acidity in NCP was both driven by aerosol
composition and particle water. The averaged nighttime pH is 0.3~0.4 unit higher than that on
daytime. The $PM_{2.5}$ pH in the northerly direction was higher than that in the southwest direction.
A sensitivity analysis was performed in this work to investigate how $SO_4^{2-}$, $NO_3^T$, $NH_4^T$, $Cl^T$,
$Ca^{2+}$, RH, and T affect ALWC, $H_{air}^+$, and $PM_{2.5}$ acidity. The RH affects ALWC most, then followed
by $SO_4^{2-}$ or $NO_3^-$. The $SO_4^{2-}$, RH, $NO_3^T$, and $NH_4^T$, especially $SO_4^{2-}$, were all important influential
factors for $H_{air}^+$. As for $PM_{2.5}$ pH, $SO_4^{2-}$, T, $NH_4^T$, and RH (only in summer) were crucial factors.
In winter, $PM_{2.5}$ pH decreased slightly with the increasing RH, whereas the $PM_{2.5}$ pH increased
with the increasing RH in summer. The dilution effect of ALWC on $H_{air}^+$ was obvious only in
summer. In spring and autumn, the RH had little impact on $PM_{2.5}$ pH due to the comparable
variations of $H_{air}^+$ and ALWC. The measured $\varepsilon(NH_4^+)$, $\varepsilon(NO_3^-)$, and $\varepsilon(Cl^-)$ increased with the
elevated RH in all four seasons. In addition, the higher ambient temperature tended to lower $PM_{2.5}$
pH due to the volatilization of $NH_4^+$, $NO_3^-$, $Cl^-$ and the decrease of ALWC.
$SO_4^{2-}$ had a key role for aerosol acidity, especially in winter and spring. In spring and winter, the
ALWC was relatively low, the variation of $SO_4^{2-}$ concentration could generate dramatic changes in
$H_{air}^+$. The impact of $NO_3^-$ on $PM_{2.5}$ pH was different in four seasons. In winter, the $PM_{2.5}$ pH
decreased with increasing $NO_3^-$ concentration due to the low $NH_4^T$ mass concentration. In summer,
the excessive $NH_3$ could provide continuous buffering to the increasing $NO_3^T$ and lead to little
change in $PM_{2.5}$ pH.

The relationship between pH and $NH_4^T$ was nonlinear, the impact of $NH_4^T$ on $PM_{2.5}$ pH gradually

weakened as $NH_4^T$ increased. Elevated $NH_4^T$ consumed $H_{air}^+$ swiftly and shifted more $NO_3^T$ and $Cl^T$
into particle phase. In NCP, $NH_3$ was much rich in spring, summer, and autumn, while less rich in
winter. Although $NH_3$ in the NCP was abundant, the $PM_{2.5}$ pH was far from neutral, which mainly
attributed to the limited ALWC.

$Cl^T$ and $Ca^{2+}$ had little impact on the $PM_{2.5}$ pH due to the low mass concentration. Elevated $Ca^{2+}$

concentration could increase the $PM_{2.5}$ pH because of the buffering capacity of $Ca^{2+}$ to the acid
species and the weak water solubility of $CaSO_4$.

The sensitivity analysis of the relationship between $NO_3^T$ and $\varepsilon(NH_4^+)$ imply that decreasing

$NO_3^T$ could reduce the $\varepsilon(NH_4^+)$ effectively, which helped keep $NH_3$ in the gas phase. In contrast,
the nitrate response to $NH_4^T$ control was highly nonlinear, the decrease of nitrate would happen only
when the $NH_4^T$ was greatly reduced.

The size-resolved results showed that the pH of coarse particles was approximately near or even

higher than 7 for all three seasons, which was quite higher than that of fine particles. The difference
of aerosol pH between size bins in fine mode was not significant. The aerosol pH in coarse mode
decreased significantly, becoming weak acid from neutral, when under heavily polluted condition.
For summer and autumn, the pH in the nighttime was higher than that in the daytime. Whereas in
winter, the pH was higher in the daytime.

*Data availability.* All data in this work are available by contacting the corresponding author P. S.
Zhao (pszhao@ium.cn).

*Author contributions.* P Z designed and led this study. J D and P Z interpreted the data and discussed
the results. J S and X D analyzed the chemical compositions from size-resolved aerosol samples. J
D and P Z wrote the manuscript.

*Competing interests.* The authors declare that they have no conflict of interest.

*Acknowledgments.* This work was supported by the National Natural Science Foundation of China
(41675131), the Beijing Talents Fund (2014000021223ZK49), and the Beijing Natural Science
Foundation (8131003). Special thanks to the Max Planck Institute for Chemistry and Leibniz
Institute for Tropospheric Research where Dr. Zhao visited as a guest scientist in 2018.

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

## Table captions

**Table 1.** Average mass concentrations of $NO_3^-$, $SO_4^{2-}$, $NH_4^+$ and $PM_{2.5}$ as well as RH, ALWC, $H_{air}^+$, and $PM_{2.5}$ pH under clean, polluted, and heavily polluted conditions over four seasons.

**Table 2.** Average $\varepsilon(NH_4^+)$, $\varepsilon(NO_3^-)$, $\varepsilon(Cl^-)$, and ambient temperature at different ambient RH levels in four seasons.

**Table 3.** Sensitivity of ALWC, $H_{air}^+$, and $PM_{2.5}$ pH to $SO_4^{2-}$, $NH_4^T$, $NO_3^T$, $Cl^T$, $Ca^{2+}$, RH, and T. The larger magnitude of the relative standard deviation (RSD) represents the larger impact derived from the variation of variables.

.


**Table 1**

| Spring | PM$_{2.5}$ | NO$_3^-$ | SO$_4^{2-}$ | NH$_4^+$ | ALWC* | H$_{air}^{+*}$ | pH* |
|---|---|---|---|---|---|---|---|
| | μg m$^{-3}$ | μg m$^{-3}$ | μg m$^{-3}$ | μg m$^{-3}$ | μg m$^{-3}$ | μg m$^{-3}$ | |
| Average | 62±36 | 14.9±14.6 | 9.7±7.9 | 7.9±7.3 | 23±35 | 6.8E-06±2.8E-05 | 4.0±1.0 |
| Clean | 44±17 | 7.9±6.6 | 6.2±3.7 | 4.8±3.2 | 14±26 | 3.2E-06±5.1E-06 | 4.1±1.1 |
| Polluted | 100±21 | 30.8±14.3 | 16.4±5.9 | 15.4±5.8 | 33±36 | 5.1E-06±4.3E-06 | 3.9±0.5 |
| Heavily polluted | 169±12 | 45.3±8.5 | 36.3±4.9 | 29.4±2.3 | 78±60 | 2.0E-05±6.5E-06 | 3.6±0.3 |
| Winter | PM$_{2.5}$ | NO$_3^-$ | SO$_4^{2-}$ | NH$_4^+$ | ALWC* | H$_{air}^{+*}$ | pH* |
| Average | 60±69 | 13.7±21.0 | 7.3±8.7 | 7.3±10.0 | 35±46 | 2.2E-05±2.3E-04 | 4.5±0.7 |
| Clean | 22±20 | 3.6±3.9 | 2.8±1.8 | 2.2±2.0 | 10±16 | 3.2E-07±4.8E-07 | 4.5±0.6 |
| Polluted | 107±21 | 18.9±8.6 | 11.0±5.7 | 11.0±4.7 | 41±45 | 1.9E-05±9.1E-05 | 4.8±1.0 |
| Heavily polluted | 209±39 | 59.7±21.8 | 26.2±6.3 | 29.1±8.7 | 80±52 | 7.0E-05±4.7E-04 | 4.4±0.7 |
| Summer | PM$_{2.5}$ | NO$_3^-$ | SO$_4^{2-}$ | NH$_4^+$ | ALWC* | H$_{air}^{+*}$ | pH* |
| Average | 39±24 | 9.5±9.5 | 8.6±7.5 | 7.2±5.6 | 50±68 | 1.6E-05±1.8E-05 | 3.8±1.2 |
| Clean | 33±18 | 7.3±6.8 | 7.0±6.0 | 5.9±4.0 | 42±61 | 1.4E-05±1.6E-05 | 3.8±1.2 |
| Polluted | 87±13 | 26.5±10.5 | 20.7±7.0 | 17.6±4.8 | 100±88 | 3.1E-05±2.0E-05 | 3.5±0.4 |
| Autumn | PM$_{2.5}$ | NO$_3^-$ | SO$_4^{2-}$ | NH$_4^+$ | ALWC* | H$_{air}^{+*}$ | pH* |
| Average | 59±48 | 18.5±19.5 | 6.5±5.9 | 8.2±8.2 | 109±160 | 8.1E-06±1.1E-05 | 4.3±0.8 |
| Clean | 33±21 | 7.6±7.4 | 4.4±4.1 | 3.8±3.5 | 49±83 | 3.8E-06±6.6E-06 | 4.5±1.0 |
| Polluted | 105±21 | 33.8±11.6 | 14.3±6.3 | 16.0±4.6 | 225±189 | 1.7E-05±1.2E-05 | 4.1±0.3 |
| Heavily polluted | 174±18 | 63.4±15.4 | 25.0±15.9 | 29.0±5.1 | 317±236 | 2.2E-05±1.0E-05 | 4.1±0.2 |

* For data with RH>30%.






 **Table 2**

|  | RH | T, °C | $\varepsilon(NH_4^+)$ | $\varepsilon(NO_3^-)$ | $\varepsilon(Cl^-)$ |
|---|---|---|---|---|---|
| Spring | ≤ 30 % | 24.8 ± 3.7 | 0.17±0.14 | 0.84±0.12 | 0.67±0.24 |
|  | 30~60 % | 20.6 ± 3.8 | 0.25±0.14 | 0.91±0.06 | 0.82±0.16 |
|  | >60 % | 15.8 ± 2.7 | 0.28±0.12 | 0.96±0.03 | 0.96±0.06 |
| Winter | ≤ 30 % | 5.4 ± 5.3 | 0.31±0.13 | 0.78±0.12 | 0.89±0.14 |
|  | 30~60 % | 1.0 ± 3.6 | 0.50±0.21 | 0.89±0.10 | 0.97±0.03 |
|  | >60 % | -1.9 ± 2.1 | 0.60±0.20 | 0.96±0.03 | 0.99±0.01 |
| Summer | ≤ 30 % | 35.6± 0.4 | 0.06±0.02 | 0.35±0.20 | 0.39±0.17 |
|  | 30~60 % | 29.6 ± 4.2 | 0.17±0.11 | 0.65±0.23 | 0.43±0.16 |
|  | >60 % | 25.2 ± 3.8 | 0.26±0.12 | 0.90±0.12 | 0.71±0.15 |
| Autumn | ≤ 30 % | 21.7± 7.5 | 0.07±0.06 | 0.49±0.25 | 0.45±0.21 |
|  | 30~60 % | 20.8± 6.3 | 0.21±0.14 | 0.82±0.19 | 0.67±0.21 |
|  | >60 % | 14.9 ± 5.7 | 0.30±0.19 | 0.92±0.10 | 0.86±0.13 |

**Table 3**

| Impact Factor | | $SO_4^{2-}$ | $NO_3^T$ | $NH_4^T$ | $Cl^T$ | $Ca^{2+}$ | RH | T |
|---|---|---|---|---|---|---|---|---|
| Spring | RSD-ALWC | 50.5% | 53.4% | 2.9% | 7.5% | 21.2% | 122% | 13.1% |
| | RSD-$H_{air}^+$ | 223% | 34.4% | 26.8% | 12.4% | 49.8% | 115% | 49.5% |
| | RSD-pH | **12.4%** | 5.2% | 3.9% | 2.4% | 5.5% | 1.3% | 7.0% |
| Winter | RSD-ALWC | 33.8% | 28.7% | 14.2% | 30.7% | 1.9% | 103% | 3.5% |
| | RSD-$H_{air}^+$ | 431% | 431% | 187.4% | 52.3% | 11.3% | 136% | 74.1% |
| | RSD-pH | **28.1%** | 8.4% | **27.0%** | 3.8% | 1.0% | 4.1% | 6.7% |
| Summer | RSD-ALWC | 49.4% | 46.0% | 6.9% | 3.6% | 9.0% | 104% | 10.8% |
| | RSD-$H_{air}^+$ | 131% | 29.9% | 78.1% | 3.4% | 18.1% | 44.6% | 33.9% |
| | RSD-pH | **7.9%** | 3.6% | **8.1%** | 0.8% | 1.9% | **8.6%** | 5.8% |
| Autumn | RSD-ALWC | 32.8% | 58.1% | 9.9% | 6.9% | 3.3% | 77.6% | 5.5% |
| | RSD-$H_{air}^+$ | 171% | 126.7% | 333.1% | 2.0% | 9.3% | 106% | 59.6% |
| | RSD-pH | **6.0%** | 3.3% | **16.1%** | 1.0% | 0.8% | 2.4% | **7.5%** |

## Figure captions

**Figure 1.** Time series of relative humidity (RH), temperature (T) (a, e, i, m); $PM_{2.5}$, $PM_{10}$, and $NH_3$ (b, f, g, n); dominant water-soluble ion species: $NO_3^-$, $SO_4^{2-}$, and $NH_4^+$ (c, g, k, o); and $PM_{2.5}$ pH colored by $PM_{2.5}$ concentration (d, h, l, p) over four seasons.

**Figure 2.** Comparisons of predicted and measured $NH_3$, $HNO_3$, $HCl$, $NH_4^+$, $NO_3^-$, $Cl^-$, $\varepsilon(NH_4^+)$, $\varepsilon(NO_3^-)$, and $\varepsilon(Cl^-)$ colored by RH. In this Figure, the data of four seasons were put together, and the comparisons for each season were shown in Figure S1-S4.

**Figure 3.** Comparisons of predicted and iterative $NH_3$, $HNO_3$, and $HCl$, as well as the predicted and measured $NH_4^+$, $NO_3^-$, $Cl^-$, $\varepsilon(NH_4^+)$, $\varepsilon(NO_3^-)$, and $\varepsilon(Cl^-)$ colored by particle size. In this Figure, all MOUDI data were put together.

**Figure 4.** Time series of mass fraction of $NO_3^-$, $SO_4^{2-}$, $NH_4^+$, $Cl^-$, and crustal ions ($Mg^{2+}$, $Ca^{2+}$) in total ions as well as $PM_{2.5}$ pH in all four seasons.

**Figure 5.** Wind dependence map of $PM_{2.5}$ pH over four seasons. In each picture, the shaded contour indicates the average of variables for varying wind speeds (radial direction) and wind directions (transverse direction).

**Figure 6.** Diurnal patterns of mass concentrations of $NO_3^-$ and $SO_4^{2-}$ in $PM_{2.5}$, predicted aerosol liquid water content (ALWC), $H_{air}^+$, and $PM_{2.5}$ pH over four seasons. Mean and median values are shown, together with 25% and 75 % quantiles. Data with RH≤30% were excluded, the shadow represents the time period when the RH lower than 30% mostly occurred.

**Figure 7.** Sensitivities of $H_{air}^+$ to $SO_4^{2-}$, $NO_3^T$, $NH_4^T$, and $Cl^T$, as well as meteorological parameters (RH, T) in summer and winter.

**Figure 8.** Sensitivities of ALWC to $SO_4^{2-}$, $NO_3^T$, $NH_4^T$, and $Cl^T$, as well as meteorological parameters (RH, T) in summer and winter.

**Figure 9.** Sensitivities of $PM_{2.5}$ pH to $SO_4^{2-}$, $NO_3^T$, $NH_4^T$, and $Cl^T$, as well as meteorological parameters (RH, T) in summer and winter.

**Figure 10.** Sensitivities of $\varepsilon(NH_4^+)$, $\varepsilon(NO_3^-)$, and $\varepsilon(Cl^-)$ to $NO_3^T$, $NH_4^T$, and $Cl^T$ colored by $PM_{2.5}$ pH in summer and winter.

**Figure 11.** Sensitivities of $\varepsilon(NH_4^+)$, $\varepsilon(NO_3^-)$, and $\varepsilon(Cl^-)$ to RH and T colored by $PM_{2.5}$ pH in summer and winter.

**Figure 12.** The size distributions of aerosol pH and all analyzed chemical components under clean
(a, d, g), polluted (b, e, h), and heavily polluted conditions (c, f, i) in summer, autumn, and winter.

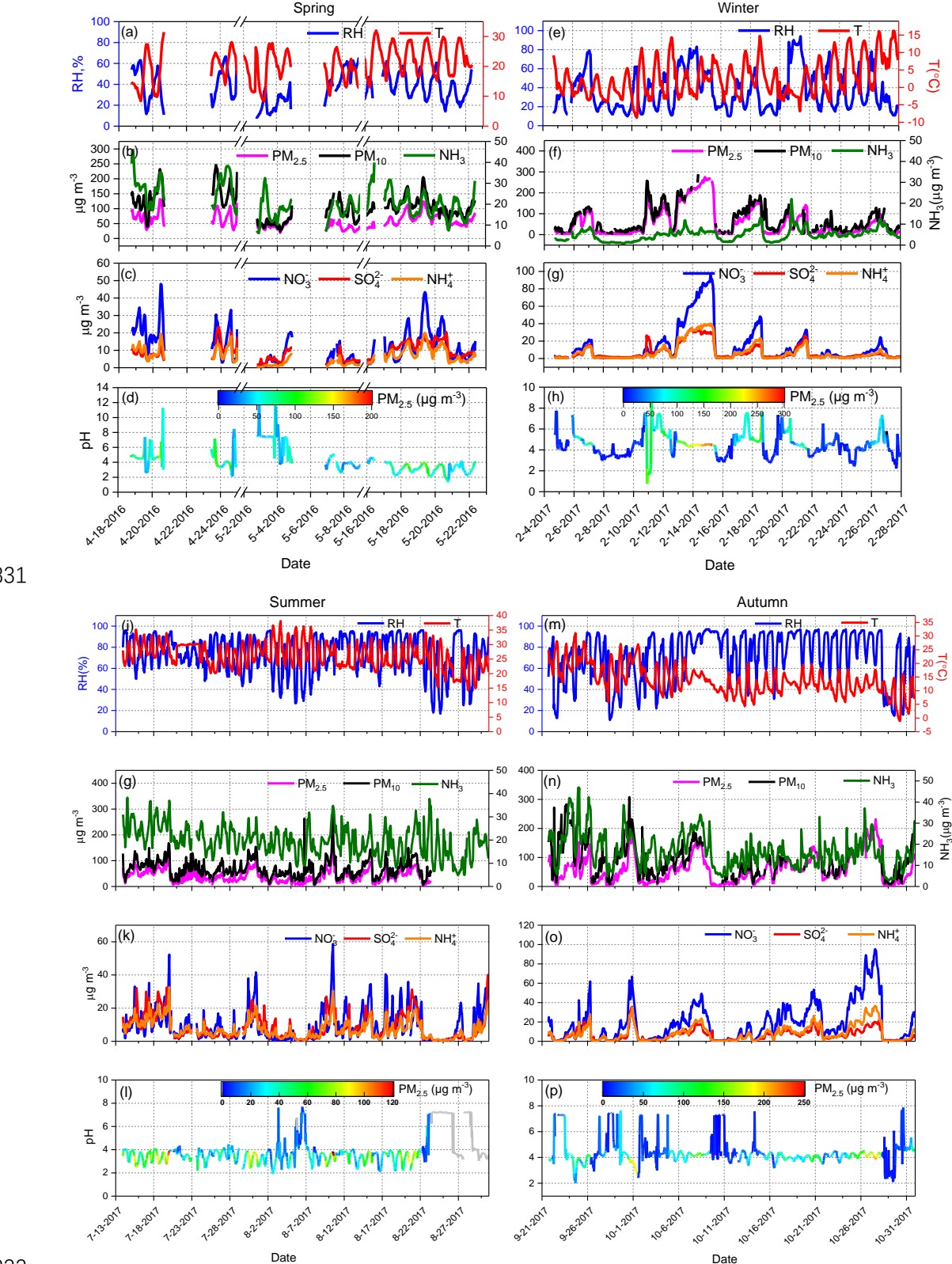



**Figure 1.**





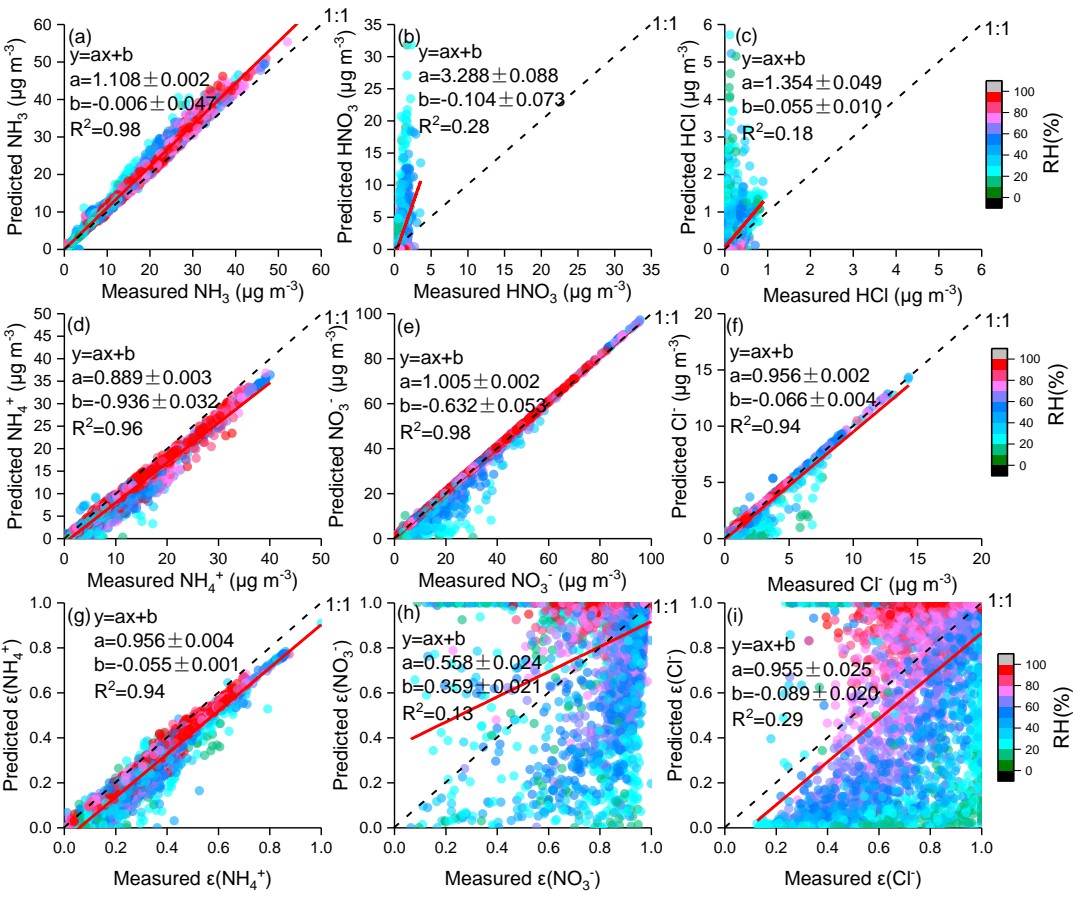


**Figure 2.**

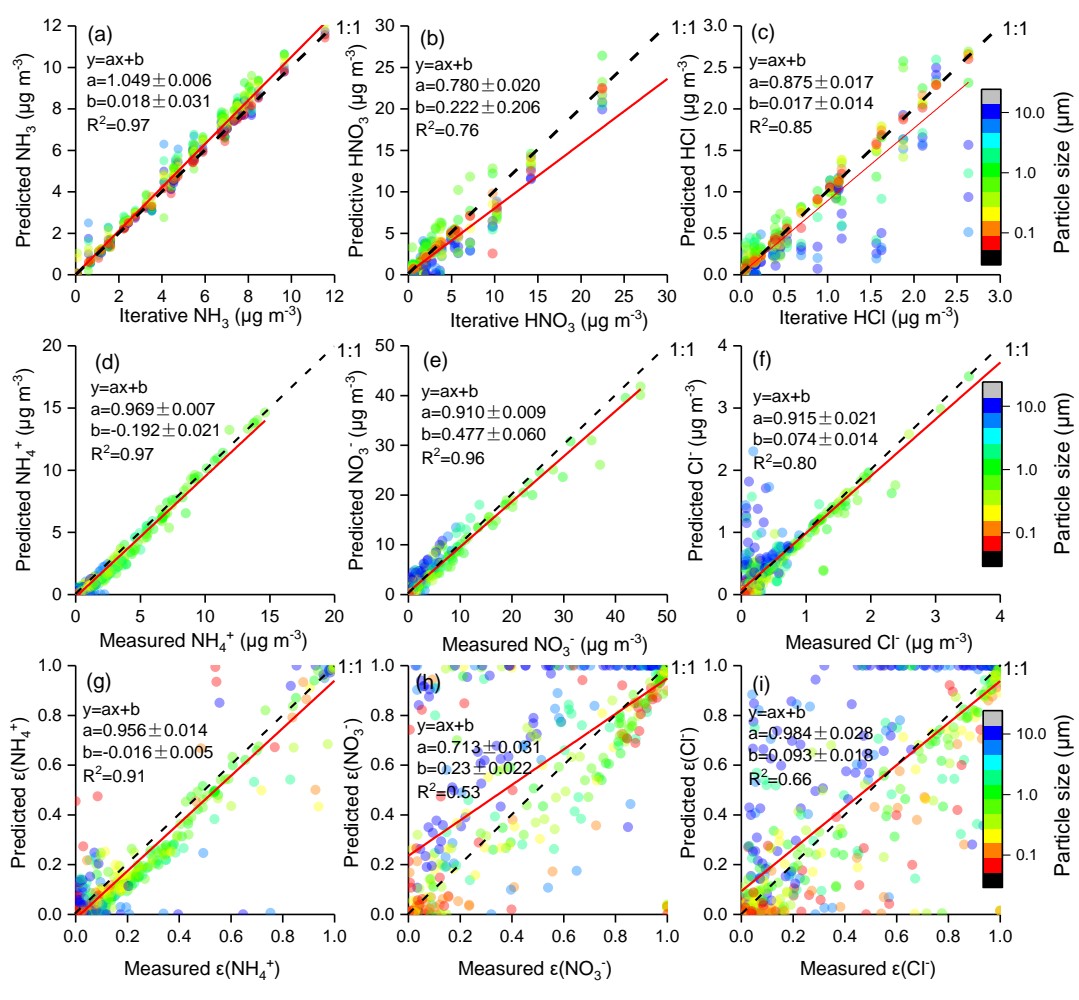



**Figure 3.**

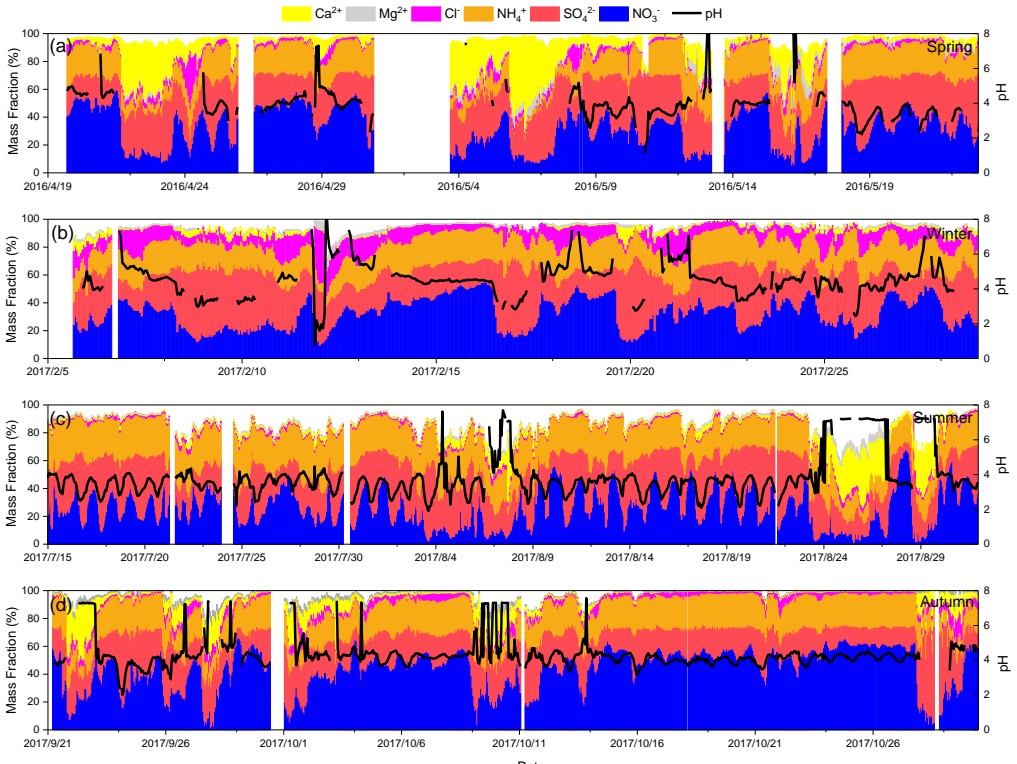

**Figure 4.**






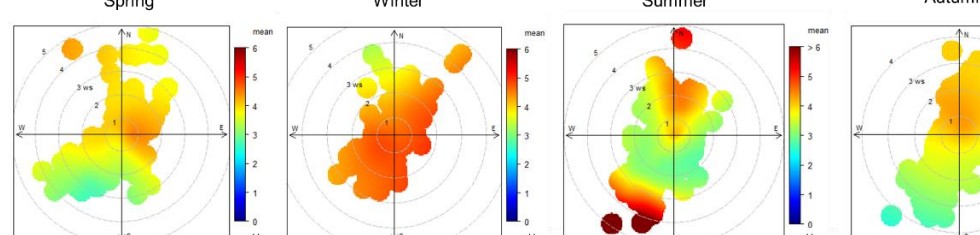

**Figure 5.**


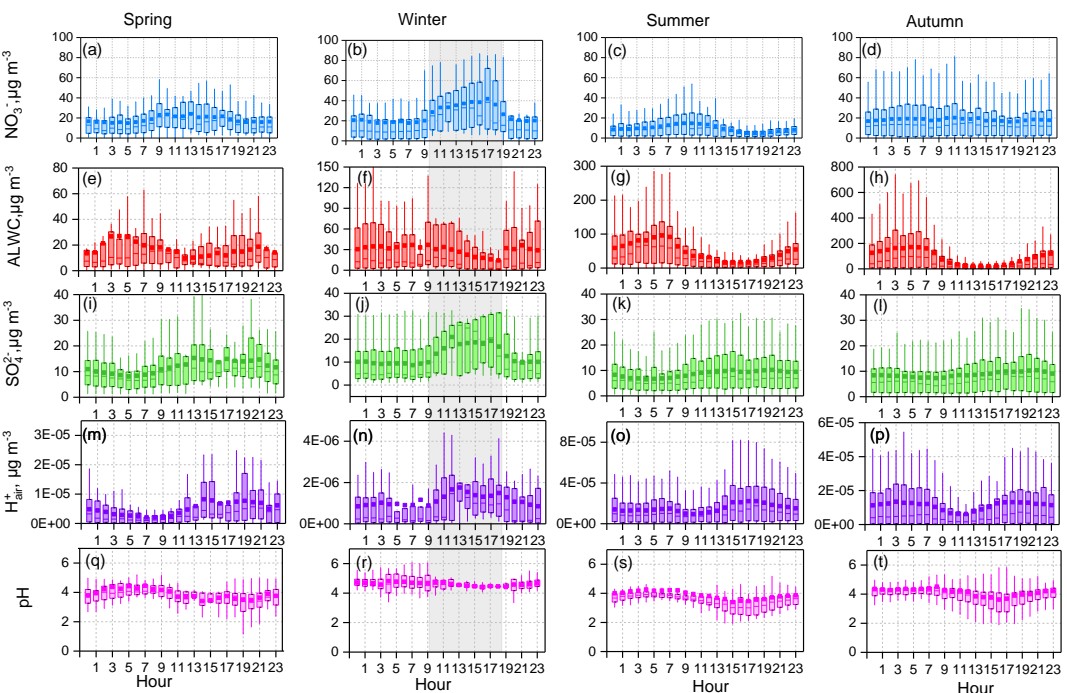


**Figure 6.**


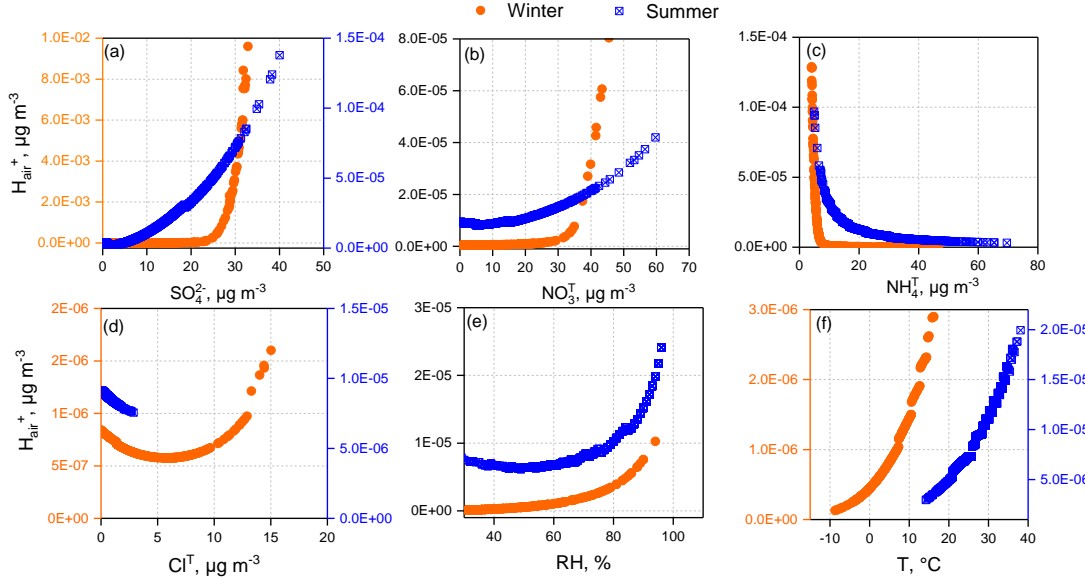


**Figure 7.**


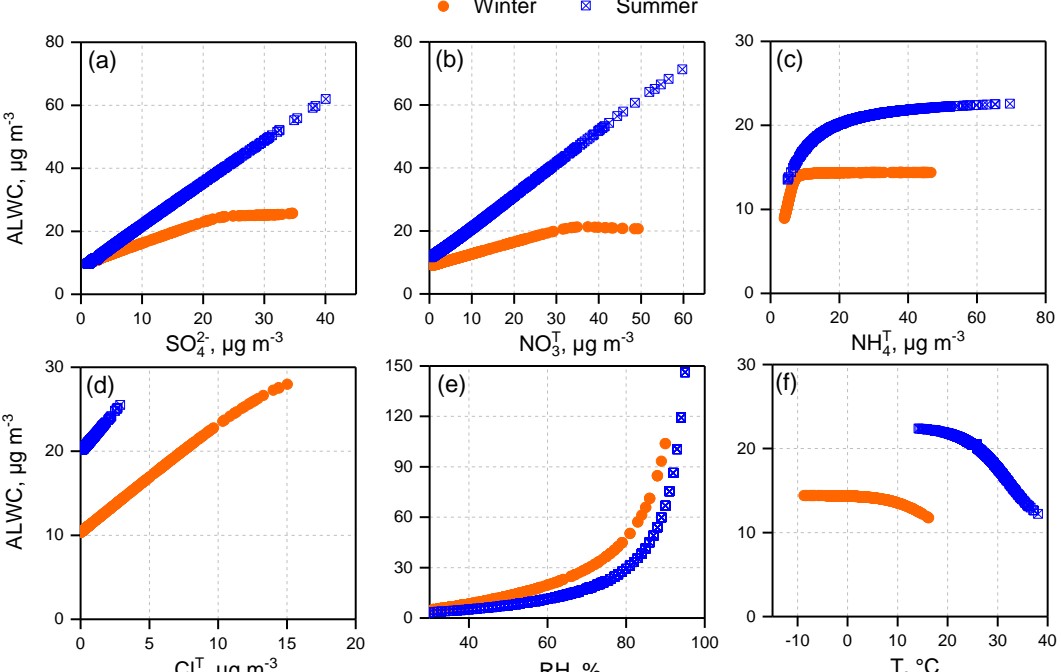


**Figure 8.**


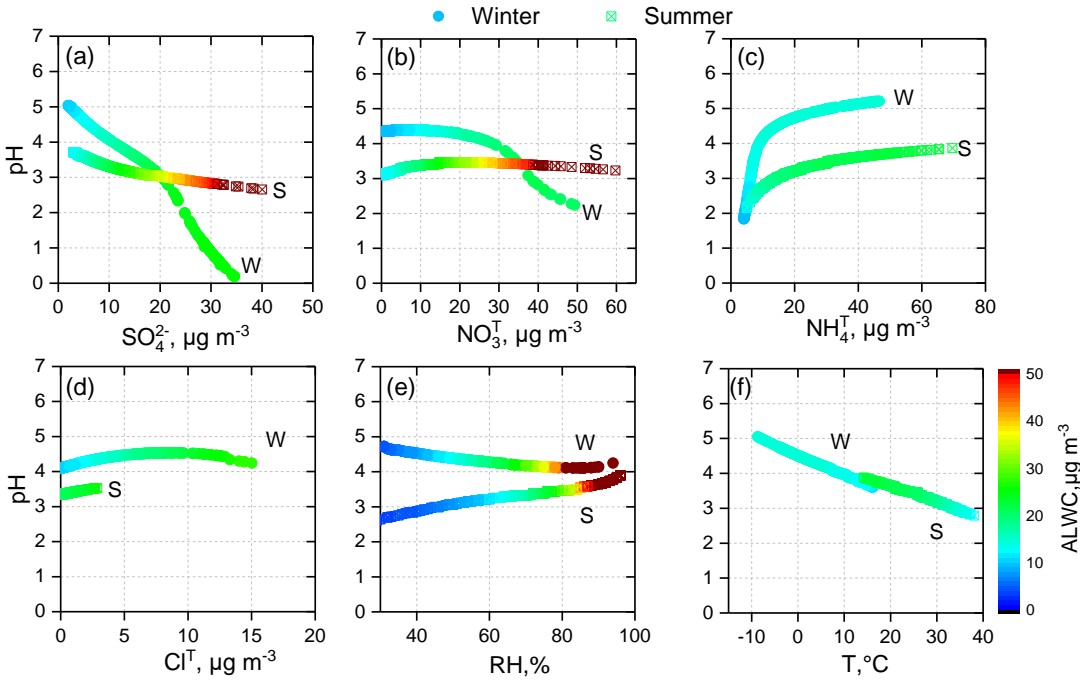


**Figure 9.**

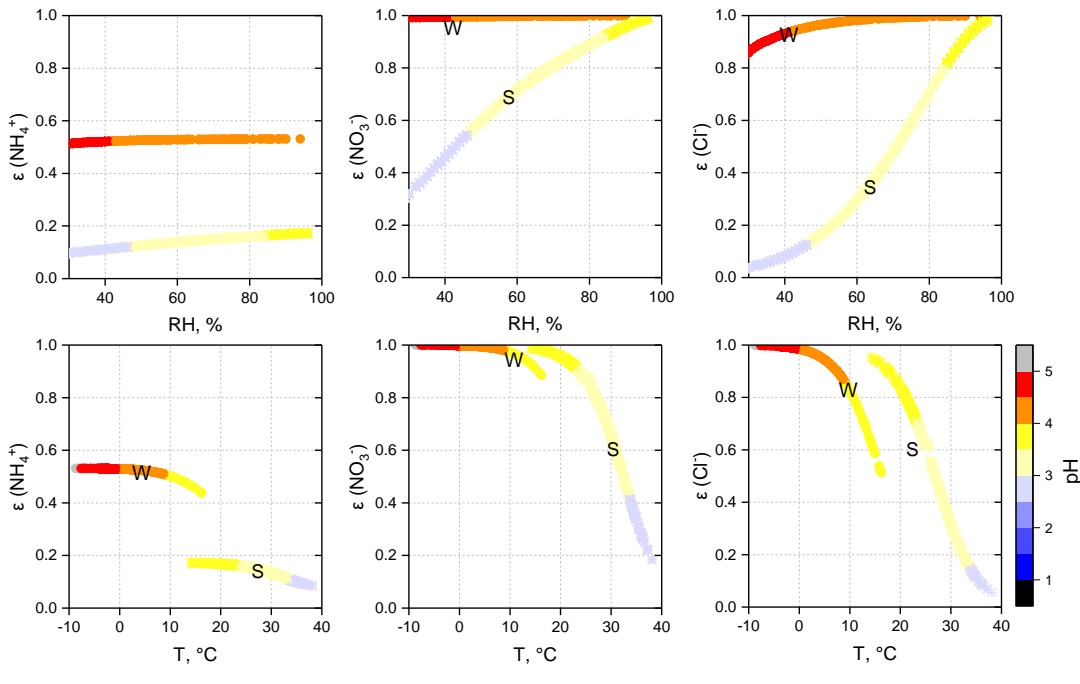


Figure 10.

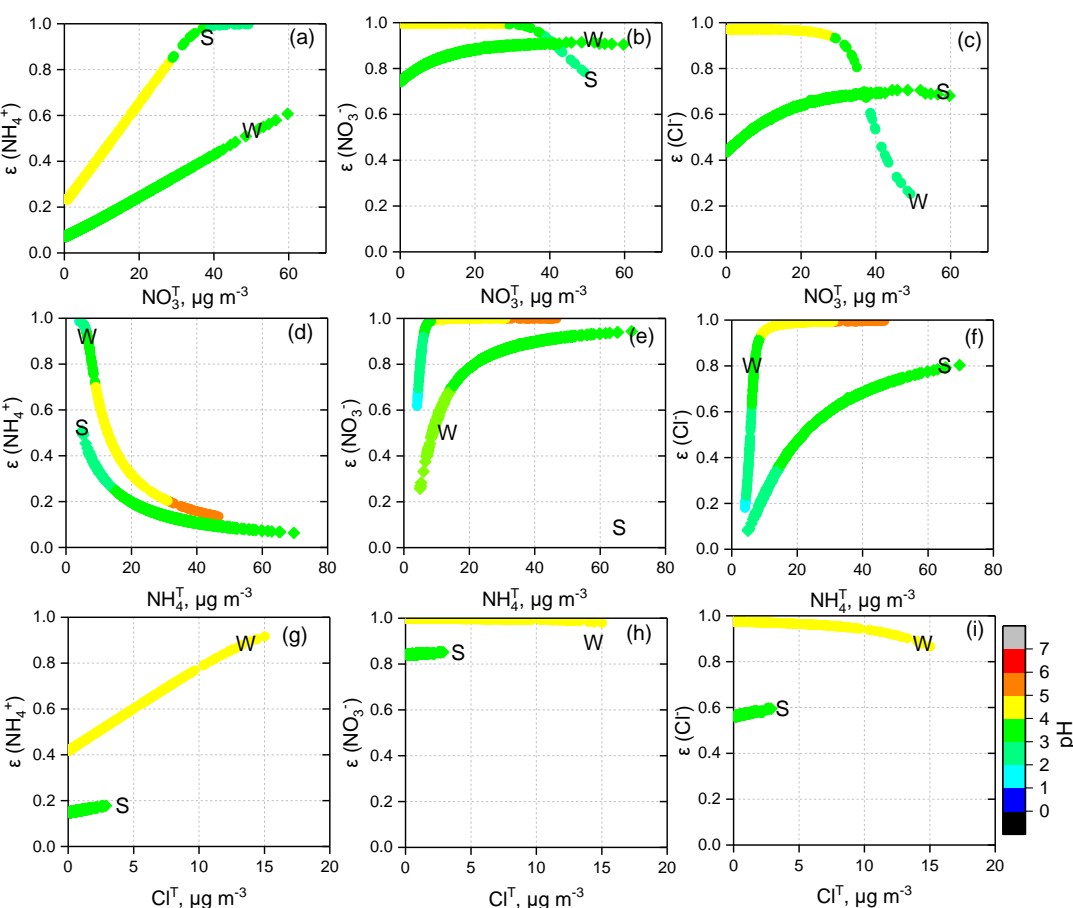


Figure 11.

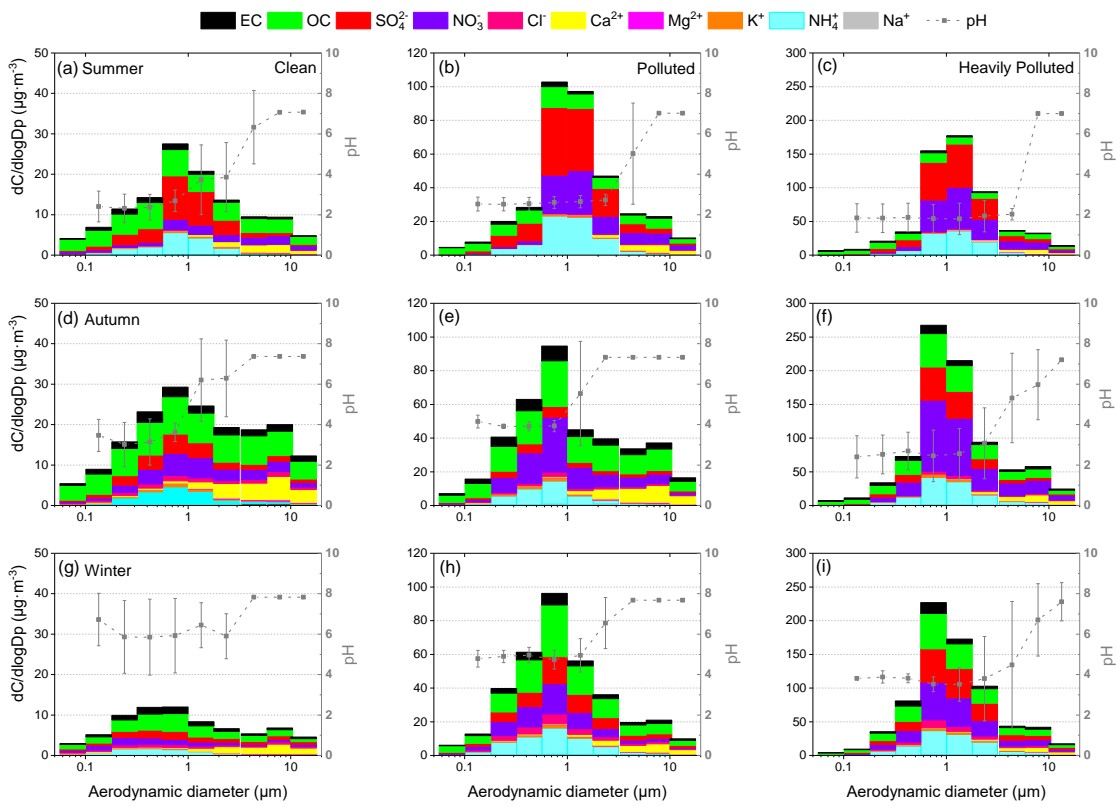

**Figure 12.**