# Peer review of "Aerosol pH and its driving factors in Beijing"

_Atmospheric Chemistry and Physics, 2018_

## Referee Comment (RC1) · Anonymous Referee #1 · 13 Jun 2018

This paper utilizes unique data sets to predict aerosol pH in the more polluted regions of China. Overall, the paper is a significant contribution since little is known about aerosol pH in these regions and even less on size resolved pH. However, in my view, the analysis is somewhat limited. The authors have an interesting data set that could be more fully utilized to assess the pH predictions, partitioning of inorganic species and understand aerosol pH from a more fundamental standpoint.

A suite of important inorganic gases was measured with the MARGA, but they are not significantly discussed in the paper. This is a major oversight. For example, in the comparison of the model to measurements the particle data are shown, but no gas data. For the MOUDI, no gas data is available so the pH is estimated by an iteration method, why not use the MARGA data, which includes gases, to test the sensitivity of pH to this approach?

[Figure]

Greater utilization of the gas data could also help the authors understand fundamentally what is driving pH and the sensitivities to various parameters. This could include the use of S curves, as done extensively by Guo et al, to go beyond just simple variation of one variable at a time. Eg, why in the sensitivity analysis do changes in HNO3 not affect pH, but changes in NH3 do? These, and possibly other, more detailed analysis would reduce the sense that the authors simply run the thermodynamic model and plotted results.

pH is calculated under the assumption of a completely deliquesced particle with no phase separation, all the way down to very low RH, ie, to 30%. These assumptions at low RH need to be justified. Eg, the predicted and measured partitioning of NH3/NH4+, HNO3/NO3-, HCl/Cl- etc (ie include analysis of the gases) could be assessed as a function of pH and see if changes occur at lower RH. Discussion of phase separation in the literature under various conditions (RH, T, O/C) etc should be discussed.

Finally, the paper needs editing for proper English usage and grammar.

Specific Comments Line 37 change specials to species

Line 202 and following, it is not just lack of NH3 data that can affect predicted pH, what about HNO3, HCl, etc?

Line 205, how much did the pH change when the iteration approach is used? Or, were the predicted gas species concentrations reasonable relative to what was measured during the MARGA study period.

Line 229 is superfluous, it is well known that low pH means high acidity.

Line 235 to 238: this paragraph seems out of place.

Fig 3 caption, what does transverse direction mean on a polar plot?

Line 246 change souther to southern.

Line 276-286. From Fig 4 it does not appear that pH and sulfate diurnal trends are

always the same (actually inverse), as stated. Looks like a stronger inverse trend with liquid water. More quantitative analysis is needed to support the statements made in this section.

Line 327, provide a physical explanation for the U shape dependency of H+ on NO3-

Line 330-331: Is it really true that there is a straightforward relationship between NH3 and H+ over broad NH3 concentration ranges? Ie, will increases in NH3 always lead to higher dissolved NH3? Technically it may be true, but the relationship may be highly nonlinear under certain conditions. This statement seems too broad.

Line 335, this is an obvious statement based on Eq (1). In fact much of the discussion throughout relating pH, H+ and LWC is obvious from Eq (1).

Line 358 and on regarding changes in pH with NO3-. The authors discuss the trends they observe in the sensitivity analysis and NO3-/SO4=, but never provide an explanation. By just reporting of results, the value of this work is greatly limited, despite the what could be done with this unique data set.

Lines 380 and on regarding TA and TS. Most of these statements are technically incorrect (although, from a broad perspective they may have a grain of truth to them). The authors data show that the pH is far from neutral despite it being NH3 rich. This analysis largely continues misconceptions of how aerosol composition depends on interactions between SO4=, NH3, NH4+, HNO3, NO3- and LWC. Eg, is HNO3 only taken up once sulfate is so-called neutralized; maybe this can be tested with the data (there should be no NO3- and then a sudden jump in NO3- when [TA]/2[TS] is greater than 1. Another example, why does pH vary, even for this data set, if NH3 is is always in great excess? It is suggested that the authors look at S curves (partitioning of say NH3 and/or HNO3 vs pH) instead of the analysis currently being used.

Line 419 to 421. The loss of buffering capacity of the coarse mode mineral dust during winter pollution events is very interesting and has direct implications for predictions of

NO2 + SO2 oxidation pathways proposed by Wang et al 2016 and Cheng et al 2016. It is suggested that this finding be noted more prominently, maybe even included in the Abstract. However, this period does not seem to be shown in the plots?

The use of the word synthetically throughout the paper is confusing, it is suggested that it not be used since its meaning is unclear.

---

## Referee Comment (RC2) · Anonymous Referee #2 · 26 Jun 2018

This paper presents observations and analysis of the inorganic aerosol system in Beijing for 2017. The pH values are realistic; however, more analysis to verify the methods would make for a stronger paper.

Major comments:

1. Clarify the methodology in terms of how pH was calculated. How was the pH in different size ranges modeled and combined? Even if that appears in other work (as indicated in the text), a quick summary of the method would be useful. Line 209 indicates pH (for the coarse mode?) was determined by ignoring the gas phase and running ISORROPIA in forward mode with zero gas. How was this assumption verified? Figure 2 shows a comparison of total species modeled vs predicted, but that doesn't give a sense of how the size dependent predictions worked. Line 438 indicates that NH3, HNO3, and HCl were determined through iteration when MOUDI data was used.

[Figure]

Was that just for the fine mode particles?

2. Driving factor analysis: The driving factors for pH were obtained by holding all composition, RH, and T parameters at average values and then varying one of the input values (line 291 and thereafter-consider putting some of this method in section 2). A larger change in ALWC, H+air, or pH due to varying one input was interpreted as that input having a major influence on pH. The authors do note that this method will not capture the effect of simultaneous changes in more than one factor.

a. Did the authors consider restricting the output values used to calculate sensitivities (e.g. Table 2) to the space actually probed in the ambient? For example, ALWC output from the simulation varying RH spans 0-140 ug/m3 while most other input parameters did not result in this range of ALWC values. Was 140 ug/m3 ALWC predicted for any of the actual atmospheric conditions? What space is actually probed in the ambient atmosphere in terms of ALWC, H+air, and pH compared to what is probed in the simulated data holding all but one parameter constant?

b. How can the method be evaluated? Does using average inputs result in the same predicted pH that would be obtained by averaging all individual pH predictions from individual inputs? Could the average pH and input be indicated on each panel of Figure 5 to 7? How evenly distributed over the input range are the various inputs? Would it be more appropriate to focus on the interquartile range instead of full range of inputs?

c. Are there units to the quantities in table 2?

d. How would a multiple linear regression analysis differ from the technique of varying one quantity at a time?

e. Could a Monte Carlo method or other technique be used to make sure atmospherically relevant combinations of inputs are being used?

2. Instead of classifying PM2.5 into clean (0-75 ug/m3), polluted, and heavily polluted (>150 ug/m3), it may be illustrative to consider PM2.5 in a continuum. 0-75 ug/m3 on

a daily average is not very clean as it includes concentrations that exceed air quality standards. In addition, by considering PM2.5 concentrations as continuous, you may be able to better determine the association of pH with PM2.5. Consider that the pH for the three classifications is reported with a range/uncertainty that indicates the differences in pH between clean, polluted, and heavily polluted conditions are not statistically significant (values on line 262 overlap). However, if considered as a continuous variable, a regression with confidence interval could be provided and might provide a more robust analysis of the association.

3. Better connect the size resolved measurements with the rest of the text. To what degree did the presence of coarse material drive ambient pH? Do figures 5-7 and the analysis regarding drives of pH only consider fine mode pH?

Minor comments:

1. Line 49. Instead of stating that aerosol acidity is "usually estimated" by the charge balance, I would indicate "sometimes" or "frequently," but not usually as many studies do use a thermodynamic model.

2. Line 52-55 wording indicates ion balance fails because acidity is estimated by aerosol water extract. This doesn't follow well as ion balance (e.g. difference between number of charge equivalent anions and cations) doesn't require extraction.

3. Line 95: may want to indicate models "often" assume internal mixtures (but that is not a requirement).

4. Line 98-99: For this statement indicating nitrate is mainly in the fine mode, does that need to be qualified by indicating a location or time of year? Does fine mode nitrate generally exceed coarse nitrate?

5. Near line 155 and Figure 1: Spring shows a fairly persistent difference in the concentration of PM10 vs PM2.5. Two dust episodes are mentioned. With the exception of these two episodes, do you have a sense of what is contributing to the PM10-PM2.5

material? Late September also indicates an episode in which PM10 is elevated compared to PM2.5.

6. Line 190 indicates water uptake onto hydrophilic organics can be ignored unless the fraction of particle water due to organics is near 1 (100%). Water due to uptake on organics is presumably important even when it is not the sole contributor to particulate water. The threshold of 1 should be removed and perhaps a statement about the potential error incurred by ignoring ALWCo should be added.

7. Text on lines 235-238 seems misplaced or unnecessary.

8. Line 277 highlights sulfate as a driving factor for pH. Sulfate peaked at night during the winter (Figure 4) when photochemical activity is lower. To what degree is the diurnal variation in sulfate driven by chemistry vs meteorology (e.g. planetary boundary layer depth)?

9. Line 284: Is the key difference between the US and Beijing more driven by the higher concentrations or the greater variability in concentrations?

10. Line 384:386 represents a simplified description of ammonia partitioning in which ammonia acts first to neutralize sulfate and then any leftover ammonia can react with nitrate to make ammonium nitrate. Perhaps the authors do not mean this so simply. Reword to reflect the semivolatile nature of ammonia and nitrate.

11. Line 388: Do the authors mean that aerosol would be fully neutralized except for the fact that ammoia is taken up into clouds and precipitation? Reword to reflect the buffering nature of ammonia.

12. Caption to table 2: This table appears to be the sensitivity of acidity, ALWC, and H+air to chemical components (not the other way around). Please clarify caption.

13. Figure 3: use a common color scale for all panels.

14. Figure 5, 6, 7, caption. These figures appear to be the sensitivity of ALWC, H+air,

and pH to chemical components. Reword caption.

15. Line 136: Have you looked at trends from 2013, 2015, and 2017 datasets you have collected?

16. Additional improvements in terms of editing would be useful.

---

## Author Comment (AC1) · 17 Aug 2018

**This paper utilizes unique data sets to predict aerosol pH in the more polluted regions of China. Overall, the paper is a significant contribution since little is known about aerosol pH in these regions and even less on size resolved pH. However, in my view, the analysis is somewhat limited. The authors have an interesting data set that could be more fully utilized to assess the pH predictions, partitioning of inorganic species and understand aerosol pH from a more fundamental standpoint.**

**Response:** Thank you for your valuable comments. Your comments have greatly improved our paper and made this work more rigorous. Please see our point-by-point responses to the comments and the revised manuscript for details. The order of the Figures or Tables in Response is the same as the corresponding Figure or Table appears in the main text and supplemental materials. Moreover, we carefully examined the grammar and expression in the text.

**A suite of important inorganic gases was measured with the MARGA, but they are not significantly discussed in the paper. This is a major oversight. For example, in the comparison of the model to measurements the particle data are shown, but no gas data. For the MOUDI, no gas data is available so the pH is estimated by an iteration method, why not use the MARGA data, which includes gases, to test the sensitivity of pH to this approach?**

**Response:** In the revised manuscript, comparisons and corresponding discussions of predicted and measured $NH_3$, $HNO_3$, $HCl$, $NH_4^+$, $NO_3^-$, $Cl^-$, $\varepsilon(NH_4^+)$ ($NH_4^+/(NH_3+NH_4^+)$, mol/mol), $\varepsilon(NO_3^-)$ ($NO_3^-/(HNO_3+NO_3^-)$, mol/mol)), and $\varepsilon(Cl^-)$ ($Cl^-/(HCl+Cl^-)$, mol/mol) based on MARGA measurement were supplemented, and the detailed information was also showed there.

The data set of MOUDI was obtained during 2013 and 2014, which was not synchronous with the online ion data (obtained in 2016 and 2017), hence an iteration method used in Fang et al. (2017) and Guo et al. (2016) was applied in this work. The MOUDI samples were mainly used to investigate the size distribution of aerosol pH.

**pH is calculated under the assumption of a completely deliquesced particle with no phase separation, all the way down to very low RH, ie, to 30%. These assumptions at low RH need to be justified. Eg, the predicted and measured partitioning of NH3/NH4+, HNO3/NO3-,**

**HCl/Cl- etc (ie include analysis of the gases) could be assessed as a function of pH and see if changes occur at lower RH. Discussion of phase separation in the literature under various conditions (RH, T, O/C) etc should be discussed.**

Response: In this work, particles were assumed in metastable, which means the aerosol is in the only liquid state. However, when the particles are exposed to the quite low RH or the ambient RH reached efflorescent RH, the state of particles may change. Figure 2 and Figure S1-S4 exhibit the comparisons between predicted and measured $NH_3$, $HNO_3$, HCl, $NH_4^+$, $NO_3^-$, $Cl^-$, $\varepsilon(NH_4^+)$ $(NH_4^+/(NH_3+NH_4^+)$, mol/mol), $\varepsilon(NO_3^-)$ $(NO_3^-/(HNO_3+NO_3^-)$, mol/mol)), $\varepsilon(Cl^-)$ $(Cl^-/(HCl+Cl^-)$, mol/mol) based on real-time ion chromatography data, which are all colored by the corresponding RH. It can be seen that agreement between predicted and measured $NH_3$, $NH_4^+$, $NO_3^-$, $Cl^-$ were pretty well. However, measured and predicted partitioning of $HNO_3$ and HCl showed significant discrepancies ($R^2$ of 0.28 and 0.18), which may be attributed to the much lower gas concentrations compared with the particle concentrations, as well as the gas denuder measurement uncertainties from particle collection artifacts (Guo et al., 2018). Obviously, more scatter points deviate from the 1:1 line when ISORROPIA-II runs at RH≤30%, which is much evident in winter and spring. For data with RH ≤ 30%, the predictions were significantly improved when assuming aerosol in stable mode (solid + liquid) (Figure S5-S6). However, the aerosol liquid water was almost zero and cannot be used to predict aerosol pH. It reveals that it is not reasonable to predict the aerosol pH using the thermodynamic model when the RH is relatively low. Consequently, in the revised manuscript, the results were only discussed for data with RH higher than 30%.

A new section (Section 3.3 Gas-particle separation) was added in the revised manuscript. Table 2 exhibited the measured $\varepsilon(NH_4^+)$, $\varepsilon(NO_3^-)$, and $\varepsilon(Cl^-)$ at different RH levels. The measured $\varepsilon(NH_4^+)$, $\varepsilon(NO_3^-)$, and $\varepsilon(Cl^-)$ increased with the elevated RH in all four seasons, indicating more $NH_4^T$, $NO_3^T$, and $Cl^T$ were partitioned into particle phase at higher RH. In winter and spring, $NO_3^T$ and $Cl^T$ were dominated by particle phases. Whereas in summer and autumn, more than half of the $NO_3^T$ and $Cl^T$ were partitioned into the gaseous phase. When the RH reaches above 60%, more than 90% of $NO_3^T$ and 70% of $Cl^T$ were in the particle phase for all four seasons. Compared with $\varepsilon(NO_3^-)$ and $\varepsilon(Cl^-)$, the $\varepsilon(NH_4^+)$ was pretty lower, which may attribute to the higher $NH_3$ mass concentration in the atmosphere. In winter, the average $\varepsilon(NH_4^+)$ were much higher than that in other seasons with the

relatively lower $NH_3$ mass concentration.

**Greater utilization of the gas data could also help the authors understand fundamentally what is driving pH and the sensitivities to various parameters. This could include the use of S curves, as done extensively by Guo et al, to go beyond just simple variation of one variable at a time. Eg, why in the sensitivity analysis do changes in HNO3 not affect pH, but changes in NH3 do? These, and possibly other, more detailed analysis would reduce the sense that the authors simply run the thermodynamic model and plotted results.**

**Response:** In the real ambient air, the thermodynamic process of the aerosol is complicated, it is not easy to tell the effect of one factor on aerosol pH. The ISORROPIA-II can well predict the effect of an input variable on output data. Thus, in this paper, we focus on the sensitivity analysis of single-factor variation, which can reflect the variation tendency of aerosol pH caused by the change of each variable. When running the ISO-II model, the total nitrate ($NO_3^T$, gas+aerosol), total ammonium ($NH_4^T$, gas+aerosol), and total chloride ($Cl^T$, gas+aerosol) are input, and the gas and aerosol phase of these three components would be reapportioned and output. In view of this, it is more reasonable to analyze the impact of $NO_3^T$, $NH_4^T$, and $Cl^T$ on aerosol pH, rather than the impact of a single gas or aerosol phase of $NO_3^T$, $NH_4^T$, and $Cl^T$ on aerosol pH. In the revised manuscript, the data analysis for the sensitivities of aerosol pH to $SO_4^{2-}$, $NO_3^T$, $NH_4^T$, $Cl^T$, RH, and T were fully reorganized and reinspected. More discussions about gas-particle partitioning were added to this section. The impacts of $NO_3^T$, $NH_4^T$, and $Cl^T$ on $\varepsilon(NH_4^+)$, $\varepsilon(NO_3^-)$, and $\varepsilon(Cl^-)$ were also discussed. More detailed information was shown in the revised manuscript.

The $SO_4^{2-}$ and T are two crucial factors affecting aerosol pH variation. Aerosol pH is also sensitive to $NH_4^T$ when $NH_4^T$ in a lower range and sensitive to RH only in summer. Figure 7-9 and S12-S17 show how these factors affecting the ALWC, $H_{air}^+$, and aerosol acidity over four seasons.

**RH:** RH has a different impact on aerosol pH in different seasons. In winter, aerosol pH decreased with the increasing RH, whereas the aerosol pH increased with the increasing RH in summer. In spring and autumn, the RH between 30~83% had little impact on aerosol pH. The explanation for this is that the increased RH actually dilutes the solution and promotes ionization, releasing $H_{air}^+$ and increasing ALWC as well, but the gradient was different. In winter, variation in $H_{air}^+$ caused by RH changes was much larger than variation in ALWC, whereas it showed an opposite tendency in

summer. In autumn and spring, variation in $H_{air}^+$ caused by RH changes was slightly higher than variation in ALWC. The different impact of RH on aerosol pH indicated that the dilution effect of ALWC on $H_{air}^+$ is obvious only in summer, the high RH during the severe haze in winter could increase the aerosol acidity.

**T:** At high ambient temperature, $\epsilon(NH_4^+)$, $\epsilon(NO_3^-)$, and $\epsilon(Cl^-)$ all showed a decreased tendency (Figure 10 and S19). And $NH_4^+$, $NO_3^-$, and $Cl^-$ were volatilized partially, the procedure of $NH_4^+$ $\rightarrow NH_3$ released one $H^+$ to particle phase, whereas the procedure of $NO_3^- \rightarrow HNO_3$ and $Cl^- \rightarrow HCl$ needs one $H^+$ from the particle phase. Compared with the loss of $NO_3^-$ from $NH_4NO_3$ as well as $Cl^-$ from $NH_4Cl$, greater loss of $NH_4^+$ from $NH_4NO_3$, $NH_4Cl$, and $(NH_4)_2SO_4$ resulted in a net increase in particle $H^+$ and lower pH. In addition, molality-based equilibrium constants ($H^*$) of $NH_3$-$NH_4^+$ partitioning decreased faster with increasing temperature when compared with that of $HNO_3$-$NO_3^-$ partitioning, resulting in a net increase in particle $H^+$ (Guo et al., 2018). Moreover, higher ambient temperature tends to lower ALWC, which further decrease the aerosol pH. The wide range of ambient temperature in autumn made a significant impact on aerosol pH in the sensitivity analysis.

**$SO_4^{2-}$:** $SO_4^{2-}$ has a key role in aerosol acidity, especially in winter and spring (Figure 9, S14, S17). More $H^+$ are released into particle phase during the formation of $SO_4^{2-}$, forming one $SO_4^{2-}$ can release two $H^+$. In the sensitivity test, the aerosol pH decreases about 1.6 (4.1 to 2.5), 4.9 (5.1 to 0.2), 1.0 (3.6 to 2.6), and 0.9 (4.0 to 3.1) unit with $SO_4^{2-}$ concentration goes up from 0 to 40 μg m$^{-3}$ in spring, winter, summer, and autumn, respectively. In spring and winter, the ALWC is low, the variation of $SO_4^{2-}$ mass concentration could generate dramatic changes in $H_{air}^+$. In section 3.1, the aerosol pH was lowest in summer whereas highest in winter, which was consistent with the $SO_4^{2-}$ mass faction in total ions. The $SO_4^{2-}$ mass faction in total ions in summer was highest among four seasons with 32.4%±11.1%, whereas it was lowest in winter with 20.9%±4.4%.

**$NO_3^T$:** The impact of $NO_3^-$ on aerosol pH was also different, which is related to the averages of input $NH_4^T$ in different seasons. In winter, the aerosol pH decreased with increasing $NO_3^T$ concentration, whereas little impact was found in summer (Figure 9). In spring and autumn, the aerosol pH increases first and then drops with the increasing $NO_3^T$ concentration (Figure S14, S17). In winter, the $NH_4^T$ mass concentration was low. As $NO_3^T$ increases, all $NH_3$ was converted into $NH_4^+$ ($\epsilon(NH_4^+) \approx 1$). However, $HNO_3$ continues to dissolve and releases $H_{air}^+$, resulting in the

decrease of aerosol pH. In summer, the averages of $NO_3^T$ and $Cl^T$ was relatively low but the $NH_4^T$ was excessive, the highest $\varepsilon(NH_4^+)$ was only 0.6 with the corresponding highest $NO_3^T$. The excessive $NH_3$ could provide continuous buffering to the increasing $NO_3^T$, together with a significant dilution of ALWC on $H_{air}^+$, leads to the little changes in aerosol pH. In spring and autumn, the increasing aerosol pH with elevated $NO_3^T$ in lower range attributed to the dilution of ALWC to $H_{air}^+$. $H_{air}^+$ concentration increased exponentially with elevated $NO_3^T$ concentration, especially at higher $NO_3^T$ concentrations, whereas the ALWC increase linearly with elevated $NO_3^T$ concentration (Figure S12-S17), hence ALWC plays a dominant role when the $NO_3^T$ concentration is low. With the further increase of $NO_3^T$, the variation in $H_{air}^+$ caused by $NO_3^T$ addition is larger than variation in ALWC, leading to the decrease of aerosol pH. Besides, the relationship between $NO_3^T$ and $\varepsilon(NH_4^+)$ in the sensitivity analysis showed that decreasing $NO_3^T$ could lower the $\varepsilon(NH_4^+)$ effectively (Figure 11 and S20), which helps $NH_3$ maintain in the gas phase.

**$NH_4^T$:** The relationship between aerosol pH and $NH_4^T$ was nonlinear. $NH_4^T$ in lower range had a significant impact on aerosol pH (Table S2), and higher $NH_4^T$ generated limited pH change (Figure 9, S14, S17). Elevated $NH_4^T$ could reduce $H_{air}^+$ exponentially and slightly increase ALWC when the other input parameters were held constant. As the $NH_4^T$ increases, $H_{air}^+$ are consumed swiftly during the dissolution of $NH_3$ and the further reaction with $SO_4^{2-}$, $NO_3^-$, and $Cl^-$. And the elevated $NH_4^T$ increased the $\varepsilon(NO_3^-)$ and $\varepsilon(Cl^-)$ when $NO_3^T$ and $Cl^T$ were fixed (Figure 11 and S20), which means the elevated $NH_4^T$ alters the gas-particle partition and shifts more $NO_3^T$ and $Cl^T$ into particle phase, leading to the deliquescence of additional nitrate and chloride and increase of ALWC. It seems that $NH_3$ emission control is a good way to reduce $NO_3^-$. However, the relationship between $NH_4^T$ and $\varepsilon(NO_3^-)$ in the sensitivity analysis (Figure 11 and S20) showed that the $\varepsilon(NO_3^-)$ response to $NH_4^T$ control is highly nonlinear, which means the decrease of nitrate is effective only when the $NH_4^T$ is greatly reduced. The same result was obtained from Guo et al (2018) using the S curve method.

The ratio of [TA]/2[TS] provides a qualitative description for the ammonia abundance, where [TA] and [TS] are the total (gas + aqueous + solid) molar concentrations of ammonia and sulfate. The rich-ammonia is defined as [TA] > 2[TS], while if the [TA] ≤ 2[TS], then it is defined as poor-ammonia (Seinfeld and Pandis, 2016). In this work, the ratio of [TA]/2[TS] is much higher than 1 and belongs to rich-ammonia (Figure. S21). Although $NH_3$ in the NCP is abundant, the aerosol pH

is far from neutral, which may attribute to the limited ALWC. Compared to the liquid water content in clouds and precipitation, ALWC is much lower, hence the dilution of aerosol liquid water to $H_{air}^+$ is weak.

**$Cl^T$:** $Cl^T$ has a relatively larger impact on aerosol pH in winter and spring compared to summer and autumn. Except for winter, the $Cl^T$ mass concentration was generally lower than 10 $\mu g$ $m^{-3}$, which accounted for the little impact on aerosol pH. On account of the low level of $Cl^T$, the dilution of ALWC on $H_{air}^+$ plays a dominant role, generating the aerosol pH increase with elevated $Cl^T$. However, similar to $NO_3^T$, higher $Cl^T$ could decrease the aerosol pH.

**$Ca^{2+}$:** In fine particles, $Ca^{2+}$ mass concentration was generally low. In the output of ISORROPIA-II, Ca existed as $CaSO_4$ (slightly soluble). Elevated $Ca^{2+}$ concentration could increase the aerosol pH by decreasing $H_{air}^+$ and ALWC (Figure. S18), the decreased $H_{air}^+$ results from the buffering capacity of $Ca^{2+}$ to the acid species, while the decreased ALWC result from the weak water solubility of $CaSO_4$. As discussed in Section 3.1, on clean conditions, the aerosol pH could reach 6~7 when the mass fraction of $Ca^{2+}$ was high, hence the role of mineral ions on aerosol pH could not be ignored in seasons (such as spring) or regions where mineral dust was an important source of fine particles. Due to the strict control measures for road dust, construction sites, and other bare ground, the nonvolatile cations in $PM_{2.5}$ decreased significantly in NCP.

**Table 2.** The averaged ambient temperature and $\varepsilon(NH_4^+)$, $\varepsilon(NO_3^-)$, $\varepsilon(Cl^-)$ at different ambient RH levels in four seasons.

| | RH | T, ℃ | $\varepsilon(NH_4^+)$ | $\varepsilon(NO_3^-)$ | $\varepsilon(Cl^-)$ |
|---|---|---|---|---|---|
| | ≤ 30 % | 24.8 ± 3.7 | 0.17±0.14 | 0.84±0.12 | 0.67±0.24 |
| Spring | 30~60 % | 20.6 ± 3.8 | 0.25±0.14 | 0.91±0.06 | 0.82±0.16 |
| | >60 % | 15.8 ± 2.7 | 0.28±0.12 | 0.96±0.03 | 0.96±0.06 |
| | ≤ 30 % | 5.4 ± 5.3 | 0.31±0.13 | 0.78±0.12 | 0.89±0.14 |
| Winter | 30~60 % | 1.0 ± 3.6 | 0.50±0.21 | 0.89±0.10 | 0.97±0.03 |

| | | | | | |
|---|---|---|---|---|---|
| | >60 % | -1.9 ± 2.1 | 0.60±0.20 | 0.96±0.03 | 0.99±0.01 |
| | ≤ 30 % | 35.6± 0.4 | 0.06±0.02 | 0.35±0.20 | 0.39±0.17 |
| Summer | 30~60 % | 29.6 ± 4.2 | 0.17±0.11 | 0.65±0.23 | 0.43±0.16 |
| | >60 % | 25.2 ± 3.8 | 0.26±0.12 | 0.90±0.12 | 0.71±0.15 |
| | ≤ 30 % | 21.7± 7.5 | 0.07±0.06 | 0.49±0.25 | 0.45±0.21 |
| Autumn | 30~60 % | 20.8± 6.3 | 0.21±0.14 | 0.82±0.19 | 0.67±0.21 |
| | >60 % | 14.9 ± 5.7 | 0.30±0.19 | 0.92±0.10 | 0.86±0.13 |

[Figure]

**Figure 2.** Comparisons of predicted and measured $NH_3$, $HNO_3$, HCl, $NH_4^+$, $NO_3^-$, $Cl^-$, $\varepsilon(NH_4^+)$, $\varepsilon(NO_3^-)$, $\varepsilon(Cl^-)$ colored by RH. In this Figure, the real-time data in four seasons were put together, and the comparisons for each season were shown in Figure S1-S4.

Spring

[Figure]

**Figure S1.** Comparisons of predicted and measured NH₃, HNO₃, HCl, NH₄⁺, NO₃⁻, Cl⁻, ε(NH₄⁺),

ε(NO₃⁻), ε(Cl⁻) colored by RH in spring.

[Figure]

**Figure S2.** Comparisons of predicted and measured NH₃, HNO₃, HCl, NH₄⁺, NO₃⁻, Cl⁻, ε(NH₄⁺), ε(NO₃⁻), ε(Cl⁻) colored by RH in winter.

[Figure]

**Figure S3.** Comparisons of predicted and measured NH$_3$, HNO$_3$, HCl, NH$_4^+$, NO$_3^-$, Cl$^-$, $\varepsilon$(NH$_4^+$), $\varepsilon$(NO$_3^-$), $\varepsilon$(Cl$^-$) colored by RH in summer.

[Figure]

**Figure S4.** Comparisons of predicted and measured NH$_3$, HNO$_3$, HCl, NH$_4^+$, NO$_3^-$, Cl$^-$, $\varepsilon$(NH$_4^+$), $\varepsilon$(NO$_3^-$), $\varepsilon$(Cl$^-$) colored by RH in autumn.

[Figure]

**Figure S5.** Comparisons of predicted and measured NH₃, HNO₃, HCl, NH₄⁺, NO₃⁻, Cl⁻, ε(NH₄⁺), ε(NO₃⁻), ε(Cl⁻) at the RH≤30%, the ISORROPIA-II runs in stable mode (solid + liquid).

[Figure]

**Figure S6.** Comparisons of predicted and measured $NH_3$, $HNO_3$, HCl, $NH_4^+$, $NO_3^-$, $Cl^-$, $\varepsilon(NH_4^+)$, $\varepsilon(NO_3^-)$, $\varepsilon(Cl^-)$ at the RH≤30%, the ISORROPIA-II runs in metastable mode.

[Figure]

**Figure 3.** Comparisons of predicted and iterative $NH_3$, $HNO_3$, $HCl$, as well as the predicted and measured $NH_4^+$, $NO_3^-$, $Cl^-$, $\varepsilon(NH_4^+)$, $\varepsilon(NO_3^-)$, $\varepsilon(Cl^-)$ colored by particle size. In this Figure, all MOUDI data were put together.

[Figure]

**Figure 7.** Sensitivities of $H_{air}^+$ to $SO_4^{2-}$, $NO_3^T$, $NH_4^T$, $Cl^T$, as well as meteorological

parameters (RH, T) in summer and winter.

[Figure]

**Figure 8.** Sensitivities of ALWC to $SO_4^{2-}$, $NO_3^T$, $NH_4^T$, $Cl^T$, as well as meteorological

parameters (RH, T) in summer and winter.

[Figure]

**Figure 9.** Sensitivities of aerosol pH to $SO_4^{2-}$, $NO_3^T$, $NH_4^T$, $Cl^T$, as well as meteorological parameters (RH, T) in summer and winter.

[Figure]

**Figure 10.** Sensitivities of $\varepsilon(NH_4^+)$, $\varepsilon(NO_3^-)$, $\varepsilon(Cl^-)$ to RH and T colored by aerosol pH in summer and winter.

[Figure]

**Figure 11.** Sensitivities of ε(NH₄⁺), ε(NO₃⁻), ε(Cl⁻) to $NO_3^T$, $NH_4^T$, $Cl^T$ colored by aerosol pH in summer and winter.

***Specific Comments:***

1. Line 37 change specials to species

**Response:** In the revised manuscript, the word "specials" has been changed to "species"

2. Line 202 and following, it is not just lack of NH3 data that can affect predicted pH, what about HNO3, HCl, etc?

**Response:** Thank you for your important advice, the gaseous precursor $NH_3$, $HNO_3$, HCl were all important for predicting pH with the forward mode. Actually, the $NH_3$, $HNO_3$, HCl obtained from the iteration method were all used in predicting the size-resolved aerosol pH. Here we missed other gases' names, in the revised manuscript, it has been corrected.

3. Line 205, how much did the pH change when the iteration approach is used? Or, were the predicted gas species concentrations reasonable relative to what was measured during the MARGA study period.

**Response:** (1) As explained above, the MOUDI sampling was not synchronous with MARGA observation in time, hence the gas species concentrations were not available for MOUDI samples.

(2) The fine mode aerosol pH determined through the iteration procedure was higher than that with no gaseous species. In summer and autumn, the difference of fine mode aerosol pH was 0.1~1 between the predictions with and without gaseous species, while it was 0.1~2.9 in winter. The overall low RH in winter resulted in the low ALWC, hence in the gas-particle portioning procedure more $NH_4^+$ was portioned into the gas phase and led to the low aerosol pH for fine mode particles.

[Figure]

**Figure R1** The averaged size-resolved aerosol pH in three seasons predicted with three assumptions: (1) predicted with no iterative gases, (2) predicted assuming lack of equilibrium with gas phase for coarse mode particles, (3) predicted assuming all particles in equilibrium with the gas phase.

4. Line 229 is superfluous, it is well known that low pH means high acidity.

**Response:** The sentence "implying the higher aerosol acidity" has been deleted in the revised manuscript.

5. Line 235 to 238: this paragraph seems out of place.

**Response:** Thank you for your advice, this paragraph has been deleted from the revised manuscript.

6. Fig 3 caption, what does transverse direction mean on a polar plot?

**Response:** In the polar plot, the shaded contour indicates the average of variables for varying wind speeds (radial direction) and wind directions (transverse direction). And this was explicated in Figure 5.

7. Line 246 change souther to southern.

**Response:** "souther" has been changed to "southern" in the revised manuscript.

8. Line 276-286. From Fig 4 it does not appear that pH and sulfate diurnal trends are always the same (actually inverse), as stated. Looks like a stronger inverse trend with liquid water. The more quantitative analysis is needed to support the statements made in this section.

**Response:** Thanks for your suggestion. In fact, we want to express that the diurnal variation of aerosol acidity (not aerosol pH) is consistent with the diurnal variation of $SO_4^{2-}$ over four seasons.

In the revised manuscript, the diurnal variation of $NO_3^-$ was added in Figure 6. The diurnal variation of $NO_3^-$ in winter and spring agreed well with the aerosol acidity. But in summer and autumn, the agreement was not well. Figure S11 shows the relationship between mass concentrations of $SO_4^{2-}$ and $NO_3^-$ and aerosol pH at different ALWC levels for all four seasons. At the relatively low ALWC, the increasing $SO_4^{2-}$ could decrease the aerosol pH obviously; at the relatively high ALWC, the negative correlation still existed between $SO_4^{2-}$ mass concentration and aerosol pH. On the contrary, a weak positive correlation was found between $NO_3^-$ and aerosol pH at the relatively low ALWC and the aerosol pH was almost invariable with the $NO_3^-$ mass concentration at the relatively high ALWC. Compared with the $NO_3^-$, the $SO_4^{2-}$ had a greater effect on aerosol pH. But when the ALWC was high enough (for example, higher than 100 μg m$^{-3}$), the impact of dilution of ALWC to the $H_{air}^+$ was more significant.

[Figure]

**Figure S10.** The relationship between $SO_4^{2-}$ and $NO_3^-$ mass concentration and aerosol pH at different ALWC levels.

9. Line 327, provide a physical explanation for the U shape dependency of H+ on NO3-

**Response:** As mentioned above, we discussed the dependency of $H^+_{air}$ on $NO_3^T$ instead of the $NO_3^-$. In addition, we find that the shape of the curve for the dependency of $H_{air}^+$ on total nitrate was also affected by the input average RH. In the revised manuscript, the data of RH lower than 30% were excluded. Similar with other seasons, the elevated $NO_3^T$ could increase the $H_{air}^+$ exponentially.

10. Line 330-331: Is it really true that there is a straightforward relationship between NH3 and H+ over broad NH3 concentration ranges? Ie, will increases in NH3 always lead to higher dissolved NH3? Technically it may be true, but the relationship may be highly nonlinear under certain conditions. This statement seems too broad.

**Response:** Thanks for your advice, the statement here is not rigorous. The relationship between the reduction of $H_{air}^+$ and the increase of $NH_3$ was indeed nonlinear, and the increasing $NH_3$ could only promote $NH_3$ dissolution to a certain extent. The purpose of the statement of Line 330-331 was to explain the decrease of aerosol pH resulting from the elevated $NH_4^T$. As you commented, the gas-particle partition ($\varepsilon(NH_4^+)$, $\varepsilon(NO_3^-)$, $\varepsilon(Cl^-)$) could help us understand fundamentally what is driving pH. We explain the decrease of aerosol pH resulting from the elevated $NH_4^T$ in detail in your 13[th] comment.

11. Line 335, this is an obvious statement based on Eq (1). In fact much of the discussion throughout relating pH, H+ and LWC are obvious from Eq (1).

**Response:** The corresponding sentences in line 335 has been deleted in the revised manuscript.

12. Line 358 and on regarding changes in pH with NO3-. The authors discuss the trends they observe in the sensitivity analysis and NO3-/SO4=, but never provide an explanation. By just reporting of results, the value of this work is greatly limited, despite the what could be done with this unique data set.

**Response:** During the thermodynamic process of aerosol, all the SO4= would dissolve in the aerosol liquid water, the amount of sulfate can be considered stable and it would not be affected by the NO3-. From the point of the model, the concentrations of NO3- and SO4= are both the output of ISO-II. Thus, the ratio of NO3-/SO4= can only reflect the objective state of particles, it is not the

cause or the indicator of aerosol pH. After careful consideration, we decide to remove this part of the discussion.

13. Lines 380 and on regarding TA and TS. Most of these statements are technically incorrect (although, from a broad perspective they may have a grain of truth to them). The authors data show that the pH is far from neutral despite it being NH3 rich. This analysis largely continues misconceptions of how aerosol composition depends on interactions between SO4=, NH3, NH4+, HNO3, NO3- and LWC. Eg, is HNO3 only taken up once sulfate is so-called neutralized; maybe this can be tested with the data (there should be no NO3- and then a sudden jump in NO3- when [TA]/2[TS] is greater than 1. Another example, why does pH vary, even for this data set, if NH3 is is always in great excess? It is suggested that the authors look at S curves (partitioning of say NH3 and/or HNO3 vs pH) instead of the analysis currently being used.

**Response:** Firstly, we think you are right, our statements here have some problems. Figure S21 showed that the elevated [TA]/2[TS] didn't increase the $NO_3^-$ mass concentration, high $NO_3^-$ mass concentration occurred when [TA]/2[TS] varies over a wide range (2~15). But in the NCP, the excess of ammonia in the atmosphere is indeed true, the ratio of [TA]/2[TS] is much higher than 1. The poor-ammonia cases were not observed in this work.

The relationship between aerosol pH and $NH_4^T$ was nonlinear. $NH_4^T$ in lower range had a significant impact on aerosol pH (Table S2), and higher $NH_4^T$ generated limited pH change (Figure 9, S14, S17). Elevated $NH_4^T$ could reduce $H_{air}^+$ exponentially and slightly increase ALWC when the other input parameters were held constant. As the $NH_4^T$ increases, $H_{air}^+$ are consumed swiftly during the dissolution of $NH_3$ and the further reaction with $SO_4^{2-}$, $NO_3^-$, and $Cl^-$. And the elevated $NH_4^T$ increased the $\varepsilon(NO_3^-)$ and $\varepsilon(Cl^-)$ when $NO_3^T$ and $Cl^T$ were fixed (Figure 11 and S20), which means the elevated $NH_4^T$ alters the gas-particle partition and shifts more $NO_3^T$ and $Cl^T$ into particle phase, leading to the deliquescence of additional nitrate and chloride and increase of ALWC.

Although $NH_3$ in the NCP is abundant, the aerosol pH is far from neutral, which may attribute to the limited ALWC. Compared to the liquid water content in clouds and precipitation, ALWC is much lower, hence the dilution of aerosol liquid water to $H_{air}^+$ is weak.

In our opinion, the ALWC, $H_{air}^+$, aerosol pH, $\varepsilon(NH_4^+)$, $\varepsilon(NO_3^-)$, and $\varepsilon(Cl^-)$ are all the output of

ISO-II. They reflect an objective state of particles. Accordingly, it is reasonable to discuss the impact of input variables on output parameters with the results of ISO-II. On the basis of overall moderate aerosol acidity, the variation of aerosol pH is related to aerosol composition and meteorological conditions (RH and T). In the sensitivity analysis of this work, the influence of single variables on aerosol acidity is explicit. In the ambient atmosphere, multiple variables interact with each other, and aerosol acidity largely depends on the dominant factor. The relationship between $\varepsilon(NO_3^-)$, $\varepsilon(Cl^-)$ and aerosol pH was analyzed by S curves proposed by Guo et al (2016, 2017), which were calculated based on the average temperature, aerosol liquid water, and activity coefficients. Their result showed that for a given ALWC and T, about 4 pH units increase are needed when the $\varepsilon(NO_3^-)$ and $\varepsilon(Cl^-)$ varies from 0 to 100%.

14. Line 419 to 421. The loss of buffering capacity of the coarse mode mineral dust during winter pollution events is very interesting and has direct implications for predictions of NO2 + SO2 oxidation pathways proposed by Wang et al 2016 and Cheng et al 2016. It is suggested that this finding be noted more prominently, maybe even included in the Abstract. However, this period does not seem to be shown in the plots?

**Response:** Wang et al (2016) and Cheng et al (2016) advocate that the aqueous oxidation of $SO_2$ by $NO_2$ is key to efficient sulfate formation but is only feasible under two atmospheric conditions: on fine aerosols with high relative humidity and $NH_3$ neutralization (aerosol pH $\sim$7) or under cloud conditions. Their results focused on the fine particles, hence whether the loss of buffering capacity of the coarse mode mineral dust during winter pollution has a direct implication on their results remains to be discussed. But for fine particles, excessive $NH_3$ does not raise aerosol pH sufficiently.

15. The use of the word synthetically throughout the paper is confusing, it is suggested that it not be used since its meaning is unclear.

**Response:** The word "synthetically" has been deleted in the revised manuscript.

**References**

Guo, H., Sullivan, A. P., Campuzano-Jost, P., Schroder, J. C., Lopez-Hilfiker, F. D., Dibb, J. E., Jimenez, J. L., Thornton, J. A., Brown, S. S., Nenes, A., Weber, R. J.: Fine particle pH and the

partitioning of nitric acid during winter in the northeastern United States, J. Geophys. Res. Atmos., 121, 10355-10376, 2016.

Guo, H., Liu, J., Froyd, K. D., Roberts, J. M., Veres, P. R., Hayes, P. L., Jimenez, J. L., Nenes, A., and Weber, R. J.: Fine particle pH and gas–particle phase partitioning of inorganic species in Pasadena, California, during the 2010 CalNex campaign, Atmos. Chem. Phys., 17, 5703-5719, 2017.

Guo, H., Otjes, R., Schlag, P., Scharr, A. K., Nenes, A., Weber, R. J.: Effectiveness of ammonia reduction on control of fine particle nitrate. Atmos. Chem. Phys. Discuss, https://doi.org/10.5194/acp-2018-378, 2018.

Huang, X. J., Liu, Z. R., Liu, J. Y., Hu, B., Wen, T. X., Tang, G. Q., Zhang, J. K., Wu, F. K., Ji, D. S., Wang, L. L., Wang, Y. S.: Chemical characterization and source identification of $PM_{2.5}$ at multiple sites in the Beijing–Tianjin–Hebei region, China, Atmos. Chem. Phys., 17, 12941–12962, 2017.

Ma, Q.X., Wu, Y.F., Zhang, D. Z., Wang, X.J., Xia, Y.J., Liu, X.Y., Tian, P., Han, Z.W., Xia, X.G., Wang, Y., Zhang, R.J.: Roles of regional transport and heterogeneous reactions in the $PM_{2.5}$ increase during winter haze episodes in Beijing, Sci. Total Environ., 599-600, 246-253, 2017.

Zhao, P. S., Chen, Y. N., Su, J.: Size-resolved carbonaceous components and water-soluble ions measurements of ambient aerosol in Beijing. J. Environ. Sci., 54, 298-313, 2017.

---

## Author Comment (AC2) · 17 Aug 2018

This paper presents observations and analysis of the inorganic aerosol system in Beijing for 2017. The pH values are realistic; however, more analysis to verify the methods would make for a stronger paper.

**Response:**

Thanks for your important comments, which are very useful to make our paper more rigorous. Please see our point-by-point responses to the comments and the revised manuscript for details. The order of the Figures or Tables in Response is the same as the corresponding Figure or Table appears in the main text and supplemental materials.

Major comments:

**1. Clarify the methodology in terms of how pH was calculated. How was the pH in different size ranges modeled and combined? Even if that appears in other work (as indicated in the text), a quick summary of the method would be useful. Line 209 indicates pH (for the coarse mode?) was determined by ignoring the gas phase and running ISORROPIA in a forward mode with zero gas. How was this assumption verified? Figure 2 shows a comparison of total species modeled vs predicted, but that doesn't give a sense of how the size-dependent predictions worked. Line 438 indicates that NH3, HNO3, and HCl were determined through iteration when MOUDI data was used. Was that just for the fine mode particles?**

**Response:**

The data set of MOUDI was obtained during 2013 and 2015, which was not synchronous with the online ion data (obtained in 2016 and 2017). There was no observation of gas precursors during the periods of MOUDI sampling, hence an iteration method used in Fang et al. (2017) and Guo et al. (2016) was applied in this work. As a brief summary, the predicted $NH_3$, $HNO_3$, and HCl concentrations from the $i$-1 run were applied to the $i$th iteration, until the gas concentrations converged. Based on these iterative gas phase concentrations, each MOUDI stage's measured aerosol ion concentrations and estimated gas concentrations, as well as the averaged RH and T during each group sampling time, were input the ISORROPIA-II to determine pH for each stage. The particles at each size bin were assumed to be internally mixed.

The comparisons of iterative and predicted $NH_3$, $HNO_3$, and HCl as well as measured and predicted

$NO_3^-$, $NH_4^+$, $Cl^-$, $\varepsilon(NH_4^+)$, $\varepsilon(NO_{3-})$, and $\varepsilon(Cl^-)$ for data from MOUDI samples were showed in Figure 3. The previous study showed that coarse mode particles are very difficult to reach equilibrium with the gaseous precursors due to kinetic limitations (Dassios et al., 1999; Cruz et al., 2000). Assuming coarse mode particles in equilibrium with the gas phase could result in a large bias between measured and predicted $NO_3^-$ and $NH_4^+$ in coarse mode particles (Fang et al, 2017). We also find that in this work, it can be clearly seen that assuming coarse mode particles in equilibrium with the gas phase could overpredict $NO_3^-$ and $Cl^-$ and underestimate $NH_4^+$ in the coarse mode (the blue scatters), which could subsequently underestimate the coarse mode aerosol pH. Compared with the coarse mode particles, the measured and predicted $NO_3^-$, $NH_4^+$, and $Cl^-$ agreed very well in fine mode particles. Considering the kinetic limitations and nonideal gas-particle partitioning in coarse mode particles, the aerosol pH in coarse mode was determined by ignoring the gas phase.

[Figure]

**Figure 3.** The comparisons of iterative and predicted $NH_3$, $HNO_3$, HCl as well as measured and predicted $NO_3^-$, $NH_4^+$, $Cl^-$, $\varepsilon(NH_4^+)$ $\varepsilon(NO_{3-})$ $\varepsilon(Cl^-)$ for data from MOUDI samples, which all

colored by particle size.

[Figure]

**Figure R1** The averaged size-resolved aerosol pH in three seasons predicted with three assumptions: (1) predicted with no iterative gases, (2) predicted assuming lack of equilibrium with gas phase for coarse mode particles, (3) predicted assuming all particles in equilibrium with the gas phase.

**2. Driving factor analysis: The driving factors for pH were obtained by holding all composition, RH, and T parameters at average values and then varying one of the input values (line 291 and thereafter-consider putting some of this method in section 2). A larger change in ALWC, H+air, or pH due to varying one input was interpreted as that input having a major influence on pH. The authors do note that this method will not capture the effect of simultaneous changes in more than one factor.**

**Response:**

Thanks for your important advice. The detailed introduction of the method about aerosol pH driving factor analysis has been put in section 2.5. In the real ambient air, the thermodynamic process of the aerosol is complicated, it is not easy to tell the effect of one certain factor on the aerosol pH. The ALWC, $H_{air}^+$, and aerosol pH are all the output of ISORROPIA-II. They reflect an objective state of particles. Considering the relative independence between input parameters, it is reasonable to discuss the influence of input variables on output parameters with the results of ISORROPIA-II. Thus, in this paper, we focus on the sensitivity analysis of single-factor variation, which can reflect the variation tendency of aerosol pH caused by the change of each variable.

**a. Did the authors consider restricting the output values used to calculate sensitivities (e.g.**

**Table 2) to space actually probed in the ambient? For example, ALWC output from the simulation varying RH spans 0-140 ug/m3 while most other input parameters did not result in this range of ALWC values. Was 140 ug/m3 ALWC predicted for any of the actual atmospheric conditions? What space is actually probed in the ambient atmosphere in terms of ALWC, H+air, and pH compared to what is probed in the simulated data holding all but one parameter constant?**

**Response:** All data used in the sensitivity analysis were based on the actual observation, not randomly generated simulation data, which helps us capture a more real impact. When the RH was considered as a variable, ALWC output spans 0-140 $\mu g/m^3$, this mainly attributed to the vital impact of RH on ALWC, especially when the RH was higher than 80% owing to the exponential increase of ALWC with the RH. Whereas in other simulated cases, the averaged RH was generally within 50% ~ 75%, hence the output ALWC was relatively low. In summer and autumn, the actual ALWC was even more than 140 $\mu g/m^3$ when both aerosol components and RH were high.

**b. How can the method be evaluated? Does using average inputs result in the same predicted pH that would be obtained by averaging all individual pH predictions from individual inputs? Could the average pH and input be indicated on each panel of Figure 5 to 7? How evenly distributed over the input range are the various inputs? Would it be more appropriate to focus on the interquartile range instead of a full range of inputs?**

**Response:** The average value and variation range for each variable in all four seasons were listed in Table S1 and Figure S7. The aerosol pH1 is the value by averaging all individual pH predictions from each input variable, for example, the average aerosol pH was 3.74±0.47 when the $SO_4^{2-}$ was regarded as an input variable while other input parameters were fixed with the average value. The aerosol pH2 is the value by using average inputs for all input parameters. In theory, pH1 and pH2 cannot be the same, otherwise, the effect of the variables on aerosol pH will not be reflected.

Table S1 The average value and range for each variable in all four seasons, as well as the two average aerosol pH types. The aerosol pH1 is the value by averaging all individual pH predictions from each continuous input variable, for example, the average aerosol pH was 3.74±0.47 when the $SO_4^{2-}$ was regarded as a continuous input variable

while other input parameters were fixed with the average value. The aerosol pH2 is the value by using average inputs

for all input parameters. The unit of chemical components is μg m$^{-3}$.

| Spring | SO$_4^{2-}$ | NH$_4^T$ | NO$_3^T$, | Cl$^T$ | RH, % | T, ℃ | Ca | Na | K | Mg |
|---|---|---|---|---|---|---|---|---|---|---|
| Average input | 8.4 | 25.7 | 13.5 | 1.1 | 52 | 20.9 | 1.29 | 0.20 | 0.34 | 0.3 |
| Variable range | 3.0~41.4 | 0.1~33.9 | 0.4~77.6 | 0.03~6.27 | 30~92 | 10.0~33.3 | 0.1~3.0 | | | |
| pH1 | 3.74±0.47 | 3.69±0.19 | 3.65±0.53 | 3.81±0.09 | 3.79±0.05 | 3.81±0.27 | 3.73±0.16 | | | |
| pH2 | | | | | 3.82 | | | | | |
| Winter | SO$_4^{2-}$ | NH$_4^T$ | NO$_3^T$, | Cl$^T$ | RH, % | T, ℃ | Ca | Na | K | Mg |
| Averaged | 7.3 | 12.2 | 14.3 | 3.0 | 52 | 2.7 | 0.2 | 0.40 | 1.0 | 0.2 |
| Ranges | 2.0~34.6 | 1.3~46.7 | 0.8~49.3* | 0.02~25.2 | 30~94 | -8.7~16.2 | 0.01~0.7 | | | |
| pH1 | 4.32±1.21 | 3.86±1.04 | 4.27±0.48 | 4.27±0.16 | 4.39±0.18 | 4.36±0.29 | 4.36±0.04 | | | |
| pH2 | | | | | 4.36 | | | | | |
| Summer | SO$_4^{2-}$ | NH$_4^T$ | NO$_3^T$, | Cl$^T$ | RH, % | T, ℃ | Ca | Na | K | Mg |
| Averaged | 8.6 | 26.8 | 10.2 | 0.6 | 74 | 26.1 | 0.5 | 0.60 | 0.2 | 0.1 |
| Ranges | 0.6~40.1 | 1.2~69.6 | 0.3~59.8 | 0.1~2.8 | 30~97 | 14.2~38.1 | 0.02~2.9 | | | |
| pH1 | 3.43±0.27 | 3.31±0.32 | 3.31±0.12 | 4.38±0.03 | 3.40±0.27 | 3.37±0.20 | 3.38±0.06 | | | |
| pH2 | | | | | 3.38 | | | | | |
| Autumn | SO$_4^{2-}$ | NH$_4^T$ | NO$_3^T$, | Cl$^T$ | RH, % | T, ℃ | Ca | Na | K | Mg |
| Averaged | 9.3 | 27.8 | 20.3 | 1.0 | 72 | 16.4 | 0.4 | 0.3 | 0.2 | 0.1 |
| Ranges | 0.3~54.7 | 3.2~67.5 | 0.2~90.5 | 0.06~5.17 | 30~97 | -1.1~33.3 | 0.02~2.3 | | | |
| pH1 | 3.85±0.23 | 3.60±0.58 | 3.70±0.12 | 3.84±0.04 | 3.94±0.10 | 3.84±0.29 | 3.84±0.03 | | | |
| pH2 | | | | | 3.84 | | | | | |

[Figure]

**Figure S7**. The distribution of each input variable for sensitivity analysis in four seasons

**c. Are there units to the quantities in table 2?**

**Response:** Units to the quantities in table 2 were missed in the manuscript, in the revised manuscript, we replace the deviation by relative standard deviation as the evaluation target, hence the unit is unified to %.

**d. How would a multiple linear regression analysis differ from the technique of varying one quantity at a time?**

**Response:** The relationships between input variables and aerosol pH are not simply linear. The method in this work based on the overall accurate relationship between variables rather than the permutation and combination in the mathematical sense, the latter may subversively change the relationship between variables and does not conform to the actual physical laws. Moreover, the predicted aerosol pH in the sensitivity analysis was realistic, which confirms that the method we used was reasonable.

**e. Could a Monte Carlo method or other technique be used to make sure atmospherically relevant combinations of inputs are being used?**

**Response:** The Monte Carlo method is a good way to evaluate the uncertainty of the predicted aerosol pH and to determine if the input parameters are appropriate. However, as mentioned above, all input variables came from the actual observation to make sure the relationships between variables could conform to the actual physical laws. Moreover, the sensitivity analysis in this work focused on the variation tendency of aerosol pH rather the absolute aerosol pH value.

**3. Instead of classifying PM2.5 into clean (0-75 ug/m3), polluted, and heavily polluted (>150 ug/m3), it may be illustrative to consider PM2.5 in a continuum. 0-75 ug/m3 on a daily average is not very clean as it includes concentrations that exceed air quality standards. In addition, by considering PM2.5 concentrations as continuous, you may be able to better determine the association of pH with PM2.5. Consider that the pH for the three classifications is reported with a range/uncertainty that indicates the differences in pH between clean, polluted, and heavily polluted conditions are not statistically significant (values on line 262 overlap).**

**However, if considered as a continuous variable, a regression with confidence interval could be provided and might provide a more robust analysis of the association.**

**Response:** Thanks for your suggestion. Firstly, three groups for $PM_{2.5}$ were classified by hourly $PM_{2.5}$ mass concentration, not daily average $PM_{2.5}$ mass concentration. Secondly, the differences in pH between clean, polluted, and heavily polluted conditions were indeed not significant, the conclusion in the manuscript was just taken from the average value of pH. More deep analysis has been added in the revised manuscript.

Table 1 showed that as the air quality deteriorates, all aerosol components, as well as ALWC and $H_{air}^+$, increased, but the differences in pH between clean, polluted, and heavily polluted conditions are not statistically significant. The relationship between $PM_{2.5}$ and aerosol pH was shown in Figure S8, the aerosol pH under clean condition spanned 2~7 while the aerosol pH under polluted and heavily polluted conditions mostly concentrated in 3~5. Time series of mass fraction of $NO_3^-$, $SO_4^{2-}$, $NH_4^+$, $Cl^-$, and crustal ions ($Mg^{2+}$ and $Ca^{2+}$) in total ions, as well as pH in all four seasons, were showed in Figure 4. It can be seen that on clean days, high aerosol pH (>6) was generally companied by high mass fraction of crustal ions, while the relatively low aerosol pH (<3) was companied by high mass fraction of $SO_4^{2-}$ and low mass fraction of crustal ion, which was most obvious in summer (large part of aerosol pH with RH≤30% were excluded in spring and winter). On polluted and heavily polluted days, the aerosol chemical composition was similar, mainly dominated by $NO_3^-$, hence the differences of aerosol pH on polluted and heavily polluted days were small. Compared with the mass concentration of $PM_{2.5}$, the different aerosol chemical compositions may be the essence that drives aerosol acidity. The impact of aerosol compositions on aerosol pH is discussed in Section 3.4.

**Table 2.** The averaged ambient temperature and $\varepsilon(NH_4^+)$, $\varepsilon(NO_3^-)$, $\varepsilon(Cl^-)$ at different ambient RH levels in four seasons.

| | RH | T, °C | $\varepsilon(NH_4^+)$ | $\varepsilon(NO_3^-)$ | $\varepsilon(Cl^-)$ |
|---|---|---|---|---|---|
| | ≤ 30 % | 24.8 ± 3.7 | 0.17±0.14 | 0.84±0.12 | 0.67±0.24 |
| Spring | 30~60 % | 20.6 ± 3.8 | 0.25±0.14 | 0.91±0.06 | 0.82±0.16 |
| | >60 % | 15.8 ± 2.7 | 0.28±0.12 | 0.96±0.03 | 0.96±0.06 |

| | | | | | |
|---|---|---|---|---|---|
| | ⩽ 30 % | 5.4 ± 5.3 | 0.31±0.13 | 0.78±0.12 | 0.89±0.14 |
| Winter | 30~60 % | 1.0 ± 3.6 | 0.50±0.21 | 0.89±0.10 | 0.97±0.03 |
| | >60 % | -1.9 ± 2.1 | 0.60±0.20 | 0.96±0.03 | 0.99±0.01 |
| | ⩽ 30 % | 35.6± 0.4 | 0.06±0.02 | 0.35±0.20 | 0.39±0.17 |
| Summer | 30~60 % | 29.6 ± 4.2 | 0.17±0.11 | 0.65±0.23 | 0.43±0.16 |
| | >60 % | 25.2 ± 3.8 | 0.26±0.12 | 0.90±0.12 | 0.71±0.15 |
| | ⩽ 30 % | 21.7± 7.5 | 0.07±0.06 | 0.49±0.25 | 0.45±0.21 |
| Autumn | 30~60 % | 20.8± 6.3 | 0.21±0.14 | 0.82±0.19 | 0.67±0.21 |
| | >60 % | 14.9 ± 5.7 | 0.30±0.19 | 0.92±0.10 | 0.86±0.13 |

[Figure]

**Figure S8.** The relationship between $PM_{2.5}$ mass concentration and aerosol pH, the dots with RH≤30% were excluded.

[Figure]

**Figure 4.** Time series of mass fraction of $NO_3^-$, $SO_4^{2-}$, $NH_4^+$, $Cl^-$, crustal ions ($Mg^{2+}$, $Ca^{2+}$) in total ions as well as aerosol pH in all four seasons.

**4. Better connect the size-resolved measurements with the rest of the text. To what degree did the presence of coarse material drive ambient pH? Do figures 5-7 and the analysis regarding drives of pH only consider fine mode pH?**

**Response:** Thanks for your suggestion. The data set of MOUDI was obtained during 2013 and 2015, whereas the online ion data was obtained in 2016 and 2017. (1) The sensitivity analysis in this work aimed at the $PM_{2.5}$ (*ie* fine particles) since the $PM_{2.5}$ components in four seasons were available and has a high temporal resolution (1h). In addition, the data set has a wild range, covering different levels of haze events, making it suitable for sensitivity analysis. The MOUDI data were only utilized to determine the size-resolved aerosol pH. (2) In this work, the coarse mode aerosol acidity was generally neutral, which mainly attributed to the higher mass concentration of mineral materials in the coarse mode. The sensitivity analysis in this work showed that the aerosol pH increased approximately linearly with the elevated $Ca^{2+}$ in $PM_{2.5}$ (Figure S18). However, the impact of $Ca^{2+}$

has a limited impact on fine mode aerosol pH due to its low mass concentration in $PM_{2.5}$. Our previous paper showed that the mineral materials such as $Ca^{2+}$ and $Mg^{2+}$ mainly concentrated in the coarse mode (Figure R2, same data set with this work, Zhao et al, 2017; Su et al., 2018). We did some supplementary simulations under extreme cases that $Ca^{2+}$ and $Mg^{2+}$ are removed from the input files. The results showed that the presence of $Ca^{2+}$ and $Mg^{2+}$ in coarse mode has a crucial effect on aerosol pH (Figure S22), the difference of aerosol pH (with and without $Ca^{2+}$ and $Mg^{2+}$) for particles larger than 1 μm increased with the increasing particle size. The aerosol pH in coarse mode decreased by 4~6.5 unit when the $Ca^{2+}$ and $Mg^{2+}$ are removed.

[Figure]

**Figure R2**. Size distributions of the mass concentration for $Ca^{2+}$ and $Mg^{2+}$ in summer, winter, and fall. (Zhao et al, 2017; Su et al., 2018)

[Figure]

**Figure S22**. Size distributions of the aerosol pH with and without $Ca^{2+}$ and $Mg^{2+}$ in summer, winter, and autumn.

**Minor comments:**

1. Line 49. Instead of stating that aerosol acidity is "usually estimated" by the charge balance, I would indicate "sometimes" or "frequently," but not usually as many studies do use a thermodynamic model.

**Response:** Thank you for your good advice, "usually" has been changed to "frequently" in the revised manuscript

2. Line 52-55 wording indicates ion balance fails because acidity is estimated by aerosol water extract. This doesn't follow well as ion balance (e.g. difference between number of charge equivalent anions and cations) doesn't require extraction.

**Response:** Thank you for your correction, here we want to express that the simple ion balance cannot predict the hydronium ion concentration in the aerosol liquid water accurately. In the revised manuscript, this statement has been reworded.

3. Line 95: may want to indicate models "often" assume internal mixtures (but that is not a requirement).

**Response:** The sentences about this assumption were deleted in the revised manuscript.

4. Line 98-99: For this statement indicating nitrate is mainly in the fine mode, does that need to be qualified by indicating a location or time of year? Does fine mode nitrate generally exceed coarse nitrate?

**Response:** Thank you for your question. This statement about nitrate is mainly aimed at the aerosol composition in China. Many studies in China showed that the fine mode nitrate generally exceeds

coarse nitrate except for the dust days. In Beijing, the fine mode (≤2.5 μm) nitrate concentration at different polluted level was 3~5 times higher than that in coarse mode (2.5~10 μm) (Meier et al., 2009; Tian et al., 2014; Sun et al., 2014), and the same size distribution was found in southern cities of China on non-dust days (Pan et al., 2009; Wang et al., 2015; Ding et al., 2017). However, in dust days, the $PM_{10}$ concentration was much higher than that of $PM_{2.5}$, resulting in the elevated nitrates in coarse mode (Pan et al., 2009; Wang et al., 2015). In the revised manuscript, the statement was qualified.

5. Near line 155 and Figure 1: Spring shows a fairly persistent difference in the concentration of PM10 vs PM2.5. Two dust episodes are mentioned. With the exception of these two episodes, do you have a sense of what is contributing to the PM10-PM2.5 material? Late September also indicates an episode in which PM10 is elevated compared to PM2.5.

**Response:** The $PM_{2.5-10}$ was generally was regarded as coarse particles. On clean days, the crustal materials could account for more than 30% of the total $PM_{2.5–10}$. During the dust events, crustal materials could account for more than 60% of the coarse particles (Xu, 2010). However, during the severe haze events, $SO_4^{2-}$, $NO_3^-$, $NH_4^+$, OM, and EC also substantially accumulated in the coarse mode (Pan et al., 2009; Tian et al., 2014).

6. Line 190 indicates water uptake onto hydrophilic organics can be ignored unless the fraction of particle water due to organics is near 1 (100%). Water due to uptake on organics is presumably important even when it is not the sole contributor to particulate water. The threshold of 1 should be removed and perhaps a statement about the potential error incurred by ignoring ALWCo should be added.

**Response:** Thank you for your good suggestion. Surely part of organic species in particles such as water-soluble secondary organic carbon is hygroscopic, especially in ultrafine particles. In the revised manuscript, the threshold of 1 has been removed and a statement about the potential error incurred by ignoring ALWCo has been added as below.

7. Text on lines 235-238 seems misplaced or unnecessary.
**Response:** This paragraph has been deleted in the revised manuscript.

8. Line 277 highlights sulfate as a driving factor for pH. Sulfate peaked at night during the winter (Figure 4) when photochemical activity is lower. To what degree is the diurnal variation in sulfate driven by chemistry vs meteorology (e.g. planetary boundary layer depth)?

**Response:** The diurnal variation in sulfate was complex, especially during the severe haze episodes, where the rapid increase in mass concentration was mainly due to the accumulation induced by the unfavorable meteorological condition. Figure R2(a) and R2(b) showed that for most of the time, the mass fraction of $SO_4^{2-}$ in total ions has little variation when $SO_4^{2-}$ mass concentration increased largely, which could be regarded as the contribution of meteorology. However, at some moments in the nighttime (gray shadow in the figure), both mass concentration and mass fraction of $SO_4^{2-}$ showed a significant increase, which mainly attributed to the secondary reaction of $SO_2$. Overall, the mean $SO_4^{2-}$ fraction in total ions at night in winter was slightly higher than that in daytime (Figure R2(c)), but differences are not statistically significant. Hence the diurnal variation in sulfate was more driven by meteorology.

[Figure]

**Figure R2.** Time series of $SO_4^{2-}$ mass concentration (a) and $SO_4^{2-}$ mass fraction in total ions (b) as well as the diurnal variation of $SO_4^{2-}$ mass fraction in total ions in winter.

9. Line 284: Is the key difference between the US and Beijing more driven by the higher concentrations or the greater variability in concentrations?

**Response:** Thank you for your question. According to the record of literature (Guo et al., 2015), $H_{air}^+$ diurnal variation was less significant while the ALWC diurnal variation was significant, hence the diurnal pattern in pH was mainly driven by particle water dilution. However, in this work, we find that both $H_{air}^+$ and ALWC had significant diurnal variation, and the aerosol acidity variation agreed with well with sulfate. In the North China Plain, the $PM_{2.5}$ mass concentration has a wide variation range and the average value was high. For example, in winter, the $PM_{2.5}$ mass concentration in Beijing was several to dozens times higher than that in the US, which means there are more seeds in the limited water vapor, hence the dilution of aerosol liquid water to $H_{air}^+$ doesn't work at all, the diurnal variation of aerosol components was more important. Therefore, we think both the higher concentrations and the greater variability in concentrations have important effects on the difference between the US and Beijing.

10. Line 384:386 represents a simplified description of ammonia partitioning in which ammonia acts first to neutralize sulfate and then any leftover ammonia can react with nitrate to make ammonium nitrate. Perhaps the authors do not mean this so simply. Reword to reflect the semivolatile nature of ammonia and nitrate.

   **Response:** The statements here indeed have some problems. In the revised manuscript, we try to give the impact of $NH_4^T$ on aerosol pH with another explanation. Elevated $NH_4^T$ could reduce $H_{air}^+$ exponentially and slightly increase ALWC when the other input parameters were held constant, leading to the decrease of aerosol pH. As the $NH_4^T$ increases, $H_{air}^+$ are consumed swiftly during the dissolution of $NH_3$ as well as the further reaction with $SO_4^{2-}$, $NO_3^-$, and $Cl^-$. And the elevated $NH_4^T$ increases the $\varepsilon(NO_3^-)$ and $\varepsilon(Cl^-)$ when $NO_3^T$ and $Cl^T$ were fixed (Figure 10), which means the elevated $NH_4^T$ alter the gas-particle partition and shifts more $NO_3^T$ and $Cl^T$ into particle phase, and the deliquescence of additional nitrate and chloride increased ALWC slightly.

11. Line 388: Do the authors mean that aerosol would be fully neutralized except for the fact that ammonia is taken up into clouds and precipitation? Reword to reflect the buffering nature of

ammonia.

**Response:** We afraid that the reviewer misunderstood what we meant. Here we want to deliver that although the ammonia in the atmosphere is excessive, the other conditions are limited, the ALWC is one of them. Compared to the liquid water content in clouds and precipitation, ALWC is much lower, hence the dilution of aerosol liquid water to $H_{air}^+$ is much weaker. In the revised manuscript, we reword Line 376-388 to more clearly express our point.

12. Caption to table 2: This table appears to be the sensitivity of acidity, ALWC, and H+air to chemical components (not the other way around). Please clarify caption.

**Response:** Thanks for your careful check, the caption to Table 2 has been clarified as below:

13. Figure 3: use a common color scale for all panels.

**Response:** Color scale in figures has been unified.

14. Figure 5, 6, 7, caption. These figures appear to be the sensitivity of ALWC, H+air, and pH to chemical components. Reword caption.

**Response:** Thanks for your careful check, captions to Figure 5, 6, 7 (7-9 in the revised manuscript) have been clarified as below:

**Figure 7.** Sensitivities of $H_{air}^+$ to $SO_4^{2-}$, $NO_3^T$, $NH_4^T$, $Cl^T$, as well as meteorological parameters (RH, T) in summer and winter.

**Figure 8.** Sensitivities of ALWC to $SO_4^{2-}$, $NO_3^T$, $NH_4^T$, $Cl^T$, as well as meteorological parameters (RH, T) in summer and winter.

**Figure 9.** Sensitivities of aerosol pH to $SO_4^{2-}$, $NO_3^T$, $NH_4^T$, $Cl^T$, as well as meteorological parameters (RH, T) in summer and winter.

15. Line 136: Have you looked at trends from 2013, 2015, and 2017 datasets you have collected?

**Response:** In this work, the water-soluble ions of $PM_{2.5}$ samples and MOUDI samples were not collected synchronously. Water-soluble ions ($SO_4^{2-}$, $NO_3^-$, $Cl^-$, $NH_4^+$, $Na^+$, $K^+$, $Mg^{2+}$, $Ca^{2+}$) of $PM_{2.5}$ and trace gases (HCl, $HNO_3$, $HNO_2$, $SO_2$, $NH_3$) in the ambient air were measured by an online analyzer (MARGA) at hourly temporal resolution during the spring (April and May in 2016), winter (February in 2017), summer (July and August in 2017) and autumn (September and October in 2017). While the size-resolved sampling was conducted during July 12-18, 2013; January 13-19,

2014; July 3-5, 2014; October 9-20, 2014; and January 26-28, 2015. Compared to the real-time $PM_{2.5}$ sampling, MOUDI sampling time is short, which is not conducive to analyze the variation tendency of aerosol composition and acidity in time. MOUDI samples were mainly used to analyze the change of aerosol composition and acidity in different particle size.

16. Additional improvements in terms of editing would be useful.

**Response:** The English in the manuscript has been improved by an English native speaker.

**References**

Cruz, C. N., Dassios, K. G., Pandis, S. N.: The effect of dioctyl phthalate films on the ammonium nitrate aerosol evaporation rate. Atmos. Environ., 34, 3897-3905, 2000.

Dassios, K. G., Pandis, S. N.: The mass accommodation coefficient of ammonium nitrate aerosol. Atmos. Environ., 33 (18), 2993-3003, 1999.

Dai, Q. L., Bi, X. H., Liu, B. S., Li, L. W., Ding, J., Song, W. B., Bi, S. Y., Schulze, B. C., Song, C.B., Wu, J. H., Zhang, Y. F., Feng, Y. C, Hopke, P. K.: Chemical nature of PM2.5 and PM10 in Xi'an, China: Insights into primary emissions and secondary particle formation. Environ. Pollu., 240 155-166, 2018, doi:10.1016/j.envpol.2018.04.111.

Ding, X. X., Kong, L. D., Du, C. T., Zhanzakova, A., Fu, H. B., Tang, X. F., Wang, L., Yang, X., Chen, J. M., Cheng, T. T.: Characteristics of size-resolved atmospheric inorganic and carbonaceous aerosols in urban Shanghai. Atmos. Environ., 167, 625-641, 2017, doi: 10.1016/j.atmosenv.2017.08.043

Fang, T., Guo, H. Y., Zeng, L. H., Verma, V., Nenes, A., Weber, R. J.: Highly acidic ambient particles, soluble metals, and oxidative potential: A link between sulfate and aerosol toxicity, Environ. Sci. Technol., 51, 2611-2620, 2017.

Gao, J. J., Tian, H. Z., Cheng, K., Lu, L., Zheng, M., Wang, S. X., Wang, K.: The variation of chemical characteristics of $PM_{2.5}$ and $PM_{10}$ and formation causes during two haze pollution events in urban Beijing, China. Atmos. Environ. 107,1-8, 2015.

Guo, H., Xu, L., Bougiatioti, A., Cerully, K. M., Capps, S. L., Hite Jr., J. R., Carlton, A. G., Lee, S.-H., Bergin, M. H., Ng, N. L., Nenes, A., Weber, R. J.: Fine-particle water and pH in the

southeastern United States, Atmos. Chem. Phys., 15, 5211-5228, 2015.

Guo, H., Sullivan, A. P., Campuzano-Jost, P., Schroder, J. C., Lopez-Hilfiker, F. D., Dibb, J. E., Jimenez, J. L., Thornton, J. A., Brown, S. S., Nenes, A., Weber, R. J.: Fine particle pH and the partitioning of nitric acid during winter in the northeastern United States, J. Geophys. Res. Atmos., 121, 10355-10376, 2016.

Meier, J., Wehner, B., Massling, A., Birmili, W., Nowak, A., Gnauk, T., Brüggemann, E., Herrmann, H., Min, H., Wiedensohler, A.: Hygroscopic growth of urban aerosol particles in Beijing (China) during wintertime: a comparison of three experimental methods, Atmos. Chem. Phys., 9, 6865–6880, 2009.

Pan, X. L., Yan, P., Tang, J., Ma, J. Z., Wang, Z. F., Gbaguidi, A., and Sun, Y. L.: Observational study of influence of aerosol hygroscopic growth on scattering coefficient over rural area near Beijing mega-city, Atmos. Chem. Phys., 9, 7519-7530, 2009.

Su, J., Zhao, P. S., Dong, Q.: Chemical Compositions and Liquid Water Content of Size-Resolved Aerosol in Beijing. Aerosol Air Qual. Res., 18, 680-692, 2018.

Sun, K., Qu, Y., Wu, Q., Han, T. T., Gu, J. W., Zhao, J. J., Sun, Y. L., Jiang, Q., Gao, Z. Q., Hu, M., Zhang, Y. H., Lu, K. D., Nordmann, S., Cheng, Y. F., Hou, L., Ge, H., Furuuchi, M., Hata, M., Liu X. G.: Chemical characteristics of size-resolved aerosols in winter in Beijing, J. Environ. Sci., 26,1641-1650, 2014, doi: 10.1016/j.jes.2014.06.004.

Tian, S. L., Pan, Y. P., Liu, Z.R., Wen, T. X., Wang, Y. S.: Size-resolved aerosol chemical analysis of extreme haze pollution events during early 2013 in urban Beijing, China. J. Hazard. Mater., 279, 452-460, 2014.

Wang, H.L., Zhu, B., Shen, L. J., Xu, H. H., An, J. L., Xue, G. Q., Cao, J. F.: Water-soluble ions in atmospheric aerosols measured in five sites in the Yangtze River Delta, China: Size-fractionated, seasonal variations and sources. Atmos. Environ., 123, 370-379,2015, doi: 10.1016/j.atmosenv.2015.05.070

Xu, C.: Characteristics,source and the formation mechanism of aerosol in mega-city, China [D], Fudan University, Shanghai, 2010.

Zhao, P. S., Chen, Y. N., Su, J.: Size-resolved carbonaceous components and water-soluble ions measurements of ambient aerosol in Beijing. J. Environ. Sci., 54, 298-313, 2017.

---

## Author Response (AR2)

Dear Editor,

We are truly grateful for your and other reviewers' second-round comments, which are very helpful for us to highlight our work. Substantial changes were made in this version of manuscript, and most sections of this manuscript were re-examined and reorganized. The revision was mainly aimed at the language editing, the reorganization of key points in discussions, and the refinement of conclusions.

1) We refined the **Abstract** and **Conclusions** to highlight the key points. And the language and figures are re-edited as recommended.

2) We re-adjusted the structure of **Introduction** to make the logic and purpose of this work more clear.

3) In Section 3.4 and 3.6 in the revised manuscript, we simplified the discussions of sensitivity tests, focusing on the factors affecting $PM_{2.5}$ pH and gas-particle partitioning, which is helpful to understand the driving factors of aerosol acidity in the North China Plain and provide the idea of controlling nitrate in the particles.

4) We seriously revised the parts of the paper that were not clear enough and not necessary. In addition, we asked a professional English editing website to revise our paper. The certificate is attached at the end of this document.

Thank you very much for your concerning.

Best regards.

Sincerely yours,

Pusheng Zhao & Jing Ding

**Anonymous Referee #1**

*Substantial changes were made to the first draft of this paper based on the comments from the reviewers. The paper still has substantial problems. First, the analysis is largely not novel; the paper seems to essentially copy the work of published papers, where the only main difference is the work was done in a different location. I suggest the authors try to add more insight to their work. Second, the paper is hard to follow and understand. The language usage and grammar is very poor; the paper needs substantial editing. The figures largely do not make sense with multiple types of plots on the same figure and no explanation in the figure caption. Third, many of the explanations for the observed sensitivities do not make sense, or are not explained in a logical way. Much of this is new text added after the first round of review. The authors might want to explain why they discuss sensitivity of Hair+ (i.e., why is Hair+ important). As the sensitivity analysis section is largely very difficult to follow, the authors may wish to completely remove it from the paper. Instead the focus could be on the bulk predicted pH when both gas and particle MARGA data are available (including it's validation, issues with RH, etc) and the MOUDI size-resolved pH.*

**Response:** We would like to express our gratitude for your comments, which are very important to help us highlight our work. In this work, the thermodynamic model ISORROPIA-II was utilized to predict aerosol pH in Beijing based on a long-term online high-temporal resolution dataset and a size-resolved offline dataset. Additionally, a sensitivity analysis was conducted to identify the key factors affecting aerosol pH and gas-particle partitioning. The main purposes of this work are to 1) obtain the PM2.5 pH level based on long-term online aerosol samples, contributing towards a global pH dataset; 2) investigate the size-resolved aerosol pH, providing useful information for understanding the formation processes of secondary aerosols; and 3) explore the main factors affecting aerosol pH and gas-particle partitioning, which can help explain the possible reasons for pH divergence in different works and provide a basis for controlling secondary aerosol generation.

As you suggested, substantial changes were made in this version of manuscript, we simplified the paper and summarized the key points of our work, including:

**1)** In 2016-2017, the mean $PM_{2.5}$ pH (at RH > 30%) over four seasons was 4.5±0.7 (winter) > 4.4±1.2 (spring) > 4.3±0.8 (autumn) > 3.8±1.2 (summer), showing moderate acidity. According to the size-resolved aerosol pH, the particles in coarse mode were neutral in most cases. However, on heavily polluted days, more secondary ions accumulated on the coarse particles, leading to a change in the acidity of the coarse particles from neutral to weakly acidic. Sensitivity tests demonstrated $Ca^{2+}$ and $Mg^{2+}$ played an important role in aerosol pH.

**2)** In the North China Plain (NCP), the common driving factors affecting $PM_{2.5}$ pH variation in all four seasons were $SO_4^{2-}$, $TNH_3$ (total ammonium (gas+aerosol)), and temperature, while the unique factors were $Ca^{2+}$ in spring and RH in summer. Elevated $SO_4^{2-}$ levels can enhance aerosol acidity due to the stronger ability of $SO_4^{2-}$ to provide hydrogen ions. The decreasing $SO_4^{2-}$ and increasing $NO_3^-$ mass fractions in $PM_{2.5}$ as well as excessive $NH_3$ in the atmosphere in the NCP in recent years are the reasons why aerosol acidity in China is lower than that in Europe and the United States. The nonlinear relationship between $PM_{2.5}$ pH and $TNH_3$ indicated that although $NH_3$ in the NCP was abundant, the $PM_{2.5}$ pH was still acidic.

**3)** Gas-particle partitioning sensitivity tests revealed that the typical high RH values and low temperatures during haze events in the NCP are conducive to the formation of secondary particles. Given that ammonia was excessive in most cases, a decrease in nitrate would occur only if $TNH_3$ were greatly reduced. Therefore, in terms of controlling the generation of nitrate, a reduction in NOx emissions is more feasible than a reduction in $NH_3$ emissions.

In brief, the revision is mainly aimed at the language editing, the reorganization of key points in discussions, and the refinement of conclusions.

1) We refined the **Abstract** and **Conclusions** to highlight the key points. And the language and figures are re-edited as recommended.

2) We re-adjusted the structure of **Introduction** to make the logic and purpose of this work more clear.

3) After careful consideration, we still believe that the sensitivity tests are important for understanding the causes of pH changes. In the revised manuscript, we rewrote this part and simplified the discussions of sensitivity tests, mainly focusing on the factors affecting PM$_{2.5}$ pH and gas-particle partitioning. Please see section 3.4 and 3.6.

4) We seriously revised the parts of the paper that were not clear enough and not necessary.

**Specific Comments.**

*Lines 226 to 228: Provide numbers to support the statement that ALWC could be off in regions of high OA fractions. That is, give some idea how high the OA fraction would need to be for it to matter. Published typical hygroscopicity parameters for OA could be assumed.*

**Response:** Thanks for your important comment. In our manuscript, we indeed notice that both inorganic and part of organic species in particles are hygroscopic. According to the literatures, the organic matter-induced aerosol water in some studies conducted in China could be negligible compared to the inorganic matter-induced particle water (Guo et al., 2015, 2016; Liu et al., 2017). In the southeastern United States, a large fraction of the PM2.5 (~70 %) was organic matter, and the corresponding ALWCo is on average 29% to 39 % of total aerosol water, PM$_{2.5}$ pH increased by 0.15 to 0.23 units when ALWCo is included. In the North China Plain, particularly in recent years, the fraction of organic matter was 20%~25% in PM$_{2.5}$, which is much lower than that in southeastern United States. In contrast, more than 50% of PM2.5 are inorganic ions in the North China Plain (Huang et al., 2017; Zhang et al., 2018; Zhang et al., 2019). The results in Liu et al., (2017) showed that the mass fraction of organic matter-induced particle water accounted for only 5% of total ALWC, indicative of a negligible contribution to aerosol acidity. Hence, the aerosol pH can be fairly predicted by ISORROPIA-II with measurements of inorganic species in most cases.

*Line 258 to 260. Explain how gas denuder artifacts would result in the model greatly over-predicting HNO3 or HCl. Artifacts associated with particles deposited in the denuder would seem to result in measured values larger than predicted, opposite what is shown.*

**Response:** The precision and accuracy performance of MARGA was assessed by the US EPA (Rumsey et al., 2014). Precision of MARGA was evaluated by calculating the median absolute relative percent difference between paired hourly results from duplicate MARGA units. The accuracy of the MARGA was evaluated by calculating the median absolute relative percent difference for each MARGA unit relative to the average of the duplicate denuder/filter pack concentration. The results demonstrated that the MARGA performed moderately well in measuring $HNO_3$ and $NH_3$. The measured $HNO_3$ and $NH_3$ by MARGA were lower than the denuder concentrations. The performance of the MARGA in measuring $HNO_3$ and $NH_3$ was likely influenced by the adsorption of $HNO_3$ and $NH_3$ onto the sampling tubing and inlet since the $HNO_3$ and $NH_3$ are all "sticky" gases, which may also apply to HCl. Thus, it is reasonable that the measured values of gas phase $HNO_3$ were lower than the predicted values in the results of this study.

[Figure]

[Figure]

**Figure 12.** Regression analysis of MARGA $HNO_3$ concentrations against denuder $HNO_3$ concentrations.

**Figure 22.** Regression analysis of MARGA $NH_3$ concentrations against denuder $NH_3$ concentrations.

**Response:** Thanks for your advice. We checked all the problems of figures you pointed out. In the revised manuscript, the figure captions are clear to understand.

*There are many issues with explanations from the sensitivity tests. Overall, the discussion is just a laundry list of how things vary with season. What is the point to this*

*discussion? The manuscript would be greatly improved if it be simplified or somehow focused more. The discussion of RH, H+ and LWC is very confusing and simplistic; details in the logic are missing. I do not understand the T discussion. Basically all the added text in the second version of the paper is hard to follow.*

**Response:** Thanks for your comments to improve our work. The discussion about factors affecting ALWC, $H_{air}^+$, $PM_{2.5}$ pH, and gas-particle partitioning (Section 3.4 in the manuscript) are simplified, more focusing on the factors affecting $PM_{2.5}$ pH, and gas-particle partitioning, which helps to understand the role of aerosol acidity in secondary particle formation. For example, Cheng et al. (2016) and Wang et al. (2016) proposed that $SO_2$ could oxidized by $NO_2$ to form sulfate, whereby high reaction rates are sustained by the high neutralizing capacity of the atmosphere in northern China. However, many studies show that the aerosol pH in North China Plain is moderately acidic (Liu et al., 2017; Shi et al., 2017; Tan et al., 2018), which means the new pathways for sulfate production in China proposed by Cheng et al. (2016) and Wang et al. (2016) should be revisited. Therefore, the sensitivity analysis is aimed to identify the crucial factor affecting aerosol pH and gas-particle portioning, which may explain the differences of aerosol acidity level of these studies. Moreover, the discussion of gas-particle portioning helps to provide an idea on controlling the secondary aerosol formation.

**References**

[revised manuscript text omitted]
 | RH | $SO_4^{2-}$ | T, °C | TNO₃ | $\varepsilon(NH_4^+)$ | TNH₃ | $\varepsilon(NO_3^-)$ | $Ca^{2+}$ | $\varepsilon(Cl^-)$ | RH | T |
|---|---|---|---|---|---|---|---|---|---|---|---|
| Spring-RSD | ≤30 12.4% | | 24.8±5.2% | 7.5% | 0.17±0.14 3.79% | 1.3% | 0.84±0.12 | 7.0% | 0.67±0.24 % | | |
| | 30-60% | | 20.6±3.8 | | 0.25±0.14 | | 0.91±0.06 | | 0.82±0.16 | | |
| Winter-RSD | >60 28.1% | | 15.8±2.7.4% | 27.0% | 0.28±0.12 | 1.0% | 0.96±0.03 | | 0.96±0.06 4.1% | 6.7% | |
| | ≤30 % | | 5.4±5.3 | | 0.31±0.13 | | 0.78±0.12 | | 0.89±0.14 | | |
| Winter | 30-60 % | | 1.0±3.6 | | 0.50±0.21 | | 0.89±0.10 | | 0.97±0.03 | | |
| | >60 % | | -1.9±2.1 | | 0.60±0.20 | | 0.96±0.03 | | 0.99±0.01 | | |

| | | | | | | | |
|---|---|---|---|---|---|---|---|
| Summer- RSD | ≤ 30 7.9% | 35 3.6± 0.4% | 0.06±0.02 % | 28.1 % | 0.35±0.20 % | 1.9 % | 0.39±0.17 8.6 % | 5.8 % |
| | 30-60 % | 29.6 ± 4.2 | 0.17±0.11 | 0.65±0.23 | 0.43±0.16 | | |
| | >60 % | 25.2 ± 3.8 | 0.26±0.12 | 0.90±0.12 | 0.71±0.15 | | |
| Autumn - RSD | ≤ 30 6.0% | 3.3 % | 16.1 % | 0.8 % | 2.4 % | 21.7± 7.5% | 0.07±0.0 6 | 0.49±0.2 5 | 0.45±0.2 1 |
| | 30-60 % | 20.8± 6.3 | 0.21±0.14 | 0.82±0.19 | 0.67±0.21 | | |
| | >60 % | 14.9 ± 5.7 | 0.30±0.19 | 0.92±0.10 | 0.86±0.13 | | |

**Table 3**

删除的单元格
删除的单元格
删除的单元格
删除的单元格
删除的单元格
删除的单元格
删除的单元格
删除的单元格

Impact Factor    SO$_4^{2-}$ RH    T    NO$_3^-$(NO$_3$-C) NH$_4^+$(NH$_4$-O) Ca$^{2+}$ ε(Cl$^-$)    RH    T

| | | | NO$_3^-$(NO$_3^-$C) | NH$_4^+$(NH$_4$-O$^{3-}$) | | |
|---|---|---|---|---|---|---|
| | 50.5% | 53.4% | 2.9% | 0 | 0 | |
| RSD ALW ≤ 30% | | | | 24.8±3.7±5% | 18.4±0.42±2% | 13.1% 0.67±0.24 |
| Spring | | | | | | |
| | 223% | 34.4% | 26.20±3.8% | 14.9.80±5% | 49.5% |
| RSD H$_{air}$ +30~60% | | | | 115% 0.82±0.16 | | |

12.4%  5.2%  3.9%

RSD  pH>60%

7.0%  0.96±0.06

W
i
n
t
e
r

RSD  AL  WC

≤ 30.7%

0.89±0.14

431%  431%  187.4%

RSD  H 30~60%

0.97±0.03  74.1%

删除的单元格
删除的单元格
删除的单元格
删除的单元格
删除的单元格
删除的单元格
删除的单元格
删除的单元格
插入的单元格
插入的单元格
插入的单元格
删除的单元格
删除的单元格
删除的单元格

[Figure]

[Figure]

RSD pH>60%

删除的单元格

删除的单元格

删除的单元格

**Figure captions**

**Figure 1.** Time series of relative humidity (RH and temperature (T) (a, e, i, m); $PM_{2.5}$, $PM_{10}$, and $NH_3$ (b, f, g, n); dominant water-soluble ions: $NO_3^-$, $SO_4^{2-}$, and $NH_4^+$ (c, g, k, o); and $PM_{2.5}$ pH coloured by $PM_{2.5}$ concentration (d, h, l, p) over four seasons.

**Figure 2.** Comparisons of predicted and measured $NH_3$, $HNO_3$, $HCl$, $NH_4^+$, $NO_3^-$, $Cl^-$, $\varepsilon(NH_4^+)$, $\varepsilon(NO_3^-)$, and $\varepsilon(Cl^-)$ coloured by RH. In this figure, the data from all four seasons were combined; comparisons of individual seasons are shown in Figure S1-S4.

**Figure 3.** Comparisons of predicted and iterative $NH_3$, $HNO_3$, and $HCl$, as well as predicted and measured $NH_4^+$, $NO_3^-$, $Cl^-$, $\varepsilon(NH_4^+)$, $\varepsilon(NO_3^-)$, and $\varepsilon(Cl^-)$ coloured by particle size. In this figure, all MOUDI data were combined.

**Figure 4.** Time series of mass fractions of $NO_3^-$, $SO_4^{2-}$, $NH_4^+$, $Cl^-$, $Mg^{2+}$, $Ca^{2+}$ $^{2+}$ with respect to the total sion content, as well as $PM_{2.5}$ pH in all four seasons. ($PM_{2.5}$ pH values at RH≤30% were excluded).

**Figure 5.** Winddependence map of $PM_{2.5}$ pH over four seasons. In each picture, the shaded contour indicates the mean value of $PM_{2.5}$ pH for varying wind speeds (radial direction) and wind directions (transverse direction) (data at RH≤30% were excluded).

**Figure 6.** Diurnal patterns of mass concentrations of $NO_3^-$ and $SO_4^{2-}$ in $PM_{2.5}$, predicted aerosol liquid water content (ALWC), $H_{air}^+$, and $PM_{2.5}$ pH over four seasons. Mean and median values are shown, together with 25% and 75% quantiles. Data at RH≤30% were excluded, and the shaded area represents the time period when most RH  lower than 30%.

**Figure 7.** Sensitivity tests of $PM_{2.5}$ pH to $SO_4^{2-}$, $TNO_3$, $TNH_3$, $Ca^{2+}$, and meteorological parameters (RH and T) in summer (S) and winter (W).

**Figure 8.**

**Figure 10.** Sensitivity tests of $\varepsilon(NH_4^+)$, $\varepsilon(NO_3$ to $TNO_3$, $TNH_3$, RH and T  coloured by $PM_{2.5}$ pH in summer (S) and winter (W).

**Figure 9.** Size distributions of aerosol pH and all analysed chemical components under clean (a, d, g), polluted (b, e, h), and heavily polluted conditions (c, f, i) in summer, autumn, and winter.

[Figure]

[Figure]

**Figure 1**

[Figure]

[Figure]

**Figure 2.**

[Figure]

[Figure]

**Figure 3.**

[Figure]

**Figure 4.**

[Figure]

**Figure 5.**

[Figure]

**Figure 6.**

[Figure]

[Figure]

Figure 7.

[Figure]

[Figure]

[Figure]

**Figure 8.**

[Figure]

[Figure]

**Figure 9.**

[Figure]

**Figure 10.**

[Figure]

**Figure 11.**

[Figure]

**Figure 12.**

**AMERICAN JOURNAL EXPERTS**

**EDITORIAL CERTIFICATE**

This document certifies that the manuscript listed below was edited for proper English language, grammar, punctuation, spelling, and overall style by one or more of the highly qualified native English speaking editors at American Journal Experts.

**Manuscript title:**

Aerosol pH and its driving factors in Beijing

**Authors:**

Jing Ding, Pusheng Zhao, Jie Su, Qun Dong, Xiang Du, and Yufen Zhang

**Date Issued:**

December 26, 2018

**Certificate Verification Key:**

4698-1920-7568-EC60-F7BP

[Figure]

This certificate may be verified at www.aje.com/certificate. This document certifies that the manuscript listed above was edited for proper English language, grammar, punctuation, spelling, and overall style by one or more of the highly qualified native English speaking editors at American Journal Experts. Neither the research content nor the authors' intentions were altered in any way during the editing process. Documents receiving this certification should be English-ready for publication; however, the author has the ability to accept or reject our suggestions and changes. To verify the final AJE edited version, please visit our verification page. If you have any questions or concerns about this edited document, please contact American Journal Experts at support@aje.com.

American Journal Experts provides a range of editing, translation and manuscript services for researchers and publishers around the world. Our top-quality PhD editors are all native English speakers from America's top universities. Our editors come from nearly every research field and possess the highest qualifications to edit research manuscripts written by non-native English speakers. For more information about our company, services and partner discounts, please visit www.aje.com.

---

## Author Response (AR3)

**Dear professor Athanasios Nenes:**

**We're appreciated for your valuable comments, which are very useful for improving the quality of this manuscript. Your comments help us better understand the thermodynamic equilibrium between aerosol and gas phase, we learned a lot from these detailed comments. Our responses to the specific points are given below.**

Comments

1. Page 1, line 19: "due to the stronger ability…ions". Affinity with the H+ is not the underlying reason (otherwise, increasing HNO3 would also drive acidity up, and it doesn't really). The very low volatility of SO4 compared to the neutralizing cations (mainly NH4) is the reason for the strong acidity associated with SO4 in aerosol. NH4 evaporates to NH3 in achieving equilibrium, and that by nature creates an ion imbalance that leads to H+ production through dissociation of H2O. The appropriate reference for this is Weber et al., 2016.

**Response: The statement has been removed from the Abstract.**

2. Page 1, line 24: "hydrolysis of … and ALWC". The authors mention hydrolysis of ammonium salts throughout the manuscript, but do not provide any calculations to support this. Even a pure ammonium sulfate particle, when deliquesced, will evaporate some NH4 to the gas phase to produce NH3 – this is the reason for acidity, as mentioned above. LWC variability does lead to a pH unit change for typical variations throughout the day (and this has also been shown before by the Guo et al. studies and others).

**Response: The statement "which might be attributed to the limited aerosol liquid water content (ALWC) and hydrolysis of ammonium salts" has been removed from the Abstract, the reasons why aerosol is acidic is interpreted by the thermodynamic equilibrium considerations between aerosol phase and gas phase.**

3. Page 3, line 40. "In addition…acid rain". Incorrect statement. Aerosol acidity is decoupled from acid rain pH. The reason being that the water per kg of aerosol "mass" is fixed by the RH – so pH changes are relatively insensitive to changes in absolute aerosol mass, while rain water is decoupled from the aerosol mass – so rainwater pH changes considerably with different aerosol loading. Otherwise the acid rain program would have failed in the US – because aerosol acidity has not gone down over time (e.g., Weber et al., 2016).

**Response: According to the theory of 'Greenfield gap'(Greenfield, 1957), the collision efficiency of raindrop on particle between 0.2 and 2.0 μm is very low. It means only the coarse particle can possibly be scavenged by wet precipitation. Furthermore, the concentration of**

**coarse particle is not high enough to affect the acidity of rain in urban area. The statement has been removed from the revised manuscript. (Page 3, line 34, in the revised manuscript)**

Greenfield S M.: Rain scavenging of radioactive particulate matter from the atmosphere. Journal of the Meteorology Sciences, 1957, 14: 115-125.

4. Page 3, line 44. "A net … alkalinity". Hennigan et al., and others have shown that the ion balance derived from observations works when all the ions (even trace ones) are well constrained and the dissociation state of multivalent ions are known. This of course cannot be satisfied for aerosol, so the statement must be erased. The studies quoted (especially Wang et al.) based most of their discussion on ion balances, and therefore are not well-supported.

**Response: The statement has been removed from the revised manuscript. The revised statement is "Nevertheless, not all ions (even trace ones) are well constrained in the observations and the dissociation state of multivalent ions are unclear, ion balance and other similar proxies fail to represent the in situ aerosol pH because such metrics cannot accurately predict the $H^+$ concentration in the aerosol liquid phase (Guo et al., 2015; Hennigan et al., 2015)." (Page 3, line 37-38, in the revised manuscript)**

5. Page 3, line 66. "NCP showed". These studies have showed to contain important issues (e.g., Song et al., ACP, 2018; Guo et al., 2017) that does not make the neutral pH inferences likely. I agree that the mildly acidic pH are quite likely, and that it is higher than the pH levels found in other locations, so please modify the sentence accordingly, perhaps removing the references to neutral pH levels.

**Response: The statement and references to neutral pH levels has been removed from the revised manuscript. (Page 3, line 58, in the revised manuscript)**

6. Page 4, line 66. "particulate matter concentration is very low". pH variations from diurnal variability in RH is always occurring, because the LWC per kg aerosol mass changes drastically with RH. Therefore please remove the "In some countries…very low".

**Response: pH diurnal variation is not driven by the particulate matter concentration. The statement has been removed from the revised manuscript. (Page 3, line 64, in the revised manuscript)**

7. Page 4, line 79. "size-resolved pH are still rare". That is true, but some studies should be cited here that have done this work.

**Response: The works published by Fang et al. (2017) and Craig et al. (2018) have been added in the revised manuscript. (Page 4, line 69, in the revised manuscript)**

Fang, T., Guo, H. Y., Zeng, L. H., Verma, V., Nenes, A., Weber, R. J.: Highly acidic ambient particles, soluble metals, and oxidative potential: A link between sulfate and aerosol toxicity, Environ. Sci.

Technol., 51, 2611-2620, 2017.

Craig, R. L., Peterson, P. K., Nandy, L., Lei, Z., Hossain, M. A., Camarena, S., Dodson, R. A., Cook, R. D., Dutcher, C. S., and Ault, A. P.: Direct determination of aerosol pH: size-Resolved measurements of submicrometer and supermicrometer aqueous particles, Anal Chem, 90, 11232-11239, 10.1021/acs.analchem.8b00586, 2018.

8. Page 5, line 128. You do not have NH3, HNO3 and HCl concentrations. How do you address this when calculating the pH? It is good to make sure people understand this issue.

**Response: The state "Gas precursors were not observed during the periods of MOUDI sampling" has been removed from the Section 2.3 in the revised manuscript. The complete method of size-resolved aerosol pH calculation is showed in Section 2.4. (Page 5, line 118, in the revised manuscript)**

9. Page 6, line 161-163. Seinfeld and Pandis is a good reference, but it is not clear from there why the DRH is low enough in the NCP to assume metastability. Metastability is supported by the RH history of the particles, and their composition (the likelihood of being in the efflorescence or deliquescence branch of the water uptake curves). I defer to the Song et al., 2018 manuscript and all the discussion in the ACPD form of the manuscript, to see what are the arguments that supports metastability.

**Response: As Song et al. (2018) mentioned in their work, there were no observational evidence so far to suggest whether the Beijing winter haze fine particles were in a metastable or stable state. We have referenced some literature and think that the assumption of metastable state is overall reasonable. In the revised manuscript, we try to provide more evidence to support this assumption.**

**In the ambient atmosphere, the aerosol chemical composition is complicated; hence, the deliquescence relative humidity (DRH) of aerosols is generally low (Seinfeld and Pandis, 2016). Once the particles are deliquescent, crystallization only occurs at a very low RH, which is called hysteresis phenomenon. The efflorescence RH (ERH) of a salt cannot be calculated from thermodynamic principles; rather, it must be measured in the laboratory. For a particle consisting of approximately 1:1 $(NH_4)_2SO_4 : NH_4NO_3$, the ERH is around 20%, while for a 1:2 molar ratio it decreases to around 10%. (Shaw and Rood 1990). Recently, $NO_3^-$ dominates the particles in the NCP (Zhao et al., 2013, 2017; Huang et al., 2017; Ma et al., 2017); therefore, we assumed that the particles are in a liquid state (metastable condition). Assumption that particles are in metastable were adopted by numerous studies in the NCP (Liu et al., 2017; Guo et al., 2017; Shi et al., 2017, 2019). (Page 6, line 148-160, in the revised manuscript)**

10. Page 6, line 177-182. I'm quite surprised that the partitioning is sensitive to the phase state. What version of ISORROPIA do you use? Are you sure you use the latest version of the code (2.3) with the latest bug fixes? If not, I can provide a copy of the code upon request. The conclusion that pH inferences when the RH is low is also supported by other studies – so it would be good to cite those. Guo et al. JGR (2017) suggest that the lower limit is about 40%, why do you think it's different from the 30% cited in this study?

**Response: We have asked Mr. Song and Mr. Shi for the ISORROPIA V2.3 (Win) and recalculated the PM$_{2.5}$ pH at RH≤30 in stable state. We found that when RH was low, the partitioning was still sensitive to the phase state. Because the subcases O7 and P13 are mostly used when the forward metastable mode simulations are performed, while subcases O1 and P1 are mostly used when forward stable mode simulations are performed. Even if Mr. Song revised the errors in the standard ISORROPIA-II for the four subcases (G1, G2, O1, and O2), there are still differences in gas-particle portioning between forward metastable mode and forward stable mode when the RH is low. However, as you mentioned in the comments to Song et al. (2018), when RH is low, the liquid water content becomes very small, PM$_{2.5}$ pH is subject to considerably more uncertain. Therefore, we removed the statement about comparison between stable and metastable, and cited the work of Guo et al. (2017). In this work, we finally set the lower RH limit as 30% due to the overall good agreement between predictions and measurements when RH was high than 30%. (Page 6, line 173-177, in the revised manuscript)**

11. Page 7, line 189-191. But you have gas-phase concentration of semi-volatile species with the MARGA. Why not use those? It is much better than the unconstrained iterative procedure of Guo et al. which really works when you have an "idea" of the expected NH3 (or other) levels. This is a major issue of the paper – that it doesn't seem to constrain the size-resolved pH well because the gas-phase is not constrained well enough.

**Response: In the revised manuscript, the gaseous precursor measured by MARGA are used to calculated the size-resolved aerosol pH. Averaged NH$_3$, HNO$_3$ and HCl measured by MARGA matched to PM$_{2.5}$ mass concentration levels during the MOUDI sampling periods were input. And the gaseous precursor measured by MARGA are also used to recalculated coarse mode aerosol pH. We are pleased to find that the measured and predicted NO$_3^-$, NH$_4^+$, and Cl$^-$ agreed very well in fine-mode particles. The related results about size-resolved aerosol pH have been revised. The overall size distribution of aerosol pH does not change. (Page 7, line 181-195, in the revised manuscript)**

[Figure]

[Figure]

12. Page 7, line 208-209. Gas-particle disequilibrium does not assume that you can neglect the gas-phase! Either remove this section, or justify why you can assume this. If you cannot, then you have to revise the section (and related calculations) to accommodate for this.

**Response: Related statement has been removed. In the revised manuscript, the gaseous precursor measured by MARGA are also used to calculated coarse mode aerosol pH. (Page 7, line 193, in the revised manuscript)**

13. Page 7, line 215. "Nh4 and Ca2". True, but Ca also associated with SO4 and makes insoluble CaSO4 which can strongly depress the amount of soluble materials (hence LWC). You really need to include the hygroscopic ions too (K, Na, Mg).

**Response: We're afraid there is a misunderstanding, in the sensitivity analysis, K, Na, Mg, and total chloride are also included in the input files, and we have clarified the statement in the revised manuscript. Here we would like to emphasize the impact of $Ca^{2+}$ on aerosol pH, because Beijing is in North China, vegetation coverage is less than that of Southern China, hence the dust is an important source of particles. (Page 7, line 203-204, in the revised manuscript)**

14. Page 8, line 252. ISORROPIA issues an error message whenever there is "too much" Ca, Na, etc. If this is the case, then you need to say this – and basically say there is unneutralized carbonates in the aerosol. In general, these cases are characterized by external mixing – so the bulk pH may be affected. This needs to be considered in the discussion. If there is unneutralized carbonates, then the pH calculation may need to be revisited.

**Response: Thanks for your careful remind, we have checked our calculated results, and no error message was found. The insoluble calcium salt was filtered during the pre-treatment process in sample box of MARGA, the input $Ca^{2+}$ was all obtained from soluble fractions.**

**In the NCP, the PM2.5 pH spanned 2~7 under clean conditions. Some higher PM2.5 pH values appeared, especially at the end of the haze, and were often companied by the cold-front systems from Siberia, the high wind speed can sweep away air pollutants but raise dust in which the crustal ion species are higher, that is the reason why high aerosol pH usually occurred on clean days. We have revised this statement. (Page 8, line 239-240, in the revised manuscript)**

15. Page 9, line 287. "indicating that … particle water". You contradict the statement above, that says composition is the only thing that matters.

**Response: Obviously, the statement caused misunderstanding. What we want to express is that PM$_{2.5}$ pH diurnal variation was both driven by meteorological conditions and aerosol composition. We have revised this statement. (Page 9, line 265-267, in the revised manuscript)**

16. Page 9, line 290-293. "Specifically … in winter". This discussion is not correct. The liquid water content scales with the aerosol mass when RH < 100% (it's in thermodynamic equilibrium), so LWC does not vary independently from aerosol mass. Therefore, you cannot talk about "more" or "less" seeds that dissolves in liquid water. This discussion is, of course, relevant for clouds – but here you talk about aerosol. So please remove this sentence overall.

**Response: The statement has been removed from the revised manuscript.**

17. Page 10, line 302. "Theoretically, … release H+". Although highly soluble, HNO3 does not deliquesce to form aqueous aerosol in the troposphere, therefore it cannot by itself "generate" H+. If there is already some aerosol present, the additional HNO3 is too small to cause H+ to form (unless if you are talking about clouds, where the water is orders of magnitude higher than aerosol). What happens with HNO3 is that it needs to co-condense with NH3 or "bind" with Na, K, Ca to form salts that generally are "neutral", so do not generate H+. Given that with the formation of the salt also generates LWC, this leads to the generally observed increase of pH that you see when NO3 increases. This has been extensively discussed in Guo et al., (2018), Shi et al. (2019) and others. Please revise this section accordingly.

**Response: Thanks for these comments, we learned a lot from these detailed comments. And we rewrote Section 3.3 to interpret the sensitivity tests results. You can see these statements below or in the revised manuscript. (Page 9, line 284-341, in the revised manuscript)**

[revised manuscript text omitted]

18. Page 10, line 307. NH3 "binding H+" doesn't really describe the situation, as the NO3 co-condenses with NH3 to form NH4NO3, either if it is in aqueous or solid phase (in the latter, there is no H+ at all).

**Response: Please see our response to comment 17.**

19. Page 10, line 315. "which might be attributed to limited ALWC". The aerosol has to be acidic for LWC changes to affect pH. So, in this sense, this segment is incorrect and should be deleted.

**Response: The statement has been removed from the revised manuscript.**

20. Page 10, line 315-318. "Compared … ions". Comparing the aerosol and cloud pH can be discussed in terms of the large difference in LWC. Because of that, ions (like HSO4) tend to become SO4-, but to talk about "hydrolysis" of ammonium salts is, in itself, not relevant here. Unless of course if I misunderstood the authors – in which case they should actually clarify (with calculations and an explanation) about what they mean and support these statements with numbers.

**Response: The statement has been removed from the revised manuscript. The reasons why aerosol is acidic is interpreted by thermodynamic equilibrium considerations between $NH_4^+$ and $NH_3$ (Line 287-288 in the revised manuscript).**

21. Page 10, line 320. "has a role … conversion process". There are very few locations in the lower troposphere where you have free H2SO4 in the air, most of the forms found are either HSO4 or SO4 salts. Therefore, one can claim that HNO3 and sulfates have comparable "H+ generation capacity" at best. However, the main issue is the relative volatility of the species. HNO3 does not by itself form aerosol (at lower tropospheric conditions), while H2SO4, HSO4 and SO4 salts always are in aerosol form. That, together with the large hygroscopicity of SO4 salts (with the exception of CaSO4)

is the reason why sulfate-rich aerosol can be much more acidic than nitrate-dominated aerosol. This is discussed in numerous references (e.g., Guo et al., 2018).

**Response: The statement "Compared with NO3-, SO42- has a key role in aerosol acidity due to its stronger ability to provide H+ during the H2SO4→SO42- conversion process" has been removed from the revised manuscript. The reasons why aerosol is acidic is interpreted by the low volatility of SO42-. (Page 10, line 294-295 in the revised manuscript)**

22. Page 10, line 330-331. This is an established fact, please cite the appropriate references here.

**Response: The reference published by Guo et al. (2017) has been cited here. (Page 10, line 322 in the revised manuscript)**

23. Page 10, line 332-334. Although the statement is correct, it does not really describe the situation in the US. In the SE US, for example, the composition of the aerosol is much more like NH4HSO4 than (NH4)2SO4 (e.g., Weber et al., 2016) – especially in current years where SO4 levels are relatively low. Given this, and that the deliquescence humidity of NH4HSO4 is 40%, while for NH4NO3 is 61.8% at 298K (Seinfeld and Pandis, 2016) and the efflorescence point of both salts is very low. Both of these facts suggest that the LWC (per mass of aerosol) in the US should tend to be higher, actually, than for China. Please correct the statement or erase it.

**Response: The statement has been removed from the revised manuscript.**

24. Page 11, line 344-345. This statement goes against all studies to date that I know of. True, RH increases partitioning of species like HNO3, but that is usually with co-condensation of e.g., NH3. If H+ increased in the aerosol phase, that would in itself promote evaporation of HNO3. Given that, and the very large increase of LWC with RH all point to an increase in pH, or decrease in H+.

**Response: This statement has been revised as below. (Page 11, line 331-336 in the revised manuscript)**
RH had different impacts on $PM_{2.5}$ pH in different seasons. In winter, elevated RH could reduce $PM_{2.5}$ pH. However, an opposite tendency was observed in summer. In spring and autumn, RH had little impact on $PM_{2.5}$ pH (Figure 7, S11, S14). Elevated RH can enhance water uptake and promote gas-to-particle conversion, resulting in the increased $H_{air}^+$ and ALWC synchronously for all four seasons. Therefore, the effect of RH on $PM_{2.5}$ pH depends on the differences in the degree of RH's effect on $H_{air}^+$ and RH's effect on ALWC.

25. Page 11, line 345-350. This discussion has, in my opinion, the flawed approach of decoupling LWC from H+. Both do not vary independently, because of thermodynamic equilibrium considerations. You can make such discussions in the cloudwater pH, because indeed water is not bound to the aerosol through a thermodynamic constraint (aw=RH).

**Response: It is not logical to analyze the impact of ALWC on $H_{air}^+$, owing to these two parameters are both the outputs of model and not independent. We removed this statement.**

**Please see the response to comment 17 and 24.**

26. Page 12, line 380. "in the fine mode … excessive NH3". The reason why under 1um size you tend to have small variations in pH is because the aerosol is in thermodynamic equilibrium with the gas phase. The pH would remain the same even if there isn't any excess NH3 (e.g., Fang et al., 2017). Besides, the authors do not define what "excess NH3" even means.

**Response: The interpretation has been revised as you recommended. (Page 11, line 366-368 in the revised manuscript)**

27. Page 13, line 422. "In summary…important". This is not a new finding.

**Response: The statement emphasizes that higher RH and lower T are typical meteorological characteristics of haze events in the NCP, which are favourable for the formation of secondary particles. The statement has been revised. (Page 13, line 409-411 in the revised manuscript)**

28. Page 13, line 447. Replace "decreased" with "increased"?

**Response: This is an unclear expression, which has been revised. (Page 13, line 431-432 in the revised manuscript)**

29. Page 13, line 447-449. "Excess … United States". This is already known (e.g. Guo et al., 2017), but it is good that the authors also find this.

**Response: The work published by Guo et al., (2017) has been added in the revised manuscript. (Page 10, line 322 in the revised manuscript)**

30. Page 14, line 454-453. "pH was still … salts". This is a strongly incorrect statement. The reasons (volatility, and thermodynamic equilibrium considerations) should be stated instead.

**Response: The revised interpretation is "the PM$_{2.5}$ pH was still acidic because 
[revised manuscript text omitted]

[Figure]

**Figure 1.**

[Figure]

**Figure 2.**

[Figure]

**Figure 3**

[Figure]

Figure 4.

[Figure]

Figure 5.

[Figure]

**Figure 6.**

[Figure]

**Figure 7.**


[Figure]

[Figure]

**Figure 8.**

[Figure]

[Figure]

**Figure 9.**